# A Unified Framework for Speculative Decoding with Multiple Drafters as a Bandit

## Abstract

Speculative decoding (SD) has emerged as a promising approach to accelerate inference in large language models (LLMs). This method drafts potential future tokens by leveraging a smaller model, while these tokens are concurrently verified by the target LLM, ensuring only outputs aligned with the target LLM's predictions are accepted. However, the inherent limitations of individual drafters, especially when trained on specific tasks or domains, can hinder their effectiveness across diverse applications. In this paper, we introduce a simple yet efficient unified framework, termed *MetaSD*, that incorporates multiple drafters into the speculative decoding process to address this limitation. Our approach employs multi-armed bandit sampling to dynamically allocate computational resources across various drafters, thereby improving overall generation performance. Through extensive experiments, we demonstrate that our unified framework achieves superior results compared to traditional single-drafter approaches.

## 1 Introduction

Large language models (LLMs) such as GPT-4 (Achiam et al., 2023), Gemini (Google et al., 2023), and Llama (Touvron et al., 2023) have revolutionized real-world applications such as search engine (Reid et al., 2024), coding assistance, and virtual assistants. However, the token-by-token generation process inherent to LLMs often leads to substantial inference times, primarily due to its memory bandwidth bound nature (Patterson, 2004; Shazeer, 2019). Speculative decoding (SD) has emerged as a promising avenue to address this challenge (Leviathan et al., 2023; Chen et al., 2023). Precisely, SD employs a smaller draft model (i.e., drafter) to predict potential future tokens. These tokens are verified concurrently by the target LLM, ensuring only outputs aligned with the LLM's predictions are accepted. This parallel process significantly accelerates the generation process, enabling faster and more efficient text generation.

Recent advancements in SD have primarily focused on architectural and training improvements to enhance the acceptance rate of drafted tokens (Liu et al., 2023; Zhou et al., 2023; Cai et al., 2024; Miao et al., 2024; Sun et al., 2023). Notably, techniques such as batched inference and tree verification (Sun et al., 2023; Miao et al., 2024; Cai et al., 2024) aim to increase the number of accepted tokens by exploring more decoding paths at one step, while training recipes with knowledge distillation (Zhou et al., 2023; Liu et al., 2023) seek to better align the drafter's distribution with that of the target model. However, despite their efficacy in certain tasks, these methods often lack the versatility required to comprehensively cover a wide range of tasks (Liu et al., 2023; Yi et al., 2024). The inherent limitations of relying on a single drafter, with its specific architectural biases and training data, can hinder performance in scenarios with held-out tasks (Detailed motivation is in Section 2.1).

To mitigate the limitations of single-drafter SD, we propose a novel framework that integrates multiple drafters into the process. Our high level idea is to *meta-draft* the optimal drafter among multiple drafters at test-time utilizing the concept of the exploration-exploitation tradeoff (Gittins et al., 2011). Effectively utilizing multiple drafters in a real-world serving system presents several challenges. For instance, imagine a scenario where you have several drafters, each specialized for a different task like translation, summarization, or question answering. Determining which drafter will perform best for a given user query is not always straightforward, especially when the query involves multiple tasks or when the topic evolves during the conversation. Furthermore, the system needs to be efficient and adaptable to varying user loads and traffic patterns, without requiring

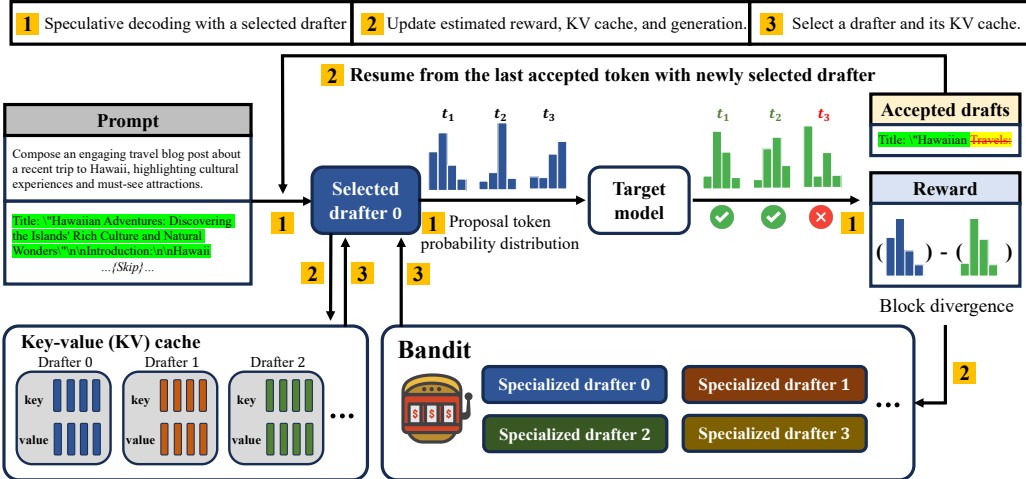

Figure 1: Overview of speculative decoding with multiple drafters in multi-armed bandit (MAB) framework. The example in this figure is from an instance in MT-Bench dataset (Zheng et al., 2024).

constant manual intervention and parameter tuning. Therefore, an ideal system should have low overhead, meaning it should be robust to variations in user scale or network traffic. It must also be scalable at test time, accurately identifying the optimal drafter for a given query, which is often infeasible in advance, as factors like topic can evolve during inference, making pre-selection unreliable. This dynamic nature of language generation necessitates an adaptive approach.

In the domain of recommendation systems, a similar challenge arises where the optimal set of items to present to a user can change based on their evolving interests and interactions (Silva et al., 2022). These systems have successfully employed multi-armed bandit (MAB) algorithms to dynamically adjust recommendations at test time, learning from user feedback to optimize the selection process. Inspired by this approach, we propose a *MetaSD* framework leveraging MAB algorithms to dynamically allocate the optimal drafter among multiple drafters during inference time (Figure 1). This approach enables the system to learn and adapt to the relative performance of each drafter on-the-fly, enabling faster inference. Our key contributions include:

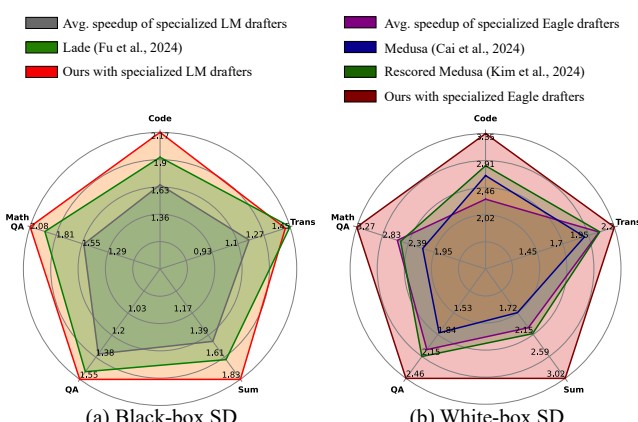

Figure 2: Comparison of average speedup ratios achieved by various SD methods relative to standard autoregressive greedy decoding on a single NVIDIA A100 GPU. The target model is Vicuna 7B v1.3. (a) Results for black-box methods. (b) Results for white-box methods. Detailed description for experimental settings are in **Section 4**.

- We introduce a simple yet efficient framework, termed MetaSD, for incorporating multiple drafters into SD, exploring both black-box approaches where drafters operate independently with access only to the target LLM's predictions and white-box approaches where drafters leverage internal latent features of the target LLM (**Section 2**).

- We establish theoretical upper bounds on the performance of our proposed framework, providing insights into its convergence properties and potential benefits (**Section 3**).

- We demonstrate through extensive experiments that our framework achieves superior inference speed compared to existing single-drafter methods (Figure 2; **Section 4**).

## 2 PROBLEM STATEMENT

### 2.1 MOTIVATION

Speculative decoding (SD) employs a *draft-verify-accept* paradigm for faster inference. A drafter $\mathcal{M}_q$, which is smaller than the target LLM $\mathcal{M}_p$, drafts the future tokens $\{x^{l+1:l+N_{max}}\}$ based on the input sequence $x^{1:l}$. The target LLM assesses each token $x^{l+j}$ ($j = 1, \ldots, N_{max}$) to determine whether $p(\cdot|x^{1:l+j-1})$ is aligned with its own predictions $q(\cdot|x^{1:l+j-1})$. Only the tokens aligned with the LLM's own predictions are accepted, ensuring the lossless generation (Details in Appendix D).

Despite its advancements, existing works often rely on a single drafter. This reliance can limit the effectiveness of SD, as the drafter's performance is inherently tied to its training data (Yi et al., 2024; Liu et al., 2023). In scenarios where the drafter's strengths do not align well with the task at hand, its predictions may be less accurate, leading to fewer accepted tokens and diminished speedup benefits of SD. As Table 1 shows, a drafter trained on a specific language pair exhibits significantly higher speedup on that pair compared to others, highlighting the need for a more adaptive approach. Therefore, integrating multiple heterogeneous drafters into the SD framework can potentially address this limitation. By leveraging a pool of drafters, the system can dynamically adapt to varying tasks and input contexts,

Table 1: Speedup ratio relative to the standard autoregressive greedy decoding on various multilingual datasets following Yi et al. (2024) where target model is Vicuna 7B v1.3 and the drafter is decoder-only 68M language model: Japanese (Ja)→English (En) (Morishita et al., 2022), Russian (Ru)→En, German (De)→En (Bojar et al., 2016), French (Fr)→En (Bojar et al., 2014), and Chinese (Zh)→En (Barrault et al., 2019). Evaluations are conducted with a NVIDIA A5000 GPU.

| Dataset | Ja-drafter | Ru-drafter | De-drafter | Fr-drafter | Zh-drafter |
|---|---|---|---|---|---|
| Ja →En | **1.757** ● | 1.109 | 1.012 | 1.018 | 1.154 |
| Ru →En | 1.055 | **1.817** ● | 0.995 | 0.963 | 1.036 |
| De →En | 1.098 | 1.369 | **2.360** ● | 1.036 | 1.099 |
| Fr →En | 1.106 | 1.445 | 1.108 | **2.135** ● | 1.122 |
| Zh →En | 1.198 | 1.086 | 1.021 | 1.023 | **1.516** ● |

selecting the most suitable drafter for each situation.[1] From a theoretical and practical viewpoint, the integration of multiple drafters into SD raises several research questions:

1. How to design an efficient and adaptive mechanism for selecting the best drafter at each generation step, considering the exploration-exploitation tradeoff?

2. How to seamlessly incorporate multiple drafters for meta-drafting while minimizing any additional computational overhead?

3. Can we provide theoretical guarantees on the performance of a multi-drafter SD system, ensuring comparable speedup to using single optimal drafter?

To address these challenges, we draw inspiration from the field of multi-armed bandits (MAB). In the MAB framework, an agent repeatedly chooses an action among different choices (arms), each with an unknown reward distribution, aiming to maximize its cumulative reward over time. This closely parallels our problem, where each drafter can be seen as an arm, and the reward is related to the number of accepted tokens or the overall speedup achieved (Algorithm 1). MAB's inherent efficiency and online learning capabilities align well with the requirements of a robust and adaptive multi-drafter SD system. MAB algorithms offer a principled way to balance exploration (trying out different drafters) and exploitation (using the seemingly best drafter) to identify the optimal drafter for each generation step, adapting to the changing context with minimal additional compute costs.

### 2.2 PROBLEM FORMULATION

**Multi-armed bandit (MAB)**  MAB framework addresses an online learning scenario where, at each round $t$, an agent takes an action by choosing an arm $a_t \in [K]$ and receives a reward $r_t$ from the environment. The goal of MAB is to design an algorithm that maximizes the expectation of cumulative reward $\mathbb{E}[\sum_{t=1}^{T} r_t]$ throughout a total of $T$ rounds. To achieve this, one can aim to design an optimal policy $\pi^\star$ to minimize the pseudo-regret, defined as: $\text{REG}(\pi, T) = \sum_{t=1}^{T} \mathbb{E}[r_{a_t^\star}] - \mathbb{E}[r_{a_t}]$. Here, $a_t$ denotes the action chosen in round $t$ by the policy $\pi$ and $a_t^\star$ represents the optimal action in round $t$ which yields the highest expected reward. For a more comprehensive review, we refer the reader to Lattimore & Szepesvári (2020).

---

[1]Further motivation can be found in the Appendix B.

---

**Algorithm 1:** MetaSD

---

INPUT : Drafter pool $[K]$, target model, initial prompt sequence $x^{1:l}$, target sequence length $B$.

1: $t \leftarrow 0$
2: **while** $l < B$ **do**
3:     Meta-draft the drafter $i$ in drafter pool $[K]$ following the bandit
4:     Execute one SD step with drafter $i$ and target model given $x^{1:l}$
5:     Compute the block divergence between drafter $i$'s predictions and target model's predictions as the reward (Section 2.3)
6:     Update the sequence length with the number of accepted tokens from the draft $N_{acc}(i, t)$:
    $l, t \leftarrow l + N_{acc}(i, t) + 1, t + 1$
7:     Update the bandit
8: **end while**

---

**MetaSD: SD with multiple drafters as a MAB problem** We formalize the integration of multiple drafters into SD as a MAB problem, termed as MetaSD framework. Each SD process, consisting of drafting, verifying, and accepting tokens, corresponds to one round in the MAB setting (Algorithm 1). At round $t$, a drafter $a_t$ is selected from a pool of heterogeneous drafters $[K]$. The round concludes when all $B$ tokens have been generated. While inspired by classical bandit problems, our MetaSD framework exhibits key distinctions. Unlike classical bandits with a fixed number of rounds, MetaSD operates under a fixed target sequence length $B$ and the number of total rounds $T$ is stochastic which depends on the policy. Although switching between drafters may incur costs such as prefill cost for tokens and KV cache I/O, we empirically observe that it is negligible in the most of our experiments. Furthermore, for the large scale scenario where switching cost might not be negligible anymore, we provide a detailed discussion with a practical algorithm in Section H.2. While the generated tokens can follow a non-stationary distribution, we assume stationarity within a single turn between the user and the LLM for theoretical analysis. This assumption is reasonable as it allows our framework to be applied with re-intialization for each new query, even in a multi-turn conversation, effectively handling the potential non-stationarity across different queries. In the experiments, MetaSD is implemented with re-initialization for every query.

## 2.3 REWARD DESIGN

Ideally, the reward in the MetaSD framework should be informative enough to effectively guide the bandit algorithm towards optimal speedup. One straightforward and readily available choice is the block efficiency (BE), which quantifies the number of mean accepted tokens until a given round (Sun et al., 2023; Chen et al., 2023; Kim et al., 2024). Formally, we define the BE reward for drafter $i$ in round $t$ as: $r_{i,t}^{BE} := N_{acc}(i, t)/N_{max}$, where $N_{max}$ is predefined maximum draft length and $N_{acc}(i, t)$ is number of accepted tokens in the $t$-th verification stage. While the BE reward provides a direct measure of a drafter's immediate success, it depends on the underlying acceptance rate, denoted as $\alpha_i$. As shown in Leviathan et al. (2023), this acceptance rate is intrinsically linked to the distance between two probability distributions $p$ and $q_i$. This implies that by estimating $\alpha_i$, we can potentially obtain more informative feedback at each round. To leverage this insight, we propose a new reward, coined as block divergence (BD) reward, which estimates the normalized expected number of accepted tokens by utilizing empirical mean of the acceptance rate.

**Definition 1** (Block divergence reward). *Let $t$ be the current round, $i$ be the drafter index, and $l(t)$ be the number of input tokens for the target model at round $t$. Denote $d_{TV}(p^{l(t)}, q_i^{l(t)}) = \frac{1}{2}\|p^{l(t)} - q_i^{l(t)}\|_1$ as the total variation (TV) of two probability measures $p^{l(t)}$ and $q_i^{l(t)}$ from the target model and the drafter $i$ given $x^{1:l(t)}$, respectively. Then, BD reward is defined as follows:*

$$r_{i,t}^{BD} = \frac{1}{N_{max}} \sum_{j=0}^{N_{max}-1} \left(1 - d_{TV}\left(p^{l(t)+j}, q_i^{l(t)+j}\right)\right). \tag{1}$$

While Leviathan et al. (2023) assume a fixed acceptance rate for the $j$-th candidate in their analysis, we relax this assumption and consider a more general scenario where the acceptance rate for each token follows stationary distribution with mean $\alpha_i \in (0, 1)$ for each drafter $i \in [K]$. Then, one can

Table 2: Reward statistics for BE and BD rewards, collected using autoregressive decoding with the same Japanese dataset and drafter configurations as in Table 1.

| Reward statistics | BE reward | | | | | BD reward | | | | |
|---|---|---|---|---|---|---|---|---|---|---|
| | Ja-drafter | Ru-drafter | De-drafter | Fr-drafter | Zh-drafter | Ja-drafter | Ru-drafter | De-drafter | Fr-drafter | Zh-drafter |
| Ratio of the number of zero rewards | 0.503 | 0.678 | 0.721 | 0.743 | 0.681 | - | - | - | - | - |
| Mean of rewards | 0.232 | 0.099 | 0.081 | 0.074 | 0.106 | 0.488 | 0.294 | 0.317 | 0.288 | 0.326 |
| Variance of rewards | 0.093 | 0.032 | 0.024 | 0.023 | 0.037 | 0.044 | 0.026 | 0.032 | 0.029 | 0.034 |

observe two reward designs are linked by $\mathbb{E}[r_{i,t}^{BE}] = \frac{1-\alpha_i^{N_{max}}}{N_{max}(1-\alpha_i)}\mathbb{E}[r_{i,t}^{BD}]$ (proof in Lemma 5). As both $\mathbb{E}[r_{i,t}^{BD}]$ and $\mathbb{E}[r_{i,t}^{BE}]$ is monotonically increasing with respect to $\alpha_i$, maximizing the BD reward aligns with the goal of SD, which is to maximize the number of accepted of tokens. We demonstrate that the BD reward empirically and theoretically facilitates the generalization of the MetaSD framework compared to the BE reward, particularly in terms of bandit algorithm performance. To begin, we compare the BD and BE rewards using the following theorem.

**Theorem 1** (Informal). *Under the stationary environment, for any reward design $r_i$ with $\mu_i = \mathbb{E}[r_i]$, $i^\star = \arg\max \alpha_i$, and $\Delta_i = \mu_i^\star - \mu_i$, we define the feedback signal for each suboptimal arm $i \neq i^\star$ as*

$$R(r_i) := \frac{\max(\mathrm{Var}[r_i], \mathrm{Var}[r_{i^\star}])}{\Delta_i^2}. \tag{2}$$

*Then, for most of the scenarios, $R(r_i^{BD}) < R(r_i^{BE})$.*

Theorem 1 demonstrates that the BD reward provides a more informative feedback signal than the BE reward. This signal, defined in eq. 2, plays a crucial role in determining the performance of bandit algorithms. Intuitively, distinguishing two distributions is easier when their expectations are further apart or their variances are smaller. In the context of bandit algorithms, this translates to a smaller regret due to decreased exploration costs. A less noisy feedback signal allows the algorithm to more quickly and accurately identify the optimal arm, reducing the need for extensive exploration of suboptimal arms, as it provides a clearer and more reliable signal for decision-making. Consequently, Theorem 1 implies that we can achieve better performance with bandit algorithms by using the BD reward. In Section G.2, we provide the formal statement of Theorem 1 along with two lemmas providing statistics of the BE reward (Lemma 3) and the BD reward (Lemma 4).

We empirically validate our theoretical analysis regarding the effectiveness of the BD reward compared to the BE reward. For the experiment, we use the same Japanese dataset and drafter configurations as in Table 1, employing autoregressive decoding to collect BE and BD rewards at each step without actual speculative execution. Table 2 reveals striking differences. The BD reward exhibits larger gaps between the expected rewards of the best and suboptimal drafters ($\Delta_i$), while also demonstrating consistently lower variance across all drafters. Consequently, the BD reward has smaller feedback signal $R$ and we can expect using the BD reward leads to more stable learning and faster convergence of the MAB algorithm, enabling faster identification of the optimal drafter. Further explanation is in Section F.6 with Figure 5.

## 3 METHOD

This section presents our main method, MetaSD-UCB, which is designed to guarantee the optimal policy for MetaSD. The main challenge arises from the fact that existing regret bounds does not fit into the objective of SD anymore. Moreover, we have to consider stochastic nature of total number of rounds $T$ with the fixed target sequence length $B$, as opposed to the classical bandit settings where $T$ is fixed. This necessitates us to design a new regret objective and we establish strong regret bounds can still be achieved under this new objective. At the end of this section, we briefly discuss potential extensions, incorporating switching costs between drafters and addressing non-stationary reward distributions.

### 3.1 ALGORITHM

**MetaSD-UCB** We introduce MetaSD-UCB in Algorithm 2, where we combine UCB algorithm (Auer, 2002) in conjunction with the BD reward design to minimize regret. Under the

---

**Algorithm 2:** MetaSD-UCB

---

INPUT Drafter pool $[K]$, initial prompt sequence $x^{1:l}$, target sequence length $B$, exploration
    strength hyperparameter $\beta$.

1: $t \leftarrow 0$
    /* Phase 1: Meta-draft each drafter in $[K]$ once and do one round of speculative decoding. */
2: **for** $i \in [K]$ **do**
3:    Do one round of SD with drafter $i$ and obtain $N_{acc}(i,t)$, $r_{i,t}$ (by eq. 1)
4:    $\hat{\mu}_{i,t}, n_i, l, t \leftarrow r_{i,t}, 1, l + N_{acc}(i,t) + 1, t + 1$
5: **end for**
    /* Phase 2: Meta-draft the draft following the UCB bandit until target sequence length $B$ */
6: **while** $l < B$ **do**
7:    $a_t \leftarrow \arg\max_{i \in [K]} \hat{\mu}_{i,t} + \beta\sqrt{\frac{2\ln t}{n_i}}$
8:    Do one round of SD with drafter $a_t$ and obtain $N_{acc}(a_t,t)$, $r_{a_t,t}$ (by eq. 1)
9:    $\hat{\mu}_{a_t,t}, n_{a_t}, l, t \leftarrow \frac{\hat{\mu}_{a_t,t}*n_{a_t}+r_{a_t,t}}{n_{a_t}+1}, n_{a_t} + 1, l + N_{acc}(a_t,t) + 1, t + 1$
10: **end while**

---

stationary environments, UCB achieves optimal log-linear regret (Lattimore & Szepesvári, 2020).
However, our problem has two key distinctions which prevent direct application of prior analysis.
First, the total number rounds required to generate all tokens (i.e., target sequence length) becomes
stochastic. Secondly, minimizing naive regret objective does not guarantee the optimal performance
(Section G.3). This arises due to the nature of SD, where the performance of the algorithm is deter-
mined by total number of rounds until EOS token (or reaching the maximum token length supported
by the target LLM). In order to better representing actual speedup, we introduce a novel regret
objective for MetaSD, defined as follows.

**Definition 2.** *Denote $\tau(\pi, B)$ as the number of total rounds of bandit policy $\pi$ with target sequence
length $B$ and $\pi^\star$ as the optimal policy which satisfies $\pi^\star = \arg\min_\pi \mathbb{E}[\tau(\pi, B)]$. Then, regret
objective of MetaSD with policy $\pi$ becomes:*

$$\text{REG}(\pi, B) = \mathbb{E}\left[\tau(\pi, B)\right] - \mathbb{E}[\tau(\pi^\star, B)]. \tag{3}$$

Minimizing eq. 3 is equivalent to maximizing expected number of accepted tokens. This can be seen
by observing that the target sequence length $B$ is consumed by the total number of rounds $\tau(\pi, B)$
plus the total number of accepted tokens across all rounds: $B = \tau(\pi, B) + \sum_{t=1}^{\tau(\pi,B)} N_{acc}(i,t)$.
Consequently, minimizing the regret (eq. 3) is directly proportional to maximizing the expected
number of accepted tokens, which aligns with the objective of SD.

## 3.2 REGRET UPPER BOUND FOR METASD-UCB

We establish that MetaSD-UCB achieves the same level of optimality as the standard UCB (Auer,
2002) by proving that the regret in eq. 3 exhibits a logarithmic growth with respect to the target
sequence length $B$, which is stated in the following theorem.

**Theorem 2** (Regret upper bound on MetaSD-UCB). *Denote $\Delta(\alpha_i) = \alpha_{i^\star} - \alpha_i$, where $i^\star$ is the index
of the drafter with the largest $\alpha_i$. Then, under i.i.d assumption of $\alpha_{i,t}$ (details in Assumption 1) and
using the BD reward, there exists a constant $C, C' > 0$ such that following bound holds:*

$$\text{REG}(\pi, B) < \sum_{i \neq i^\star} \frac{8}{(N_{max})\Delta(\alpha_i)^2}(\ln B + \ln(\ln(\sum_{i \neq i^\star} \frac{1}{\Delta(\alpha_i)^2})) + C') + C. \tag{4}$$

In Section G.4, we prove the log-linear regret upper bound holds with general reward design but
with the higher constant factor $8/\Delta(\alpha_i)^2$. The improvement in eq. 4 stems directly from using the
BD reward in Algorithm 2. Since the number of observations within each round grows with $N_{max}$,
the variance of the BD reward is effectively reduced by a factor of $N_{max}$. This, in turn, leads to a
smaller constant term in the regret upper bound compared to using the BE reward. The following
corollary captures this observation:

**Corollary 1** (Informal). *In most scenarios, the regret upper bound in eq. 4 is tighter than the regret
upper bound obtained when using the BE reward with MetaSD-UCB.*

A complete proof of Theorem 2 and a formal statement of Collorary 1 with the proof are in Section G.5.

### 3.3 EXTENSIONS OF METASD FRAMEWORK

**Switching costs**   In practical implementations, switching between drafters at each round incurs a computational cost due to the need to recalculate previous KV-cache values for the new drafter. This aligns with the concept of bandits with switching costs (Banks & Sundaram, 1994). However, unlike traditional settings where a fixed cost is incurred per switch, the cost in MetaSD is proportional to the number of unprocessed tokens in the current block. To address this, we propose Algorithm 4 with Sequential Halving (SH) (Karnin et al., 2013), designed specifically for this scenario. A detailed analysis along with theoretical guarantees on its performance is provided in Section H.1.

**Non-stationary environment**   Our prior analysis assumes stationary reward distributions, where the reward feedback for each drafter follows a fixed distribution. However, in certain scenarios, the reward distribution can be non-stationary. For instance, in long-context generation, the optimal drafter might change as the topic or style of the generated text evolves. Despite this challenge, our MetaSD framework remains applicable by leveraging non-stationary bandit algorithms. These algorithms are designed to adapt to changing reward distributions, enabling the system to continuously learn and adjust its drafter selection strategy. Detailed discussions for non-stationary algorithms within the context of MetaSD are in Section H.2.

## 4 EXPERIMENT

### 4.1 EXPERIMENTAL SETUP

**Models**   We adopt Vicuna 7B (Chiang et al., 2023) as our target LLM for both black-box and white-box SD. The distinction between two paradigms lies in the drafter's access to the target LLM's internal representations. Black-box drafters operate independently, with access only to the final logit of the target LLM. In contrast, white-box drafters can leverage intermediate activations and hidden states within the target LLM. For black-box SD, we utilize Vicuna 68M (Yang et al., 2024) as the base architecture for our independent drafters. Each drafter is trained on a distinct task-specific dataset to ensure heterogeneity. Following established practices (Kim & Rush, 2016; Zhou et al., 2023; Cai et al., 2024; Yi et al., 2024), the training data for these drafters is generated via self-distillation from the target LLM. For white-box SD, we integrate Eagle (Li et al., 2024) into the target Vicuna 7B to enable white-box SD. Similar to the black-box setting, multiple Eagle drafters share the same underlying architecture but are fine-tuned on distinct task-specific datasets generated via self-distillation from the target LLM. To ensure a fair comparison for the baseline, we introduce the One-size Fits All (OFA) drafter, which is trained on a mixed dataset spanning all tasks. Further details on the training procedures and datasets used for both black-box and white-box drafters are provided in Appendix F.

**Number of drafts $N_{max}$**   For black-box SD, we employ speculative sampling (SpS) (Chen et al., 2023), generating one draft candidate per drafter, termed as MetaSpS. For multi-draft methods like Medusa (Cai et al., 2024) and Eagle (Li et al., 2024), we adhere to their original settings with a tree-attention mechanism. We employ the same tree structure for multiple Eagle drafters described in Li et al. (2024), termed as MetaEagle. Unless explicitly stated otherwise, all approaches utilize a maximum of 5 drafts ($N_{max} = 5$).

**Evaluation**   We conduct evaluations using a NVIDIA A5000, A6000, and A100 GPU under greedy decoding settings. We re-initialize the bandit for each new query, even within multi-turn conversations. Two types of scenarios are evaluated:

1. Diverse task: We evaluate on a diverse range of tasks, including coding (Code) from MT-Bench (Zheng et al., 2024), summarization (Sum) on CNN/Daily (Hermann et al., 2015), De-En translation (Trans) on WMT16 (Bojar et al., 2016), natural question answering (QA) (Kwiatkowski et al., 2019), and mathematical reasoning (Math) on GSM8K (Cobbe et al., 2021). The datasets are randomly shuffled to create a non-stationary environment.

Table 3: (Black-box SD) Speedup ratio relative to standard autoregressive greedy decoding on various datasets, comparing single specialized independent drafters, other methods (PLD (Saxena, 2023) and Lookahead (Fu et al., 2024)), and bandit-based drafter selection (Rand (uniformly random), EXP3 (Auer et al., 2002), SH (Karnin et al., 2013), UCB). Evaluations are conducted with a single NVIDIA A6000 GPU under greedy decoding settings. Drafter specializations: 1: Code, 2: Translation, 3: Summarization, 4: QA, 5: Math.

| Speedup | SpS with specialized drafters | | | | | SpS | Other methods | | Bandit in MetaSpS | | | |
|---|---|---|---|---|---|---|---|---|---|---|---|---|
| | Drafter1 | Drafter2 | Drafter3 | Drafter4 | Drafter5 | OFA | PLD | Lookahead | Rand | EXP3 | SH | UCB |
| Code | **2.437** ● | 1.224 | 1.565 | 1.814 | 1.687 | **2.435** ● | 1.923 | 1.542 | 1.640 | 1.919 | 2.148 | 2.300 |
| Trans | 0.991 | **2.076** ● | 1.000 | 1.019 | 0.950 | 1.032 | 1.076 | 1.133 | 1.150 | 1.217 | 1.422 | **1.587** ● |
| Sum | 1.513 | 1.087 | **2.133** ● | 1.510 | 1.387 | **2.501** ● | 1.275 | 1.526 | 1.429 | 1.606 | 1.812 | 1.971 |
| QA | 1.332 | 1.200 | 1.343 | **1.960** ● | 1.252 | 1.267 | 1.178 | 1.208 | 1.294 | 1.437 | 1.599 | **1.711** ● |
| Math | 1.483 | 1.228 | 1.378 | 1.486 | **2.454** ● | 1.571 | 1.653 | 1.533 | 1.471 | 1.690 | 2.144 | **2.280** ● |

Table 4: (White-box SD) Speedup ratio relative to standard autoregressive greedy decoding on various datasets, comparing single specialized drafters, other methods (blockwise parallel decoding (BPD) (Stern et al., 2018), Medusa, Rescored-BPD (R-BPD) and Rescored-Medusa (Kim et al., 2024)), and bandit-based drafter selection. Evaluations are conducted with a single NVIDIA A100 GPU under greedy decoding settings.

| Speedup | Specialized Eagle drafters | | | | | Eagle | Other methods | | | | Bandit in MetaEagle | | | |
|---|---|---|---|---|---|---|---|---|---|---|---|---|---|---|
| | Eagle1 | Eagle2 | Eagle3 | Eagle4 | Eagle5 | OFA | BPD | R-BPD | Medusa | R-Medusa | Rand | EXP3 | SH | UCB |
| Code | **3.934** ● | 1.303 | 1.776 | 2.150 | 2.427 | **3.776** ● | 1.963 | 2.146 | 2.661 | 2.822 | 2.310 | 2.858 | 3.650 | 3.724 |
| Trans | 1.750 | **2.496** ● | 2.281 | 2.131 | 1.714 | 2.143 | 1.626 | 1.442 | 1.909 | 2.056 | 2.036 | 2.171 | 2.225 | **2.318** ● |
| Sum | 1.707 | 1.507 | **3.382** ● | 2.005 | 1.589 | 2.640 | 1.509 | 1.455 | 1.723 | 2.136 | 2.261 | 2.261 | 2.801 | **3.057** ● |
| QA | 1.842 | 1.579 | 2.181 | **2.916** ● | 1.783 | 2.446 | 1.489 | 1.468 | 1.817 | 2.154 | 2.006 | 2.128 | 2.466 | **2.641** ● |
| Math | 2.584 | 1.618 | 2.337 | 2.433 | **3.903** ● | 3.049 | 1.696 | 1.696 | 2.142 | 2.519 | 2.449 | 2.811 | 3.339 | **3.520** ● |

2. Multilingual task: We assess the effectiveness in handling multilingual scenarios by evaluating on the multilingual tasks presented in Table 1, following the Yi et al. (2024).

The chosen tasks represent a diverse range of applications. Code involves generating text within the constraints of a formal programming language, while Math often requires manipulating symbolic expressions and numerical values. Multilingual tasks introduce challenges related to vocabulary space and token distribution, necessitating drafters tailored to specific language pairs. Summarization highlights the dependency of generation on the input space, where drafters must effectively capture and condense information from diverse articles. Finally, QA represents a core natural language understanding task, requiring drafters to comprehend and extract information from complex contexts. For both settings, we utilize a pool of 5 heterogeneous drafters in the MetaSD framework.

## 4.2 MAIN RESULT

**Diverse task (black-box SD)** Table 3 presents the speedup ratios achieved by various methods on a diverse set of tasks using black-box SD. As expected, specialized drafters excel on their respective tasks, as indicated by the highlighted best results. However, their performance suffers significantly on unrelated tasks, demonstrating the limitations of relying on a single drafter. Our MetaSpS-UCB consistently achieves competitive speedup compared to both specialized drafters and other state-of-the-art techniques across most tasks. While the OFA drafter/Eagle perform well, our MetaSD framework mostly outperforms OFA. This highlights the effectiveness of our adaptive selection mechanism in leveraging the strengths of multiple drafters to optimize performance across diverse scenarios. Notably, MetaSpS-UCB reaches the near-optimal performance of the corresponding specialized drafter on several tasks, demonstrating its ability to dynamically identify and utilize the most suitable drafter for the given context. Furthermore, when comparing MetaSpS-UCB to other bandit such as SH and EXP3, considering switching costs and non-stationarity, we observe that MetaSpS-UCB consistently outperforms others. This supports the theoretical advantages of UCB.

**Diverse task (white-box SD)** Table 4 presents the results for white-box SD with MetaEagle, utilizing EAGLE drafters integrated into the target LLM. Similar to the black-box setting, specialized drafters excel on their designated tasks but struggle on others. MetaEagle-UCB again demonstrates competitive performance, consistently achieving high speedup ratios across all tasks and often out-

performing other bandit-based selection strategies. This highlights the adaptability and effectiveness of our proposed framework in both black-box and white-box SD scenarios.

**Multilingual task (black-box SD)**  Table 5 shows the speedup ratios on multilingual tasks.    Consistent with the observations in diverse tasks, specialized drafters demonstrate superior performance on their matched language pairs. MetaSps-UCB consistently outperforms other bandit-based selection strategies (EXP3, SH) and remains competitive even with specialized drafters, showcasing its ability to adapt effectively to varying language pairs and achieve notable speedup gains in multilingual scenarios.

Table 5: Speedup ratio relative to standard autoregressive greedy decoding on various multilingual datasets, comparing single specialized drafters to bandit-based drafter selection (EXP3, SH, UCB). Evaluations are conducted with a single NVIDIA A5000 GPU under greedy decoding settings. Drafter specializations: 1: Ja →En, 2: Ru →En, 3: De →En, 4: Fr →En, 5: Zh →En.

| Speedup | SpS with specialized drafters | | | | | Bandit in MetaSpS | | |
|---|---|---|---|---|---|---|---|---|
| | Drafter1 | Drafter2 | Drafter3 | Drafter4 | Drafter5 | EXP3 | **SH** | UCB |
| Ja → En | **1.757** ● | 1.109 | 1.012 | 1.018 | 1.154 | 1.260 | **1.368** ● | 1.161 |
| Ru → En | 1.055 | **1.817** ● | 0.995 | 0.963 | 1.036 | 1.259 | 1.403 | **1.503** ● |
| De → En | 1.098 | 1.369 | **2.360** ● | 1.036 | 1.099 | 1.472 | 1.656 | **1.693** ● |
| Fr → En | 1.106 | 1.445 | 1.108 | **2.135** ● | 1.122 | 1.506 | 1.607 | **1.775** ● |
| Zh → En | 1.198 | 1.086 | 1.021 | 1.023 | **1.516** ● | 1.204 | 1.297 | **1.369** ● |

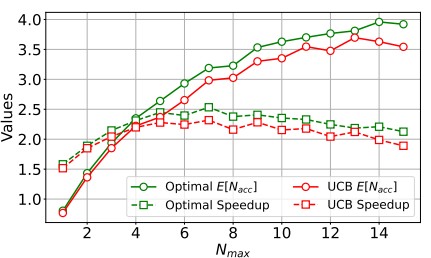

Figure 3: Ablations on $N_{max}$. 'Optimal' represents the optimal drafter and UCB denotes MetaSps-UCB with BD reward.

Table 6: Average of speedup ratio comparing the BE and BD rewards for MetaSD-UCB with both SpS and EAGLE drafters over 3 different runs.

| Task | MetaSpS-UCB | | MetaEagle-UCB | |
|---|---|---|---|---|
| | BE | BD | BE | BD |
| Code | $2.052_{\pm 0.004}$ | $2.231_{\pm 0.006}$ ● | $3.590_{\pm 0.017}$ | $3.661_{\pm 0.003}$ ● |
| Trans | $1.465_{\pm 0.004}$ | $1.554_{\pm 0.001}$ ● | $2.228_{\pm 0.009}$ ● | $2.201_{\pm 0.001}$ |
| Sum | $1.770_{\pm 0.002}$ | $1.929_{\pm 0.001}$ ● | $3.038_{\pm 0.005}$ | $3.043_{\pm 0.001}$ ● |
| QA | $1.591_{\pm 0.003}$ | $1.698_{\pm 0.001}$ ● | $2.629_{\pm 0.003}$ ● | $2.608_{\pm 0.001}$ |
| Math | $1.992_{\pm 0.003}$ | $2.238_{\pm 0.002}$ ● | $3.461_{\pm 0.009}$ | $3.515_{\pm 0.001}$ ● |

### 4.3 Ablation study

**Draft length**  To analyze the impact of draft length on the performance of MetaSps-UCB with the BD reward, we conduct experiments on the Code task using 5 drafters following the same setting in Table 3. The maximum draft length $N_{max}$ is varied to measure the resulting speedup. Figure 3 shows that increasing the draft length initially leads to higher $\mathbb{E}[N_{acc}]$ and speedup due to the increased parallelism in token generation. However, beyond a certain threshold, further increasing the draft length yields diminishing returns and can even decrease performance due to the higher probability of rejection and the associated overhead.

**Reward design**  To assess the impact of our reward function choice, we compare the performance of MetaSD using both BE and BD rewards. In the black-box setting, BD consistently outperforms BE across various tasks, as shown in Table 6. This highlights the importance of utilizing a reward signal that accurately captures the underlying dynamics of the SD process. However, for the MetaEagle-UCB (white-box) setting, both BE and BD rewards exhibit comparable performance. We hypothesize that this is due to Eagle's tree-attention mechanism, which effectively explores multiple decoding paths and implicitly captures the divergence between the drafter and target LLM distributions. This suggests that in white-box settings with multi-path exploration, the choice of reward function might have a less significant impact on the overall performance. Nonetheless, the consistent superiority of BD in the black-box setting underscores its potential benefits in scenarios where such multi-path exploration is not available.

**Best arm ratio**  To further analyze the behavior of MetaSD, we examine the best arm ratio, which represents the frequency of selecting the optimal drafter for a given task. Figure 4 illustrates how this ratio evolves over speculative decoding rounds, comparing different reward types (BE and BD) and bandit algorithms (SH, EXP3, UCB) for both MetaSpS (black-box SD) and MetaEagle (white-box SD). Across all configurations, UCB consistently identifies the best arm more rapidly than other bandit algorithms. This trend is particularly pronounced in the MetaSpS setting. Additionally, the BD reward generally leads to a higher best arm ratio compared to BE, suggesting that BD provides

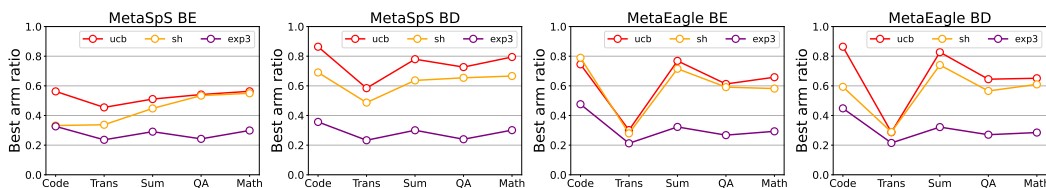

Figure 4: Best arm ratio over rounds for various configurations. (Left) MetaSpS (black-box SD) with BE and BD rewards. (Right) MetaEagle (white-box SD) with BE and BD rewards.

a more informative signal for drafter selection. This observation aligns with our earlier hypothesis that BD better captures the underlying dynamics of SD. Overall, the combination of UCB with the BD reward exhibits the most rapid convergence towards the optimal drafter.

**Temperature sampling** We investigate the impact of temperature sampling on MetaSpS performance. Table 7 presents the speedup ratios achieved with temperature sampling with temperature 0.7 on an NVIDIA A6000 GPU. Consistent with the trends observed in our main experiments with greedy decoding, MetaSD continues to achieve competitive speedup.

Table 7: Speedup ratio with temperature sampling as temperature is set to 0.7 over a NVIDIA A6000 GPU.

| Dataset | SpS with specialized drafters | | | | | Bandit |
|---|---|---|---|---|---|---|
| | Drafter1 | Drafter2 | Drafter3 | Drafter4 | Drafter5 | UCB |
| Code | **2.250** | 1.215 | 1.379 | 1.532 | 1.513 | 1.896 |
| Trans | 1.086 | **1.886** | 1.096 | 1.130 | 1.078 | 1.431 |
| Sum | 1.461 | 1.165 | **1.874** | 1.463 | 1.353 | 1.744 |
| QA | 1.316 | 1.193 | 1.324 | **1.776** | 1.272 | 1.534 |
| Math | 1.450 | 1.258 | 1.355 | 1.616 | **2.379** | 2.046 |

## 5 DISCUSSION

**Regret upper bound for MetaSD-UCB** Theorem 2 provides a regret upper bound for MetaSD-UCB, demonstrating that the number of rounds required to identify the optimal drafter is inversely proportional to the predefined draft length $N_{max}$. This aligns with the intuition that longer drafts provide more information about the relative performance of each drafter, leading to faster convergence towards the optimal choice. The logarithmic dependence on the target sequence length $B$ further highlights the efficiency of MetaSD-UCB in minimizing regret. These theoretical guarantees are supported by our empirical observations, where MetaSD-UCB consistently demonstrates strong performance and rapid convergence towards the best-performing drafter.

**Memory bandwidth bound** A potential concern with our MetaSD framework is the increased memory bandwidth requirement due to loading multiple drafter models. However, our approach incurs minimal memory overhead. By storing all drafter weights in GPU DRAM, we avoid frequent accesses to slower system memory, which are a primary bottleneck for LLMs. For instance, with a 7B target LLM and float16 precision, our MetaEagle framework utilizes at most 19GB of GPU DRAM during generation, compared to 17GB for a single Eagle drafter. This represents only a small increase in memory usage, and importantly, it does not increase the memory bandwidth requirement during inference since only one drafter is active at a time.

## 6 CONCLUSION

In this paper, we introduce a unified framework for incorporating multiple drafters into speculative decoding, addressing the limitations of single-drafter approaches. We formalize this problem as a multi-armed bandit problem, termed as MetaSD, and proposed MetaSD-UCB, a novel algorithm that leverages the Upper Confidence Bound (UCB) principle to dynamically select the optimal drafter at each generation step. We also provide theoretical guarantees on the performance of MetaSD-UCB, establishing its effectiveness in achieving near-optimal speedup even with a stochastic number of rounds. Through extensive experiments on diverse and multilingual tasks, we demonstrate the superior performance of MetaSpS and MetaEagle compared to both specialized drafters and other state-of-the-art methods. Our work opens up new avenues for further research in speculative decoding, including exploring more sophisticated reward designs, incorporating switching costs, and addressing non-stationary environments.

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

# Appendices

## A OVERVIEW OF APPENDIX

This appendix provides supplementary material that expands on the main contents. Each section is designed to complement the research presented:

- **Appendix B**: Discusses the broader impact and further motivations of our work.
- **Appendix C**: Acknowledges the limitations of our current approach and outlines promising directions for future research.
- **Appendix D**: Provides a preliminary for speculative sampling (SpS).
- **Appendix E**: Provides a comprehensive review of related work, situating our contributions within the broader context of speculative decoding with LLMs and multi-armed bandit research.
- **Appendix F**: Details additional experimental setups, offering further insights into the performance, behavior of our proposed method, and additional experimental results including long-context experiments, out-of-domain experiments, and evaluations with perturbed prompts.
- **Appendix G**: Presents rigorous mathematical proofs for the theoretical guarantees established in the main paper.
- **Appendix H**: Explores extensions to the MetaSD framework, addressing practical considerations such as switching costs and non-stationary environments.
- **Appendix I**: Offers further discussion and analysis of the results presented in the main paper, potentially including additional insights, interpretations, or comparisons.

**Ethics statement**  This work primarily focuses on improving the efficiency of LLMs through algorithmic advancements and does not directly involve sensitive data or applications that could raise immediate ethical concerns.

**Reproducibility statement**  To facilitate reproducibility, we provide a comprehensive exposition of the materials and experimental configurations within this paper and its accompanying appendices. The organization is as follows:

- **Section 2** - This section presents the problem statement and pseudocode for the MetaSD framework.
- **Section 3** & **Section H.3** - This section provide detailed MAB algorithms for the MetaSD framework under various scenarios.
- **Section 4** - This section elaborates on the implementation specifics, including the pretrained models, datasets, and evaluation metrics.
- **Appendix F** - This section delves into additional details of the experimental settings.

## B BROADER IMPACT AND FURTHER MOTIVATION

### B.1 BROADER IMPACT

**Generalized speedup**  Our MetaSD framework for multi-drafter speculative decoding has the potential to enhance the robust speedup capabilities of LLMs. By dynamically selecting from a diverse pool of drafters, the system can better adapt to a wider range of tasks and input contexts, potentially leading to reduced latency on unseen or less frequently encountered scenarios. This increased generalization could benefit various applications, such as machine translation, summarization, and creative writing, where models are often required to handle diverse and unpredictable inputs.

**Efficiency**   The primary goal of our framework is to accelerate the inference process of LLMs. By leveraging speculative decoding with multiple drafters, we aim to achieve significant speedup gains compared to traditional single-drafter approaches. This improved efficiency could enable the deployment of large language models in resource-constrained environments or real-time applications where latency is critical. Faster inference could also facilitate broader accessibility to powerful language models, making them more practical for a wider range of users and use cases.

**Systematic impact**   Our work remains various potential societal impact. Faster and more efficient language models could lead to advancements in various domains, such as healthcare, education, and customer service, where natural language understanding and generation play crucial roles.

### B.2   FURTHER MOTIVATION

This subsection provides another line of research motivation in Section 2.1. MetaSD addresses the practical challenge of managing diverse and heterogeneous drafters often found in real-world systems (e.g., HuggingFace, Google Cloud, Azure, AWS, etc..). These drafters, pre-trained with varying objectives and frequently lacking detailed training documentation, pose significant obstacles to deployment frameworks that assume uniformity or rely on static selection strategies (e.g., rule-based strategies).

MetaSD provides a robust and adaptive mechanism for optimizing performance in environments characterized by task variability and drafter heterogeneity. By operating dynamically at the token level, it ensures task-specific efficacy without requiring retraining or fine-tuning of existing drafters. This flexibility allows MetaSD to excel in scenarios where traditional methods struggle, such as managing pre-trained drafters with black-box environment regarding the information for the use of drafters such as incomplete training histories or handling tasks with unpredictable distributions. Unlike frameworks that depend on rigid assumptions or predefined similarity metrics, MetaSD makes serving system particularly well-suited for organizations leveraging public repositories or heterogeneous resources.

## C   LIMITATION & FUTURE WORK

### C.1   LIMITATION

**Scalability**   It is important to acknowledge that the scalability of our approach may be challenged when dealing with an extremely large number of drafters. In such scenarios, the computational overhead associated with evaluating multiple drafters at each step could potentially outweigh the speedup benefits. To address this limitation, future work could explore strategies for pre-selecting a smaller subset of promising drafters based on initial query analysis or other heuristics, before applying the MetaSD framework. This would help to maintain the efficiency and scalability of our approach even in the presence of a vast pool of potential drafters.

**Diverse target LLMs**   While our framework is designed to be agnostic to the target LLM architecture, extensive empirical evaluation across a wider range of LLMs is needed. Future work will assess the generalizability of our approach across different LLM architectures and sizes.

**Batched inference**   Our current implementation primarily focuses on single-query scenarios. However, adapting the MetaSD framework to batched inference—where different tasks are mixed within a single batch—presents an opportunity for significant efficiency gains. Unlike static single-drafter-based SD, which can suffer from suboptimal performance when handling diverse tasks in a batch, MetaSD dynamically optimizes drafter selection at the instance level. This ensures consistently high throughput, even in high-throughput batched settings.

### C.2   FUTURE WORK

**Reward design and exploration-exploitation balance**   The choice of reward function and the exploration-exploitation tradeoff significantly impact the performance of MetaSD. Exploring alter-

native reward designs and adaptive exploration strategies could lead to further improvements in speedup and adaptability.

**Non-stationarity** While we briefly discuss handling non-stationarity in Appendix H, more sophisticated techniques could be investigated. This could involve incorporating change detection mechanisms or developing MAB algorithms specifically tailored to the non-stationary nature of language generation.

**Contextual bandits** Our current framework primarily relies on observed rewards for drafter selection. Incorporating additional contextual information, such as the query type, user history, or drafter metadata, could lead to more informed decisions. Integrating contextual bandit algorithms into the MetaSD framework is a promising direction for future research.

**Reinforcement learning (RL) formulation** The MetaSD framework could also be formulated as an RL problem, where the agent learns to select the optimal drafter based on the current state (input context and generated text) to maximize a long-term reward (e.g., overall speedup). Exploring RL-based approaches could potentially uncover novel strategies for adaptive drafter selection.

**MAB framework over different SD algorithms** Our current work focuses on applying the MAB framework to select among heterogeneous drafters sharing the same SD algorithm (e.g., SpS or EAGLE). While this approach demonstrates significant benefits, it is worth noting that the MAB framework could potentially be extended to encompass a more diverse set of SD algorithms (e.g., Sps, PLD, Lookahead, EAGLE, and others). This would involve designing a reward function and selection strategy that can effectively compare and choose between fundamentally different SD approaches, each with its own strengths and weaknesses. Exploring this broader application of the MAB framework in speculative decoding is an interesting direction for future research.

## D PRELIMINARY: SPECULATIVE SAMPLING

Speculative decoding accelerates LLM inference by employing a smaller draft model to predict future tokens, which are then verified by the target LLM. This parallel token generation can significantly reduce latency, especially when the draft model's predictions align well with the target LLM's output distribution.

Algorithm 3 outlines the speculative sampling procedure (Leviathan et al., 2023; Chen et al., 2023). Given an initial prompt sequence, the draft model generates $E$ potential future tokens. Concurrently, the target LLM computes the probabilities of these tokens, as well as the probability of its own prediction for each subsequent token position. A drafted token is accepted if its probability, according to the target LLM, exceeds a certain threshold. This threshold is determined by comparing the target LLM's probability for the drafted token to both the draft model's prediction and a random sample, ensuring only high-confidence drafts are accepted. If a drafted token is rejected, the target LLM samples a token from the residual distribution, which represents the difference between its own prediction and the draft model's. This process iterates until the desired sequence length is reached.

Speculative sampling allows the target LLM to process multiple tokens in parallel by drafting them in advance, reducing the overall generation time. When the draft model's predictions are accurate, a significant portion of the generated tokens are accepted, leading to substantial speedup. The verification step and residual sampling ensure that the final generated sequence remains consistent with the target LLM's distribution, preserving generation quality. Speculative sampling provides a foundation for our proposed framework, where we extend this approach to incorporate multiple drafters and dynamically select the optimal one using MAB algorithms.

## E RELATED WORK

### E.1 SPECULATIVE DECODING

Speculative decoding employs a draft-then-verify paradigm to enhance LLM inference speed. This approach tackles the latency bottleneck in autoregressive decoding, where extensive memory trans-

---

**Algorithm 3:** Speculative sampling (SpS)

---

INPUT : Target LLM $\mathcal{M}_p$, a small drafter $\mathcal{M}_q$, initial prompt sequence $x_1, \ldots, x_l$ and target sequence length $B$.

1: **while** $l < B$ **do**
2:    **for** $e \leftarrow 1, \ldots, E$ **do**
3:       $x_{l_e} \sim \mathcal{M}_q(x|x_1, \ldots, x_l, x_{l_1}, \ldots, x_{l_{e-1}})$
4:    **end for**
5:    In parallel, compute $E + 1$ sets of logits drafts $x_{l_1}, \ldots, x_{l_E}$ with the target LLM $\mathcal{M}_p$:
      $\mathcal{M}_p(x|x_1, \ldots, x_l), \mathcal{M}_p(x|x_1, \ldots, x_l, x_{l_1}), \ldots, \mathcal{M}_p(x|x_1, \ldots, x_l, x_{l_1}, \ldots, x_{l_E})$
6:    **for** $j \leftarrow 1, \ldots, E$ **do**
7:       Sample $r \sim U[0, 1]$ from a uniform distribution
8:       **if** $r < \min(1, \frac{\mathcal{M}_p(x|x_1, \ldots, x_{l+j-1})}{\mathcal{M}_q(x|x_1, \ldots, x_{l+j-1})})$ **then**
9:          Set $x_{l+j} \leftarrow x_{l_j}$ and $l \leftarrow l + 1$
10:      **else**
11:         Sample $x_{l+j} \sim (\mathcal{M}_p(x|x_1, \ldots, x_{l+j-1}) - \mathcal{M}_q(x|x_1, \ldots, x_{l+j-1}))_+$ and exit for loop.
12:      **end if**
13:    **end for**
14:    If all tokens $x_{l+1}, \ldots, x_{l+E}$ are accepted, sample extra token
      $x_{l+E+1} \sim \mathcal{M}_p(x|x_1, \ldots, x_l, x_{l+E})$ and set $l \leftarrow l + 1$
15: **end while**

---

fers for each token generation lead to underutilized compute resources (Patterson, 2004). Pioneering works by Leviathan et al. (2023); Chen et al. (2023) introduced speculative decoding and sampling, enabling lossless acceleration of diverse sampling methods. These methods leverage smaller models within the same model family (e.g., T5-small for T5-XXL) without additional training. Recent advancements have further refined speculative decoding. Models like Eagle (Li et al., 2024) and Medusa (Cai et al., 2024) integrate lightweight feedforward neural network heads into the LLM architecture, enabling early drafting of token sequences and improving throughput.

Despite their efficacy, these methods often rely on a single drafter or a fixed set, limiting adaptability to diverse tasks and input contexts. Yi et al. (2024) propose specialized drafters based on the self-distilled dataset training, but dynamically selecting among heterogeneous drafters remains an open challenge. Liu et al. (2023) suggest online training of specialized drafters, but their reliance on query-based classification and limited speedup gains highlight the need for a more comprehensive solution.

### E.2 BANDIT ALGORITHMS

**Multi-armed bandit** Multi-armed bandit (MAB) problem has been extensively studied for decades with various settings. For stochastic MAB setting, Lai & Robbins (1985) and Agrawal (1995) provided asymptotic optimal regret bounds that is logarithmic to the total round $T$ and Auer (2002); Audibert et al. (2007) and Honda & Takemura (2010) proved this result also holds when $T$ is finite. For another variant, EXP3 algorithm (Auer et al., 2002) proves the optimal regret bound in adversarial environment where reward distribution of each arm can change by adversary in every round.

**Budgeted bandit** The budgeted MAB problem address a bandit scenario where each arm pull yields both a reward and a cost drawn from individual distributions. Here, the goal is to maximize the cumulative reward until sum of the cost reaches the budget. Then, the optimal arm would be the one with the highest reward-to-cost ratio. $\epsilon$-First policies (Tran-Thanh et al., 2010) and KUBE (Tran-Thanh et al., 2012) assumed a non-stochastic fixed cost for each arm pull. Ding et al. (2013) provided UCB-BV algorithm where cost for each arm is assumed to be a bounded discrete random variable.

**Bandits with switching costs** In real-world scenarios, a cost may be incurred whenever switching arms. This is related to the MAB problem with switching costs. (Dekel et al., 2014; Gao et al., 2019; Rouyer et al., 2021; Esfandiari et al., 2021; Amir et al., 2022). For stochastic MAB, Gao et al. (2019) and Esfandiari et al. (2021) assume a fixed cost is incurred whenever switching arms. They

proved an instance-dependent regret bound $O(\log T)$ which does not depend on the unit switching cost value.

**Pure exploration**  Pure exploration or best arm identification (BAI) problems (Even-Dar et al., 2002; 2006; Audibert & Bubeck, 2010) aim to explore as much as possible throughout the round to obtain the best arm at the end of the round. This contrasts with the traditional MAB objective which is maximizing cumulative reward. Even-Dar et al. (2002); Mannor & Tsitsiklis (2004) and Even-Dar et al. (2006) investigated pure exploration in MAB under the PAC learning framework. BAI problems are primarily categorized into two settings. First, in the fixed budget setting (Audibert & Bubeck, 2010; Karnin et al., 2013; Carpentier & Locatelli, 2016), the goal is to minimize the chance of selecting sub-optimal arms within a fixed number of rounds. The other problem targets fixed confidence setting (Karnin et al., 2013; Jamieson et al., 2014; Garivier & Kaufmann, 2016; Chen et al., 2017) whose objective is to minimize number of rounds required to achieve a desired confidence level.

**Non-stationary bandit**  Non-stationary bandit problems assume that reward distribution of each arm changes over time. The goal in non-stationary bandit problems is to find a balance between exploration and exploitation while carefully managing past information to adapt to the dynamic environment. Among the earliest works, Gittins (1974) assumed that only the best arm changes over time. This assumption was later relaxed in Whittle (1988), where the authors allow the mean reward for each arm to change at every round. Slivkins & Upfal (2008) assumed reward distribution follows a Brownian motion and established a regret upper bound that grows linear in rounds. Another line of works quantifies the degree of non-stationarity in the bandit instance by assuming a fixed value of $L$ which represents a number of times reward distributions change. Auer et al. (2002) suggested EXP3.S algorithm and proved regret upper bound with given $L$ but slightly worse when $L$ is not given. Kocsis & Szepesvári (2006) suggested Discounted-UCB, where they obtain reward estimates with discounting factor over time. Garivier & Moulines (2011) introduced Sliding-window UCB, where they used fixed-size window to retain information of the rounds within the window for estimating mean reward. ADSWITCH in Auer et al. (2019) is proven to be nearly minimax optimal, achieving the state-of-the art regret bound without any prior knowledge of $L$.

### E.3  LARGE LANGUAGE MODELS AND BANDITS

Recently, several works have made connections between LLMs with bandits using the emergent abilities of LLMs. One side of works utilize LLM as an agent to solve decision making problems combining with bandit framework (Baheri & Alm, 2023; Felicioni et al., 2024; Xia et al., 2024a; Park et al., 2024). On the otherside, some of the works use bandit algorithms for improve the performance guarantee of LLMs with certain tasks such as for efficient prompt optimization (Shi et al., 2024) and online model selection (Xia et al., 2024c).

Most relevant to ours, several concurrent works investigate how bandit framework can be incorporated into SD. Liu et al. (2024) used Thomson sampling algorithm (which is one of the most popular bandit algorithm) to adaptively choose maximum candidate length $N_{max}$ combining with early-exit framework. Huang et al. (2024) assumed existence of multiple drafters and formulate SD as a contextual bandit problem. However, they rely on collecting offline samples for the policy learning which can be costly. Furthermore, their approach is regarded as a classification problem that the selected drafter is fixed in a single query. **To the best of our knowledge, our work is the first to use MAB framework within every speculation round and provide its theoretical guarantees.**

## F  EXPERIMENT DETAIL

### F.1  TRAINING SPECIALIZED DRAFTERS WITH SELF-DISTILLED DATA

Following the Yi et al. (2024), we use their training strategy consisting of two steps:

1. Pretraining drafters on a portion of C4 dataset (Raffel et al., 2019) and ShareGPT dataset (ShareGPT, 2023).
2. Finetuning the models with self distilled data having the target task with templates.

**Self-distilled data**   Following prior work (Kim & Rush, 2016; Zhou et al., 2023; Cai et al., 2024; Yi et al., 2024), we generate the training data for specialized drafters through self-distillation from the target LLM. To capture the full spectrum of its output variability, we generate multiple responses at various temperatures—{0.0, 0.3, 0.7, 1.0}. We utilize this self-distilled dataset for training both independent small drafter models and dependent Eagle drafters. For Eagle-specific training details, we adhere to the settings outlined in the original Eagle paper (Li et al., 2024).

### F.2   DRAFTER DETAILS

All independent drafters are based on a decoder-only Llama transformer model with 68M parameters. The model configuration includes 2 hidden layers, 768 hidden size, 12 attention heads, and a vocabulary size of 32,000. Other key settings are: silu activation function, 0.0 attention dropout, and no weight decay. The training recipe involves pretraining on a subset of the C4 and ShareGPT datasets, followed by fine-tuning on task-specific data generated through self-distillation from the target LLM. We employ 4 NVIDIA A100 GPUs with 80GB memory, utilizing techniques like FSDP (Fully Sharded Data Parallelism), gradient checkpointing, and lazy preprocessing to optimize training efficiency. Hyperparameters include a batch size of 8, 3 training epochs, a learning rate of 2e-5, and a cosine learning rate scheduler with a warmup ratio of 0.03. We maintain consistent architecture and training procedures across all white-box drafters, ensuring their heterogeneity stems solely from the diverse task-specific datasets they are fine-tuned on. For further specifics on Eagle drafter training, we refer readers to the original Eagle paper (Li et al., 2024).

### F.3   DATASETS

**Training dataset**   We utilize a diverse collection of datasets to train our specialized drafters, ensuring their proficiency across various tasks and languages:

- ShareGPT (ShareGPT, 2023): A dataset of approximately 58,000 conversations scraped. These conversations include both user prompts and responses from OpenAI's ChatGPT.
- WMT16 De→En (Bojar et al., 2016): A dataset for German-to-English machine translation, providing high-quality parallel text data.
- JparaCrawl-v3.0 (Morishita et al., 2022): A large-scale Japanese web corpus, enabling training of a drafter specialized in Japanese-to-English translation.
- WMT16 Ru→En (Bojar et al., 2016): A parallel corpus for Russian-to-English machine translation, similar to the WMT16 De→En dataset but focusing on the Russian language.
- WMT14 Fr→En (Bojar et al., 2014): A dataset for French-to-English machine translation, providing additional multilingual training data.
- WMT19 Zh→En (Barrault et al., 2019): A dataset for Chinese-to-English machine translation, further expanding the language coverage of our drafter pool.
- Code alpaca (Chaudhary, 2023): A dataset of code generation instructions and corresponding outputs, facilitating the training of a drafter specialized in code-related tasks.
- CNN/Daily mail (Hermann et al., 2015): A dataset for summarization, comprising news articles and their corresponding summaries.
- Natural question answering (Kwiatkowski et al., 2019): A large-scale question answering dataset based on real user queries and Wikipedia passages, aiding in training a drafter for question answering tasks.
- Meta math question answering (Yu et al., 2023): A dataset focusing on mathematical question answering, providing specialized training data for a math-oriented drafter.

**Evaluation dataset**

- Multilingual translation: Ja to En (Morishita et al., 2022), Ru to En, De to En (Bojar et al., 2016), Fr to En (Bojar et al., 2014), and Zh to En (Barrault et al., 2019).
- Code generation: Code tasks from the MT-Bench dataset (Zheng et al., 2024).
- Summarization: CNN/Daily summarization dataset (Hermann et al., 2015).

- Question answering: Natural Questions dataset (Kwiatkowski et al., 2019).
- Math reasoning: GSM8K mathematical reasoning dataset (Cobbe et al., 2021).

**Templates** We employ specific prompt templates during model evaluation to guide the behavior of the target LLM and drafters, ensuring consistency and clarity in task execution. These templates are carefully designed to elicit desired responses and provide relevant context for each task category. Before the data templates, system prompts of LLMs are positioned at the front to provide additional context or instructions.

- Multilingual translation: 'Translate this sentence from [source language] to English: [source sentence]'.
- Code generation: Its instruction depends on the query.
- Summarization: 'Summarize: [article text]'.

### F.4 MAB SETTINGS

In our experiments, we set the exploration strength $\beta$ for MetaSD-UCB to 0.01, balancing exploration and exploitation. For MetaSD-EXP3, we use a gamma value of 0.4 to control the degree of exploration. In the SH algorithm, we set the period to 1, ensuring frequent elimination of underperforming drafters.

### F.5 BASELINE

We conduct several SD methods, ensuring their open-source availability and robust performance. Each method embodies a distinct strategy for accelerating LLM inference:

- SpS (Chen et al., 2023): SpS employs a smaller LM from the same model series as the drafter. In the verification stage, if a token is rejected, SpS corrects it using residual probability to maintain generation quality.
- BPD, Medusa, and Eagle (Stern et al., 2018; Cai et al., 2024; Li et al., 2024): These methods enhance the target LLM by incorporating additional lightweight FFN heads. These heads draft potential token sequences based on the penultimate layer representations from the target LLM.
- PLD (Saxena, 2023): Implementing the ideas of (Yang et al., 2023), PLD selects text spans directly from the input to serve as drafts, aiming for relevant and accurate initial predictions.
- R-BPD (Rescored blockwise parallel decoding) and R-Medusa (Rescored Medusa) (Kim et al., 2024): This method enhances BPD by rescoring the drafts at test-time, aiming to increase the number of accepted tokens.

### F.6 REWARD DISTRIBUTION

Figure 5 and Table 2 present a statistical analysis of the BE and BD reward distributions, collected using autoregressive decoding with the same Japanese dataset and drafter configurations as in Table 1. Several key observations emerge:

- Lower variance: The BD reward exhibits lower variance compared to the BE reward across all drafters. This suggests that BD provides a more stable and consistent feedback signal, leading to faster convergence with less sample complexity.
- Improved discrimination: The difference in mean reward between the optimal drafter (Drafter 1; Ja-drafter) and the suboptimal drafters is more pronounced with the BD reward. This improved discrimination between drafters can facilitate quicker identification of the optimal drafter by the MAB algorithm.
- Reduced sparsity: A significant portion of the BE rewards are zero, particularly for the suboptimal drafters. This sparsity can hinder the learning process of the MAB algorithm. In contrast, the BD reward consistently provides non-zero feedback, enabling continuous learning and adaptation.

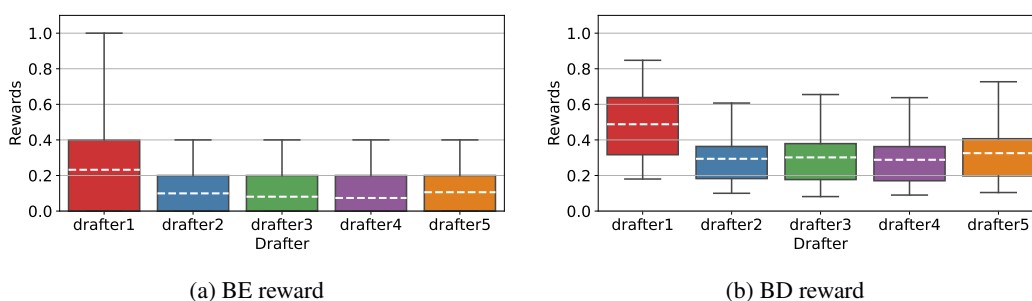

(a) BE reward          (b) BD reward

Figure 5: Comparison of rewards on the Ja→En dataset across different drafters in two scenarios: (a) BE and (b) BD. Box plots show the distribution of rewards, with whiskers extending to the 5th and 95th percentiles. Drafter specializations: 1: Ja →En, 2: Ru →En, 3: De →En, 4: Fr →En, 5: Zh →En.

Table 8: Speedup ratio on long-context De→En translation with the same settings in Table 5.

| Dataset | Drafter1 | Drafter2 | Drafter3 | Drafter4 | Drafter5 | UCB |
|---|---|---|---|---|---|---|
| Long De→En | 1.238 | 1.316 | 2.044 | 0.970 | 1.187 | 2.031 |

These observations collectively suggest that the BD reward offers several advantages over the BE reward in the context of MetaSD. Its lower variance, improved discrimination between drafters, and reduced sparsity contribute to a more informative and efficient learning signal for the MAB algorithm, potentially leading to faster convergence and better overall performance.

### F.7   LONG-CONTEXT DE →EN TRANSLATION

While our results in Table 3 and Table 5 have the relatively less effectiveness of MetaSpS on the WMT16 De→En translation task than other tasks, it is worth noting that this dataset primarily consists of relatively short sentences with an average length of fewer than 100 tokens. To assess the performance of our framework in a more challenging long-context scenario, we evaluate it on a new De→En translation dataset with an average context length of 500 tokens generated by GPT-4o. As shown in Table 8, MetaSpS-UCB achieves a speedup ratio of 2.031 on this long-context dataset, approaching the performance of the optimal drafter (Drafter3).

### F.8   EVALUATIONS ON OUT-OF-DOMAIN DATASETS

To evaluate the adaptability and performance of our MetaSD framework in out-of-domain settings, we conduct additional experiments using the Alpaca-Finance (Bhartia, 2023) and RAG datasets (Xia et al., 2024b). These datasets fall outside the domains of the specialized drafters used in our main experiments, providing a robust test of MetaSD's ability to generalize. The results in Table 9, measured using an NVIDIA A100 GPU, are presented below:

**Superior adaptability**   The results indicate that MetaSD consistently outperforms both OFA drafters and most of individual specialized drafters in out-of-domain scenarios. This highlights its ability to dynamically adapt to new tasks without relying on prior assumptions about domain similarity. The following provides the limitations of similarity-based selection:

- Computing similarity between sentence embeddings requires encoding the context to generate embeddings. For inputs exceeding 128 tokens, this process can significantly increase inference time. For example, with over 100 tokens, similarity computation becomes slower than MetaSD's dynamic drafter selection.
- High accuracy in selecting the correct drafter based on embeddings is challenging, leading to potential misclassifications. Errors in this step can result in suboptimal drafter performance. For example, as input lengths increase, the performance gap between static Math drafters and MetaSD-UCB narrows, reducing the benefits of static drafter selection.

Table 9: Performance of MetaSpS, MetaEagle, and baselines on out-of-domain datasets (measured on A100 GPU).

| Dataset | Drafter1 | Drafter2 | Drafter3 | Drafter4 | Drafter5 | OFA Drafter | MetaSpS-UCB | EAGLE1 | EAGLE2 | EAGLE3 | EAGLE4 | EAGLE5 | OFA Eagle | MetaEagle-UCB |
|---|---|---|---|---|---|---|---|---|---|---|---|---|---|---|
| RAG | 1.720 | 1.373 | 1.752 | 1.944 | 1.552 | 1.638 | **1.799** | 1.844 | 1.568 | 2.566 | 2.535 | 1.793 | 2.175 | **2.238** |
| Finance | 1.416 | 1.284 | 1.414 | 1.550 | 1.397 | 1.367 | **1.436** | 2.432 | 2.175 | 2.494 | 2.826 | 2.175 | 2.435 | **2.517** |

Table 10: Black-box performance with perturbed prompts (speedup relative to greedy decoding, measured on A100 GPU).

| Task | Drafter1 | Drafter2 | Drafter3 | Drafter4 | Drafter5 | OFA Drafter | MetaSpS - UCB |
|---|---|---|---|---|---|---|---|
| **Code** | 2.368 | 1.158 | 1.521 | 1.763 | 1.633 | 1.937 | **2.139** |
| **Translation** | 0.997 | 1.986 | 0.973 | 1.036 | 0.935 | 0.969 | **1.422** |
| **CNN** | 1.458 | 1.016 | 1.895 | 1.458 | 1.318 | 1.521 | **1.779** |
| **NQA** | 1.297 | 1.158 | 1.285 | 1.907 | 1.237 | 1.387 | **1.610** |
| **MathQA** | 1.482 | 1.184 | 1.357 | 1.470 | 2.346 | 1.895 | **2.149** |

**Intractability with heterogeneous drafters**   In practical scenarios, heterogeneous drafters often lack complete or uniform training descriptions. Under such conditions, similarity-based selection becomes infeasible. MetaSD's dynamic and adaptive approach offers a scalable alternative, ensuring robust performance even with limited information about drafter specialization.

## F.9    EVALUATIONS WITH PERTURBED PROMPTS

To better reflect real-world use cases, we conduct additional experiments using perturbed prompts. In this setting, the prompts for each query were slightly varied while remaining semantically equivalent to the original. These perturbations, generated using GPT-4o, ensured diverse yet natural variations. For example:

- In the translation task, the original prompt 'Translate German to English' used in the training was perturbed to 'Convert this text from German to English'.

- In the summarization task, the original prompt 'Summarize:' used in the training was perturbed to 'Provide a concise overview of the following text:'.

We find two key observations from the result. First, perturbed prompts introduce a performance drop across all methods, including OFA drafter/Eagle and individual specialized drafters. This degradation highlights that real-world variability in prompts can challenge any static drafter selection strategy, suggesting the need for more adaptive mechanisms. Second, despite the increased variability, MetaSD consistently outperforms all baselines, including OFA and individual drafters. The results demonstrate the strength of MetaSD's dynamic token-level selection mechanism, which adapts to the token distributions during inference rather than relying solely on the characteristics of the input prompt. The performance, measured as speedup relative to standard greedy decoding, is presented in Table 10 and Table 11.

## F.10    THROUGHPUT OVER EAGLE DRAFTERS

To evaluate the throughput efficiency of our proposed method, particularly in distributed system deployments where batch processing plays a critical role, we conduct experiments under the same settings described in the original Eagle paper Li et al. (2024). Using an RTX 3090 (24GB) with the Vicuna 7B model, we measured throughput across a diverse set of tasks. The results demonstrate that MetaEagle-UCB achieves superior throughput compared to the single OFA Eagle, with a speedup factor of **2.427** versus **2.235** for single drafters.

A key strength of our drafter management mechanism lies in its ability to maintain throughput efficiency comparable to single-drafter methods. This is facilitated by preloading drafter parameters into DRAM, thereby avoiding frequent memory transfers to VRAM during computation. As a result, both the number of memory movements and the overall memory bandwidth requirements remain consistent with those of single-drafter configurations, even in scenarios involving multiple drafters. Additionally, the computational structure of MetaSD is designed to scale effectively across batches.

Table 11: White-box performance with perturbed prompts (speedup relative to greedy decoding, measured on A100 GPU).

| Task | EAGLE1 | EAGLE2 | EAGLE3 | EAGLE4 | EAGLE5 | OFA Eagle | MetaEagle-UCB |
|------|--------|--------|--------|--------|--------|-----------|---------------|
| Code | 3.748 | 1.335 | 1.697 | 2.030 | 2.451 | **3.626** | 3.563 |
| Translation | 1.757 | 2.553 | 2.161 | 2.035 | 1.677 | 2.293 | **2.375** |
| CNN | 1.671 | 1.529 | 3.084 | 1.939 | 1.639 | 2.648 | **2.742** |
| NQA | 1.837 | 1.616 | 2.094 | 2.932 | 1.722 | 2.439 | **2.516** |
| MathQA | 2.511 | 1.664 | 2.207 | 2.913 | 3.844 | 3.136 | **3.366** |

Table 12: Performance comparison of MetaSD-UCB with different KV cache strategies (speedup relative to standard greedy decoding, measured on A100 GPU).

| Task | MetaEagle-UCB (Recomputing KV) | MetaEagle-UCB with StreamingLLM |
|------|--------------------------------|----------------------------------|
| Code | 3.724 | 3.624 |
| Trans | 2.318 | 2.352 |
| Sum | 3.057 | 2.986 |
| NQA | 2.641 | 2.654 |
| Math | 3.520 | 3.338 |

Performance gains observed in single-batch scenarios carry over seamlessly to multi-batch settings, ensuring throughput efficiency in real-world distributed environments.

### F.11 MetaEagle-UCB with Efficient KV Cache Strategies

In our framework, the KV cache is recalculated for the previous context whenever a drafter switch occurs. Despite this recalculation, the computational overhead is negligible, even for relatively long contexts. This efficiency arises from the minimal cost of prefilling the KV cache for a small drafter. For instance, in the Eagle drafter, only one layer of KV cache is computed for the unseen context, ensuring computational efficiency.

To further validate the framework's efficiency, we conducted additional experiments incorporating StreamingLLM techniques (Xiao et al., 2023). These techniques circumvent the need for full KV cache recalculation, offering an alternative method for reducing computational costs. The results, summarized in Table 12, demonstrate that StreamingLLM achieves comparable performance to the default approach of KV cache recalculation, highlighting the robustness of MetaSD.

These results confirm two key observations. First, the computational overhead introduced by full KV cache recalculation is minimal, as evidenced by MetaEagle-UCB maintaining high performance across tasks. This demonstrates that recalculating the KV cache is not a significant bottleneck. Second, Streaming Decode techniques provide an effective alternative, yielding similar overall performance with slight improvements observed in specific cases such as Translation and QA. These findings underscore the flexibility and efficiency of MetaSD in managing KV cache strategies.

## G Proofs

To begin, we provide the mathematical terms and notations in Table 13.

### G.1 Basic lemmas

First, we provide basic concentration inequalities which will be used to prove our theoretical results.

**Lemma 1** (Chernoff-Hoeffding bound). *Suppose there are $n$ random variables $X_1, X_2, \ldots, X_n$ whose value is bounded in $[0,1]$ and $\mathbb{E}[X_t|X_1, \ldots, X_{t-1}] = \mu$ for $2 \leq t \leq n$. Then, for $S_n = \sum_{i=1}^{n} X_i$ and $a \geq 0$, following inequalities holds:*

$$\mathbb{P}(S_n \geq n\mu + a) \leq e^{-2a^2/n}, \mathbb{P}(S_n \leq n\mu - a) \leq e^{-2a^2/n}.$$

Table 13: Mathematical terms and notations in our work.

| Notation | Descriptions |
|---|---|
| $K$ | Number of drafters |
| $[K]$ | For a given integer $K$, denotes the set $\{1, ..., K\}$ |
| $i$ | Drafter index $i \in [K]$ |
| $\alpha_i$ | True mean of acceptance rate when using drafter $i$ |
| $i^{\star}$ | Drafter index with the highest $\alpha_i$ |
| $t$ | Number of current round |
| $B$ | Total number of tokens to generate |
| $l(t)$ | Number of input tokens at round $t$ |
| $x^{1:l}$ | Token sequence of first $l$ tokens |
| $\mathcal{M}_q$ | Target model |
| $\mathcal{M}_{q_i}$ | The i-th drafter |
| $p^l$ | Probability distribution of target model output given token sequence $x^{1:l}$ |
| $q_i^l$ | Probability distribution of output of drafter $i$ given token sequence $x^{1:l}$ |
| $N_{acc}(i, t)$ | Number of accepted tokens using drafter $i$ in round $t$ |
| $N_{max}$ | Number of candidate tokens |
| $r$ | Arbitrary reward distribution with bounded support [0,1] |
| $r_{i,t}$ | General reward feedback using drafter $i$ in round $t$ |
| $r_{i,t}^{BE}$ | BE reward using drafter $i$ in round $t$ |
| $r_{i,t}^{BD}$ | BD reward using drafter $i$ in round $t$ |
| $n_i(t)$ | Number of selecting drafter $i$ until round $t$ |
| $a_t$ | Index of selected drafter in round $t$ |
| $\beta$ | The exploration strength hyperparamter in UCB |
| $\gamma$ | The exploration hyperparameter used in EXP3 |
| $\mu_i$ | Expectation of the reward distribution of drafter $i$ |
| $\pi$ | Bandit policy (algorithm) |
| $\tau(\pi, B)$ | Stopping time for the policy $\pi$ with given total number of tokens $B$ |
| $\lambda$ | Switching cost constant factor |
| $\Delta_i$ | Suboptimality gap for the arbitrary reward distribution $r$: $\mu_i^{\star} - \mu_i$ |
| $\Delta(\alpha_i)$ | Suboptimality gap for the BD reward: $\alpha_i^{\star} - \alpha_i$ |
| $\Delta_i^{BE}$ | Suboptimality gap for the BE reward |
| $R(r_i)$ | Feedback signal for reward distribution when using drafter $i$ (Theorem 1) |
| $d_{TV}(\cdot, \cdot)$ | The total variation distance between probability measures |
| $\mathbb{I}$ | Indicator function |
| $O(\cdot)$ | Big O notation |

**Lemma 2** (Bernstein inequality). *Suppose there are $n$ random variables $X_1, X_2, \ldots, X_n$ whose value is bounded in $[0,1]$ and $\sum_{t=1}^{n} \text{Var}[X_t | X_{t-1}, \ldots, X_1] = \sigma^2$. Then, for $S_n = \sum_{i=1}^{n} X_i$ and $t \geq 0$, following inequalities holds:*

$$\mathbb{P}(S_n \geq \mathbb{E}[S_n] + t) \leq \exp(-\frac{t^2}{\sigma^2 + t/2}).$$

## G.2 PROOF OF THEOREM 1

In order to prove the theorem, we first provide statistics for the BE and BD rewards by the following lemmas.

**BE reward statistics**   Here, we explicitly calculate expectation and variance of the BE reward in one round of speculative decoding. The result is presented in the following lemma.

**Lemma 3** (BE reward statistics)**.** *The expectation and variance of the number of accepted tokens is as follows:*

$$\mathbb{E}[r_{i,t}^{BE}] = \frac{\alpha_i - \alpha_i^{N_{max}+1}}{N_{max}(1 - \alpha_i)},$$

$$\mathrm{Var}[r_{i,t}^{BE}] = \frac{\alpha_i \left( 1 - (2N_{max}+1)\alpha_i^{N_{max}} + (2N_{max}+1)\alpha_i^{N_{max}+1} - \alpha_i^{2N_{max}+1} \right)}{(N_{max})^2(1 - \alpha_i)^2}. \tag{5}$$

**Proof of Lemma 3**   We first start with calculating the expectation and variance of $N_{acc}$ which can be obtained in a closed form. Suppose we conduct one round of speculative decoding for candidate token indices $l + j$ for $j = 1, \ldots, N_{max}$. Now, define $E_{l+j}^i$ as the event of $(l+j)$-th token generated by drafter $i$ is accepted in the verification stage. Also, define random variable $X_{l+j}^i$ to be 1 when $E_{l+j}^i$ occurs and 0 otherwise. With the stationary assumption, one can observe $X_{l+j}^i$ follows Bernoulli distribution with mean $\alpha_i$. Now, expectation can be obtained as:

$$\mathbb{E}[N_{acc}(i,t)] = \sum_{l=1}^{N_{max}} \mathbb{E}[X_{l+j}^i] = \sum_{l=1}^{N_{max}} \alpha_i^l = \frac{\alpha_i - \alpha_i^{N_{max}+1}}{1 - \alpha_i}. \tag{6}$$

To obtain variance, from $X_{L+l}^i \sim Ber(\alpha_i^l)$, following holds:

$$\mathrm{Var}(X_{L+l}^i) = (\alpha_i^l - \alpha_i^{2l})$$

Now, we can directly obtain a closed form of the variance by,

$$
\begin{aligned}
\mathrm{Var}(N_{acc}(i,t)) &= \mathrm{Var}(\sum_{l=1}^{N_{max}} X_{L+l}^i) \\
&= \sum_{l=1}^{N_{max}} \mathrm{Var}(X_{L+l}^i) + 2 \cdot \sum_{l<m} \mathrm{Cov}(X_{L+l}^i, X_{L+m}^i) \\
&= 2 \cdot \sum_{l=1}^{N_{max}} \sum_{m=l}^{N_{max}} \mathrm{Cov}(X_{L+l}^i, X_{L+m}^i) - \sum_{l=1}^{N_{max}} \mathrm{Var}(X_{L+l}^i) \\
&= 2 \cdot \sum_{l=1}^{N_{max}} \sum_{m=l}^{N_{max}} (\alpha_i^m - \alpha_i^{m+l}) - \sum_{l=1}^{N_{max}} (\alpha_i^l - \alpha_i^{2l}) \\
&= 2 \cdot \sum_{l=1}^{N_{max}} \sum_{m=l}^{N_{max}} \alpha_i^m - 2 \cdot \sum_{l=1}^{N_{max}} \sum_{m=l}^{N_{max}} \{ \alpha_i^{m+l} - \frac{\alpha_i(1-\alpha_i^{N_{max}})(1-\alpha_i^{N_{max}+1})}{1-\alpha_i^2} \} \\
&= 2 \cdot \sum_{l=1}^{N_{max}} l \cdot \alpha_i^l - 2 \cdot \sum_{l=1}^{N_{max}} \{ \alpha_i^l \left( \frac{\alpha_i^l - \alpha_i^{N_{max}+1}}{1-\alpha_i} \right) - \frac{\alpha_i(1-\alpha_i^{N_{max}})(1-\alpha_i^{N_{max}+1})}{1-\alpha_i^2} \} \\
&= 2 \cdot \sum_{l=1}^{N_{max}} l \cdot \alpha_i^l - 2 \cdot \frac{1}{1-\alpha_i} \sum_{l=1}^{N_{max}} \alpha_i^{2l} \\
&\quad + 2 \cdot \frac{\alpha_i^{N_{max}+1}}{1-\alpha_i} \sum_{l=1}^{N_{max}} \{ \alpha_i^l - \frac{\alpha_i(1-\alpha_i^{N_{max}})(1-\alpha_i^{N_{max}+1})}{1-\alpha_i^2} \} \\
&= \frac{2\alpha_i(N_{max} \cdot \alpha_i^{N_{max}+1} - (N_{max}+1)\alpha_i^{N_{max}} + 1)}{(1-\alpha_i)^2} - \frac{2\alpha_i^2(1-\alpha_i^{2N_{max}})}{(1-\alpha_i)(1-\alpha_i^2)} \\
&\quad + \frac{2\alpha_i^{N_{max}+2}(1-\alpha_i^{N_{max}})}{(1-\alpha_i)^2} - \frac{\alpha_i(1-\alpha_i^{N_{max}})(1-\alpha_i^{N_{max}+1})}{1-\alpha_i^2}.
\end{aligned}
\tag{7}
$$

The second equality comes from the basic property of variance, the fourth equality is from observing $\mathrm{Cov}(X_{L+l}^i, X_{L+m}^i) = \mathbb{E}[X_{L+l}^i X_{L+m}^i] - \mathbb{E}[X_{L+l}^i]\mathbb{E}[X_{L+m}^i] = \alpha_i^m - \alpha_i^{l+m}$. After rearranging the terms, we can obtain closed form of the variance as follows.

$$
\mathrm{Var}(N_{acc}(i,t)) = \frac{\alpha_i \left( 1 - (2N_{max}+1)\alpha_i^{N_{max}} + (2N_{max}+1)\alpha_i^{N_{max}+1} - \alpha_i^{2N_{max}+1} \right)}{(1-\alpha_i)^2}.
\tag{8}
$$

Since $r_{i,t}^{BE} = \frac{1}{N_{max}} N_{acc}(i,t)$ by definition, plugging this into eq. 6 and eq. 8 concludes the proof.

$\square$

**BD reward statistics**  Next, we obtain the expectation and variance of the BD reward by following lemma.

**Lemma 4.** *Following the relationships hold for $r_{i,t}^{BD}$ for all $i,t$:*

$$
\mathbb{E}[r_{i,t}^{BD}] = \alpha_i, \mathrm{Var}[r_{i,t}^{BD}] \le \frac{1}{4N_{max}}
\tag{9}
$$

**Proof of Lemma 4**  Under stationary assumption, any random variable which is bounded in $[0,1]$ has variance less than $\frac{1}{4}$. Since in eq. 1, $r_{i,t}^{BD}$ is constructed by empirical mean of $N_{max}$ numbers of samples under stationary assumption, following holds:

$$
\mathrm{Var}[r_{i,t}] = \mathrm{Var} \left[ \frac{1}{N_{max}} \sum_{j=0}^{N_{max}-1} (1 - d_{TV}(p^{l(t)+j}, q_i^{l(t)+j}) \right] \le \frac{1}{4N_{max}},
$$

and this concludes the proof. $\qquad\square$

Next, we formally define the bandit signal ratio as follows.

**Definition 3** (Feedback signal). *Under stationary environment, any reward design $r_i$ with $\mu_i = \mathbb{E}[r_i]$, $i^\star = \arg\max \mu_i$, and $\Delta_i = \mu_i^\star - \mu_i$, we define feedback signal for each suboptimal arm $i \neq i^\star$ as follows.*

$$R(r_i) := \frac{\max(\mathrm{Var}[r_i], \mathrm{Var}[r_{i^\star}])}{\Delta_i^2}$$

As we will see, $R$ become crucial factor that governs regret upper bound of our MetaSD-UCB algorithm. Specifically, the lower $R(r_i)$ guarantees smaller amount of regret by picking suboptimal arm $i$.

Then, we provide a formal version of [Theorem 1](#) which states the BD reward actually has lower feedback signal compared to the BE reward.

**Theorem 3** (Formal version of [Theorem 1](#)). *Denote $\Delta(\alpha_i) := \alpha_{i^\star} - \alpha_i$ for any suboptimal arm $i$ and $n := N_{max}$ for notational convenience. For any $n \in \mathcal{N}$, define functions $f_n, g_n, h_n$ on $(0,1)$ by $f_n(x) = \frac{x - x^{n+1}}{1-x}$, $g_n(x) = f_n'(x) = \sum_{s=1}^n s x^{s-1}$, and $h_n(x) = \sum_{s=1}^n s(x^{s-1} - x^{2n-s})$. Then following holds:*

$$R(r_i^{BD}) \leq \frac{1}{4(\Delta(\alpha_i))^2 N_{max}}. \tag{10}$$

*Also, following holds for any drafter configuration satisfying $h_n(\alpha_{i^\star}) \geq \frac{g_n(\alpha_{i^\star})^2}{4n\alpha_{i^\star}}$ and $\mathrm{Var}[r_i^{BE}] < \mathrm{Var}[r_{i^\star}^{BE}]$:*

$$R(r_i^{BD}) < R(r_i^{BE}). \tag{11}$$

*Proof.* Upper bound for the BD reward can be directly obtained from [Lemma 4](#). To prove [eq. 11](#), denote $N_{max} = n$ for notational convenience. Then, by directly applying [Lemma 3](#), it is observed that

$$
\begin{aligned}
R(r_i^{BE}) &= \frac{\max(\mathrm{Var}[r_i^{BE}], \mathrm{Var}[r_{i^\star}^{BE}])}{\Delta_i^2} \\
&= \frac{\alpha_{i^\star}(1 - (2n+1)\alpha_{i^\star}^n + (2n+1)\alpha_{i^\star}^{n+1} - \alpha_{i^\star}^{2n+1})}{(f_n(\alpha_i^\star) - f_n(\alpha_i))^2(1 - \alpha_{i^\star})^2} \\
&> \frac{\alpha_{i^\star}(1 - (2n+1)\alpha_{i^\star}^n + (2n+1)\alpha_j^{n+1} - \alpha_{i^\star}^{2n+1})}{(g_n(\alpha_{i^\star})\Delta(\alpha_i))^2(1 - \alpha_{i^\star})^2} \\
&= \frac{\alpha_{i^\star} h_n(\alpha_{i^\star})}{(g_n(\alpha_{i^\star})\Delta(\alpha_i))^2} \\
&\geq \frac{1}{4(\Delta(\alpha_i))^2 N_{max}},
\end{aligned}
\tag{12}
$$

where the first inequality is from $f_n$ is a convex function, the second equality comes from [Lemma 3](#), and the last line comes from the assumption.

$\qquad\square$

**Practical considerations** While [Theorem 3](#) provides a general scenario, the inequalities used in its derivation can be quite loose in certain cases. In practice, the BD reward often exhibits a significantly smaller feedback signal $R(r_i)$ than the BE reward. For example, consider the case where $N_{max} = 5$, which is the setting used in our main experiments. The condition $h_n(\alpha_{i^\star}) > \frac{g_n(\alpha_{i^\star})}{4n\alpha_{i^\star}}$ holds for $0.06 < \alpha_{i^\star} < 0.8$, which covers most of the practical range of $\alpha_{i^\star}$. This implies that, in many

realistic scenarios, the BD reward leads to a substantially tighter regret bound compared to the BE reward, further supporting its effectiveness in the MetaSD framework. Moreover, assumption of $\text{Var}[r_i^{BE}] < \text{Var}[r_{i^\star}^{BE}]$ covers most of the practical scenarios. As an example, if $n = 5$, $\text{Var}[r_i^{BE}]$ is monotonically increasing until $\alpha_i = 0.815$. Consequently, for any drafter set with $\alpha_{i^\star} < 0.815$, $\text{Var}[r_i^{BE}] < \text{Var}[r_{i^\star}^{BE}]$ holds for all suboptimal drafters.

**Relationship between expectations of two rewards.** Combining Lemma 3 and Lemma 4, one can show that the expectation of the BD reward is proportional to the BE reward.

**Lemma 5.** *Following relationship holds between the expectation of the BE reward and the expectation of the BD reward:*

$$\mathbb{E}[r_{i,t}^{BE}] = \frac{1 - \alpha_i^{N_{max}}}{N_{max}(1 - \alpha_i)} \mathbb{E}[r_{i,t}^{BD}]. \tag{13}$$

### G.3 STOPPING TIME REGRET

In this subsection, we provide the equivalence relation between two objectives, maximizing the reward and minimizing the stopping time. First, we define the regret of MetaSD in terms of the stopping time. Denote $\tau(\pi, B)$ as the stopping time for any policy $\pi$ with target sequence length $B$ and $\pi^\star$ as the optimal policy. In Definition 2, stopping time regret of policy $\pi$ with $B$ is defined as:

$$\text{REG}^s(\pi, B) = \mathbb{E}[\tau(\pi, B)] - \mathbb{E}[\tau(\pi^\star, B)].$$

Intuitively, minimizing $\text{REG}^s(\pi, B)$ should guarantee optimal speedup since minimizing $\tau(\pi, B)$ implies minimizing the number of total SD round. The following lemma proves that our reward design is well aligned with such objective.

**Lemma 6** (BE reward original regret). *For any policy $\pi$ with the target sequence length $B$, denote the original regret objective using the BE reward as $\text{REG}^{o,BE}(\pi, T) = \sum_{t=1}^T (\mathbb{E}[r_{i^\star}] - \mathbb{E}[r_{a_t}])$. Then, the following equation holds:*

$$\text{REG}^{o,BE}(\pi, T) = \frac{1}{N_{max}} \text{REG}^s(\pi, B)$$

*Consequently, minimizing the regret in terms of accepted tokens is equivalent to minimizing $\text{REG}^{(s)}(\pi, B)$.*

*Proof.* It is observed that

$$B = \sum_{t=1}^{\tau(B)} (N_{acc}(i, t) + 1) = \tau(B) + \sum_{t=1}^{\tau(B)} N_{acc}(i, t) = \tau(B) + N_{max} \sum_{t=1}^{\tau(B)} r_{a_t, t}.$$

Thus,

$$\tau(\pi, B) - \tau(\pi^\star, B) = N_{max} \sum_{t=1}^{\tau(B)} (r_{a_t^\star, t} - r_{a_t, t}), \tag{14}$$

where $a_t^\star$ is the action from the optimal policy $\pi^\star$ in round $t$. By taking the expectation on both sides, we get the result. $\square$

However, we can show that above result does not hold in every reward design.

**Lemma 7** (BD reward original regret). *For any policy $\pi$ with the fixed target sequence length $B$, denote the original regret objective using the BE reward as $\text{REG}^{o,BD}(\pi, T) = \sum_{t=1}^T (\mathbb{E}[r_{i^\star}] - \mathbb{E}[r_{a_t}])$. Then, there exists a bandit instance with the two different policies $\pi_1, \pi_2$ such that:*

$$\mathbb{E}[Reg^{o,BD}(\pi_1, B)] < \mathbb{E}[Reg^{o,BD}(\pi_2, B)],$$
$$\mathbb{E}[Reg^s(\pi_1, B)] > \mathbb{E}[Reg^s(\pi_2, B)].$$

*Proof.* Suppose we have three drafters with $\alpha_1 = 0.1, \alpha_2 = 0.5, \alpha_3 = 0.8$ with $N_{max} = 2$. Consider $\pi_1$ as the deterministic policy where it picks the drafter 1 for the first round and pick the drafter 3 rest of the rounds. Also, $\pi_2$ be the policy which picks drafter 2 for the first two rounds and drafter 3 for the rest of the rounds. For the original regret objective, $\pi_1$ has expected regret of $0.7$ while $\pi_2$ has expected regret $0.6$. However, it can be observed that the number of expected tokens until first two rounds is $(0.1 + 0.1^2) + (0.8 + 0.8^2) = 1.55$ for $\pi_1$ and $2(0.5 + 0.5^2) = 1.50$ for $\pi_2$. Since policy for the rest of the rounds are the same, we can conclude that the expected stopping time of policy $\pi_1$ is less then that of policy $\pi_2$. As a result, $\pi_2$ is better in terms of original regret objective and $\pi_1$ is better with stopping time regret objective. $\qquad\square$

### G.4 METASD-UCB WITH GENERAL REWARD

In this subsection, we provide a generic theorem which is stated as follows.

**Theorem 4** (Generic regret upper bound). *For any reward design $r$, Denote $\mu_i = \mathbb{E}[r_{i,t}]$, $\Delta_i = \mu_{i^\star} - \mu_i$, and $i^\star = \arg\max \alpha_i$. If $i^\star = \arg\max \mu_i$, then there exists a constant $C', C''' > 0$ such that following bound holds:*

$$\text{REG}(\pi, B) < \sum_{i \neq i^\star} \frac{8}{\Delta_i^2}(\ln B + \ln(\ln(\sum_{i \neq i^\star} \frac{1}{\Delta_i^2})) + C') + C'''. \tag{15}$$

Above theorem holds for any reward design as long as the drafter with the maximum expected reward $\mathbb{E}[r_{i,t}]$ also has the highest acceptance rate $\alpha_i$. Since both the BD and BE rewards satisfy this condition, Theorem 4 applies to both of the reward designs. The proof of Theorem 4 consists of two main parts. First, given total round, we can bound the expected number of selecting suboptimal arms using the same anlysis in Auer (2002). Next, we get the upper bound on expected stopping time of MetaSD-UCB algorithm.

**Bounding suboptimal selection** Given fixed stopping time, we can bound the expectation of number of selecting suboptimal arms as follows:

**Lemma 8** (Theorem 1 from Auer (2002)). *Let $n_i(t)$ be the number of pulling sub-optimal drafter ($i \neq i^\star$) by the MetaSD-UCB until round $t$. Also, denote $\Delta_i := \mu_{i^\star}^r - \mu_i^r$ be the sub-optimal gap. Then, following inequality holds for $\beta = 1$ :*

$$\mathbb{E}[n_i(\tau(B))|\tau(B)] \leq \frac{8\ln\tau(B)}{\Delta_i^2} + 1 + \frac{\pi^2}{3}. \tag{16}$$

**Proof of Lemma 8** For the analysis, we restate the proof in Auer (2002) for MetaSD-UCB algorithm with our notations. One can observe $n_i(\tau(B))$, the number of times drafter $i$ is chosen for the one round of speculative decoding until the end of generation, can be bounded as follows:

$$
\begin{aligned}
n_i(\tau(B)) &= 1 + \sum_{t=K+1}^{\tau(B)} \mathbb{I}[a_t = i] \\
&\leq l + \sum_{t=K+1}^{\tau(B)} \mathbb{I}[a_t = i, n_i(t-1) \geq l] \\
&\leq l + \sum_{t=K+1}^{\tau(B)} \mathbb{I}\left[\hat{\mu}_{i,t-1} + \sqrt{\frac{2\ln(t-1)}{n_i(t-1)}} \geq \hat{\mu}_{i^\star,t-1} + \sqrt{\frac{2\ln(t-1)}{n_{i^\star}(t-1)}}, n_i(t-1) \geq l\right] \\
&\leq l + \sum_{t=1}^{\tau(B)} \sum_{s=1}^{t-1} \sum_{n_i=l}^{t-1} \mathbb{I}\left[\hat{\mu}_{i,n_i} + \sqrt{\frac{2\ln(t-1)}{n_i}} \geq \hat{\mu}_{i^\star,s} + \sqrt{\frac{2\ln(t-1)}{s}}\right].
\end{aligned}
\tag{17}
$$

Here, $\mathbb{I}$ is an indicator function and $l$ is a positive integer. Now, one can see following holds:

$$\mathbb{P}\left(\hat{\mu}_{i,n_i} + \sqrt{\frac{2\ln t}{n_i}} \geq \hat{\mu}_{i^\star,s} + \sqrt{\frac{2\ln t}{s}}\right) \leq$$

$$\mathbb{P}\left(\hat{\mu}_{i^\star,s} \leq \mu_{i^\star} - \sqrt{\frac{2\ln t}{s}}\right) + \mathbb{P}\left(\hat{\mu}_{i.n_i} \geq \mu_i + \sqrt{\frac{2\ln t}{n_i}}\right) + \mathbb{P}\left(\mu_{i^\star} < \mu_i + 2 \cdot \sqrt{\frac{2\ln t}{n_i}}\right).$$

First term and the second term in the above equation is bounded by Lemma 1 as:

$$\mathbb{P}\left(\hat{\mu}_{i^\star,s} \leq \mu_{i^\star} - \sqrt{\frac{2\ln t}{s}}\right) \leq \exp(-4\ln t) = t^{-4},$$
$$\mathbb{P}\left(\hat{\mu}_{i.n_i} \geq \mu_i + \sqrt{\frac{2\ln t}{n_i}}\right) \leq \exp(-4\ln t) = t^{-4}. \tag{18}$$

By choosing $l = \lceil\frac{8\ln\tau(B)}{\Delta_i^2}\rceil$, one can see that the last term is 0 since,

$$2\cdot\sqrt{\frac{2\ln t}{n_i}} \leq 2\cdot\sqrt{\frac{2\ln t}{\left(\frac{8\ln\tau(B)}{\Delta_i^2}\right)}} \leq \Delta_i. \tag{19}$$

Finally, taking expectation of eq. 17 and put the above result, one can see that:

$$
\begin{aligned}
\mathbb{E}[n_i(\tau(B))|\tau(B)] &\leq \lceil\frac{8\ln\tau(B)}{\Delta_i^2}\rceil + 2\sum_{t=1}^{\tau(B)}\sum_{s=1}^{t-1}\sum_{n_i=l}^{t-1}2t^{-4} \\
&\leq \lceil\frac{8\ln\tau(B)}{\Delta_i^2}\rceil + 2\sum_{t=1}^{\infty}\sum_{s=1}^{t-1}\sum_{n_i=l}^{t-1}2t^{-4} \\
&\leq \frac{8\ln\tau(B)}{\Delta_i^2} + 1 + \frac{\pi^2}{3}.
\end{aligned} \tag{20}
$$

$\square$

**Bounding stopping time** The overall structure of the proof in bounding the stopping time is based on the proof of Lemma 2 in Ding et al. (2013) while we provide additional details that suits with our problem formulation. First, we obtain upper bound on stopping time by following lemma:

**Lemma 9.** *Following inequalities holds for some constants $C', C'' > 0$ :*

$$\mathbb{E}[\tau(\pi, B)] \leq \frac{B(1-\alpha_{i^\star})}{1-\alpha_{i^\star}^{N_{max}+1}} + \sum_{i\neq i^\star}\frac{8}{\Delta_i^2}\left(\ln B + \ln\left(\ln\left(\sum_{i\neq i^\star}\frac{1}{\Delta_i^2}\right)\right) + C'\right) + C''.$$

In order to prove Lemma 9, we first present two lemmas for bounding stopping time for a single armed bandit process i.e., we play only the single arm consecutively until the end of the round. Then, we provide how can we decouple stopping time of multi-armed bandit process of UCB policy.

**Lemma 10.** *Let $\tau(\pi^i, B)$ be a stopping time for the single armed bandit process $\pi^i$ which chooses only same drafter $i$ throughout the generation (i.e. $a_t = i$ for all t). Then the stopping time can be bounded as:*

$$\frac{B(1-\alpha_i)}{1-\alpha_i^{N_{max}+1}} - 1 < \mathbb{E}[\tau(\pi^i, B)] \leq \frac{(B+1)(1-\alpha_i)}{1-\alpha_i^{N_{max}+1}}. \tag{21}$$

*Proof.* One can see the expected number of generated tokens in each round is $\mu_i^c = \frac{1-\alpha_i^{N_{max}+1}}{1-\alpha_i}$ and the remaining number of tokens in the last round is contained in $\{1, 2, \cdots, N_{max}\}$. Now,

suppose eq. 21 holds for all $B < B_0$. Then one can observe:

$$\mathbb{E}[\tau(\pi^i, B_0)] = \mathbb{E}\left[\sum_{j=0}^{N_{max}} \left(\tau(\pi^i, B_0 - 1 - j) + 1\right) \mathbb{P}[r_i^{BE} = j]\right]$$

$$\leq \sum_{j=0}^{N_{max}} \frac{(B_0 - j)(1 - \alpha_i)}{1 - \alpha_i^{N_{max}+1}} \mathbb{P}[r_i^{BE} = j] + 1$$

$$\leq \sum_{j=0}^{N_{max}} \frac{(B_0 + 1)(1 - \alpha_i)}{1 - \alpha_i^{N_{max}+1}} \mathbb{P}[r_i^{BE} = j] - \frac{(1 - \alpha_i)}{1 - \alpha_i^{N_{max}+1}} \mathbb{E}[r_i^{BE} = j] + 1$$

$$= \sum_{j=0}^{N_{max}} \frac{(B_0 + 1)(1 - \alpha_i)}{1 - \alpha_i^{N_{max}+1}} \mathbb{P}[r_i^{BE} = j].$$

Since it is trivial to see that eq. 21 holds for $B = 1$, by mathematical induction, one can conclude the proof. The lower bound can be proved by the exactly same manner as in the upper bound.

$\square$

Now, we propose a lemma which provides an upper bound on expected stopping time.

**Lemma 11.** *For MetaSD-UCB algorithm $\pi$ with given token ~~budget~~ target sequence length $B$, expectation of stopping time $\tau(B)$ can be bounded as follows:*

$$\mathbb{E}[\tau(B)] \leq \mathbb{E}[\tau(\pi^{i^\star}, B)] + \sum_{i \neq i^\star} \mathbb{E}[n_i(\pi, B)], \tag{22}$$

*where, $n_i(\pi, B)$ is number of selecting drafter $i$ by policy $\pi$ during the generation.*

*Proof.* We first prove the upper bound (eq. 22). For policy $\pi$ with the ~~budget~~ target sequence length $B$, define a corresponding process $\pi^u$ which is defined by extending the process with the new stopping time, which is:

$$\tau^u(\pi^u, B) = min\{\tau > 0 \mid \sum_{t=1}^{\tau} (N_{acc}(a_t^u, t) + 1) \cdot \mathbb{I}[a_t^u = i^\star] \geq B\}.$$

where, $a_t^u = a_t$ for $t \leq \tau(B)$ and $a_t^u = i^\star$ for $\tau(B) < t \leq \tau^u(\pi^u, B)$. In other words, $\tau^u(\pi^u, B)$ is the time where total number of generated tokens by optimal drafter exceeds $B$. Then, one can see from the construction of $\pi^u$ and by observing that $\tau^u$ does not depend on the number of tokens generated by suboptimal drafters, ~~$\mathbb{E}[n_{i^\star}(\pi, B)] \leq \mathbb{E}[\tau^u(\pi^u, B)] = \mathbb{E}[\tau(\pi^{i^\star}, B)]$.~~

$$\mathbb{E}[n_{i^\star}(\pi, B)] \leq \mathbb{E}[n_{i^\star}(\pi^u, B)] = \mathbb{E}[\tau(\pi^{i^\star}, B)]. \tag{23}$$

$\square$

**Proof of Lemma 9**    To prove the upper bound, from Lemma 8 and Lemma 11, it is shown that

$$\mathbb{E}[\tau(B)] \leq \mathbb{E}[\tau(\pi^{i^\star}, B)] + \sum_{i \neq i^\star} \mathbb{E}[n_i(\pi, B)]$$

$$\leq \frac{(B+1)(1 - \alpha_{i^\star})}{1 - \alpha_{i^\star}^{N_{max}+1}} + \sum_{i \neq i^\star} \mathbb{E}[n_i(\pi, B)]$$

$$\leq \frac{(B+1)(1 - \alpha_{i^\star})}{1 - \alpha_{i^\star}^{N_{max}+1}} + \sum_{i \neq i^\star} \frac{8}{\Delta_i^2} \mathbb{E}[\ln \tau(B)] + (K-1)(1 + \frac{\pi^2}{3}), \tag{24}$$

$$\leq \frac{(B+1) \cdot (1 - \alpha_{i^\star})}{1 - \alpha_{i^\star}^{N_{max}+1}} + \frac{\alpha_{i^\star} - \alpha_{i^\star}^{N_{max}+1}}{1 - \alpha_{i^\star}^{N_{max}+1}} \sum_{i \neq i^\star} \frac{8}{\Delta_i^2} \ln \mathbb{E}[\tau(B)].$$

$$\leq \frac{(B+1)(1 - \alpha_{i^\star})}{1 - \alpha_{i^\star}^{N_{max}+1}} + \sum_{i \neq i^\star} \frac{8}{\Delta_i^2} \ln \mathbb{E}[\tau(B)] + (K-1)(1 + \frac{\pi^2}{3})$$

where the second inequality holds from Lemma 10, the third inequality holds by Lemma 8, and the last inequality holds from Jensen's inequality. Now, using $\ln(x) \leq \frac{x}{\epsilon} + \ln(\epsilon) - 1$ and taking $\epsilon = \sum_{i \neq i^\star} \frac{16}{\Delta_i^2}$, one can obtain:

$$\mathbb{E}[\tau(B)] \leq \frac{(2B+2) \cdot (1-\alpha_{i^\star})}{1-\alpha_{i^\star}^{N_{max}+1}} + 2\ln(\sum_{i \neq i^\star} \frac{16}{\Delta_i^2}) - 2 + (2K-2)(1+\frac{\pi^2}{3}).$$

If we again put the above equation into the eq. 24, one can obtain:

$$\mathbb{E}[\tau(B)] \leq \frac{(B+1)(1-\alpha_{i^\star})}{1-\alpha_{i^\star}^{N_{max}+1}} + \sum_{i \neq i^\star} \frac{8}{\Delta_i^2} \ln\left( \frac{(2B+2) \cdot (1-\alpha_{i^\star})}{1-\alpha_{i^\star}^{N_{max}+1}} + 2\ln(\sum_{i \neq i^\star} \frac{1}{\Delta_i^2}) + C_1 \right) + C_2$$

$$\leq \frac{B(1-\alpha_{i^\star})}{1-\alpha_{i^\star}^{N_{max}+1}} + \sum_{i \neq i^\star} \frac{8}{\Delta_i^2}(\ln B + \ln(\ln(\sum_{i \neq i^\star} \frac{1}{\Delta_i^2})) + C') + C'',$$

where $C_1, C_2, C', C'' > 0$ are constants that are independent of $B$ and $\Delta_i$.

$\square$

**Proof of Theorem 4**  The theorem is proved by observing:

$$\mathbb{E}[\tau(\pi, B)] - \mathbb{E}[\tau(\pi^\star, B)]) = (\mathbb{E}[\tau(\pi, B)] - \mathbb{E}[\tau(\pi^{i^\star}, B)])$$

$$\leq \frac{B(1-\alpha_{i^\star})}{1-\alpha_{i^\star}^{N_{max}+1}} + \sum_{i \neq i^\star} \frac{8}{\Delta_i^2}(\ln B + \ln(\ln(\sum_{i \neq i^\star} \frac{1}{\Delta_i^2})) + C') + C'' - \mathbb{E}[\tau(\pi^{i^\star}, B)])$$

$$< \frac{B(1-\alpha_{i^\star})}{1-\alpha_{i^\star}^{N_{max}+1}} + \sum_{i \neq i^\star} \frac{8}{\Delta_i^2}(\ln B + \ln(\ln(\sum_{i \neq i^\star} \frac{1}{\Delta_i^2})) + C') + C'' - \frac{B(1-\alpha_i)}{1-\alpha_i^{N_{max}+1}} - 1$$

$$< \sum_{i \neq i^\star} \frac{8}{\Delta_i^2}(\ln B + \ln(\ln(\sum_{i \neq i^\star} \frac{1}{\Delta_i^2})) + C') + C'''.$$

where $C'' > 0$ is an appropriate constant which doesn't depend on $B$. $C', C''' > 0$ are constants independent of $B$ and $\Delta_i$. Here, first equality comes from Lemma 6, first inequality is from Lemma 9, and second inequality holds by putting $i^\star$ to the lower bound of Lemma 10.

$\square$

Note that above analysis holds for every $\beta > 0$ in Algorithm 2. However, when the target sequence length $B$ is finite, constant terms in the regret bound becomes important which makes the performance of the algorithm dependent on $\beta$. We empirically found the optimal $\beta$ in our experiments. We provide further discussion on using different $\beta$ in Appendix Section G.8

G.5  PROOF OF THEOREM 2

**Concentration inequality**  Denote empirical mean of the BD and BE rewards as follows.

$$\mu_{i,t}^{BD} = \frac{1}{n_i(t)} \sum_{\tau=1}^{t} r_{i,\tau} \cdot \mathbb{I}[a_\tau = i], \mu_{i,t}^{BE} = \frac{1}{n_i(t)N_{max}} \sum_{\tau=1}^{t} N_{acc}(i,t) \cdot \mathbb{I}[a_\tau = i],$$

where $n_i(t)$ is number of times drafter $i$ is selected until round $t$ and $\mathbb{I}$ is indicator function.

Then, following inequalities can be derived for $\epsilon > 0$:

$$\mathbb{P}\left( \hat{\mu}_i^{BE} \geq \frac{\alpha_i - \alpha_i^{N_{max}+1}}{N_{max}(1-\alpha_i)} + \epsilon \right) \leq \exp\left( -\frac{n_i(t)\epsilon^2}{2Var[r_i^{BE}] + \epsilon} \right), \tag{25}$$

$$\mathbb{P}\left( \hat{\mu}_i^{BD} \geq \alpha_i + \epsilon \right) \leq \exp\left( -2(N_{max})n_i(t)\epsilon^2 \right). \tag{26}$$

eq. 25 comes from combining Bernstein's inequality (Lemma 2) with Lemma 3 and eq. 26 is from combining Hoeffding's inequality (Lemma 1) with Lemma 4.

**Bandit algorithm guarantee** Using concentration inequalities for both rewards, we provide how the bandit signal defined in eq. 2 directly related to our algorithm Algorithm 2. In the proof of Theorem 4, one can observe that bounding number of suboptimal arm selection (Lemma 8) directly related to the regret under the new regret object defined by stopping time (Definition 2). Leveraging above results, the regret upper bound for MetaSD-UCB algorithm with the BD and BE rewards can be proved.

**Proof of Theorem 2** For the BD reward, by putting $\beta = \frac{1}{\sqrt{N_{max}}}$ in the UCB algorithm and apply eq. 26, one can directly observe eq. 18 becomes:

$$\mathbb{P}\left(\hat{\mu}_{i^\star,s} \leq \mu_{i^\star} - \frac{1}{\sqrt{N_{max}}} \cdot \sqrt{\frac{2\ln t}{s}}\right) \leq \exp(-4\ln t) = t^{-4},$$

$$\mathbb{P}\left(\hat{\mu}_{i.n_i} \geq \mu_i + \frac{1}{\sqrt{N_{max}}} \cdot \sqrt{\frac{2\ln t}{n_i}}\right) \leq \exp(-4\ln t) = t^{-4}. \tag{27}$$

By choosing $l = \lceil \frac{8\ln \tau(B)}{(N_{max})\Delta(\alpha_i)^2}\rceil$, one can see for $n_i \geq l$ :

$$\frac{2}{\sqrt{N_{max}}} \cdot \sqrt{\frac{2\ln t}{n_i}} \leq \Delta_i.$$

Rest of the proof is same as in Theorem 4 and we can obtain:

$$\text{REG}(B) \leq \sum_{i \neq i^\star} \frac{8}{(N_{max})\Delta(\alpha_i)^2}(\ln B + \ln(\ln(\sum_{i \neq i^\star} \frac{1}{\Delta_i^2})) + C') + C,$$

for some constants $C > 0$ and this concludes the proof of Theorem 2.

$\square$

**BE reward regret** For MetaSD-UCB algorithm with BE reward, we can obtain regret upper bound by the following theorem.

**Theorem 5.** *Define $\Delta_i^{BE} := \mu_{i^\star}^{BE} - \mu_i^{BE}$ where $\mu_i^{BE} = \mathbb{E}[r_i^{BE}]$. If $\text{Var}[r_i^{BE}] < \text{Var}[r_{i^\star}^{BE}]$, we can obtain the following regret upper bound for the MetaSD-UCB algorithm using BE reward:*

$$\text{REG}(\pi^{BE}, B) \leq \sum_{i \neq i^\star}\left(\frac{(32\text{Var}[r_{i^\star}^{BE}] + 16)}{(\Delta_i^{BE})^2}\right)(\ln B + \ln(\ln(\sum_{i \neq i^\star} \frac{1}{\Delta_i^2})) + C') + C, \tag{28}$$

*where $C, C' > 0$ are constants independent of $B, \Delta_i^{BE}$.*

*Proof.* From eq. 25, one can similarly modify the original proof of the UCB (Auer, 2002).

Then, putting $\epsilon = \sqrt{(8\text{Var}[r_{i^\star}^{BE}] + 4)\ln t}$ into eq. 25 make eq. 18 becomes:

$$\mathbb{P}\left(\hat{\mu}_{i^\star,s} \leq \mu_{i^\star} - \sqrt{\frac{(8\text{Var}[r_{i^\star}^{BE}] + 4)\ln t}{s}}\right) \leq \exp(-4\ln t) = t^{-4},$$

$$\mathbb{P}\left(\hat{\mu}_{i.n_i} \geq \mu_i + \sqrt{\frac{(8\text{Var}[r_{i^\star}^{BE}] + 4)\ln t}{n_i}}\right) \leq \exp(-4\ln t) = t^{-4}. \tag{29}$$

By choosing $l = \lceil \frac{(32\text{Var}[r_{i^\star}^{BE}] + 16)\ln \tau(B)}{(\Delta_i^{BE})^2}\rceil$, one can see for $n_i \geq l$ :

$$2 \cdot \sqrt{\frac{(8\text{Var}[r_{i^\star}^{BE}] + 4)\ln t}{n_i}} \leq \Delta_i^{BE}.$$

Rest of the proof is similar as in Theorem 4.

$\square$

**Regret comparison** We restate the Collorary 1 formally as follows:

**Corollary 2.** *For any* $n \in \mathcal{N}$, *define functions* $f_n, g_n, h_n$ *on* $(0,1)$ *by* $f_n(x) = \frac{x - x^{n+1}}{1-x}$, $g_n(x) = f'_n(x) = \sum_{s=1}^{n} sx^{s-1}$, *and* $h_n(x) = \sum_{s=1}^{n} s(x^{s-1} - x^{2n-s})$. *If* $h_n(\alpha_{i^\star}) \geq \frac{g_n(\alpha_{i^\star})^2}{4n\alpha_{i^\star}}$ *and* $\mathrm{Var}[r_i^{BE}] < \mathrm{Var}[r_{i^\star}^{BE}]$, *then the regret of our algorithm* $\pi^{BE}$ *with the BE reward feedback is upper bounded by some function* $f(B)$, *where* $f(B) > \frac{8}{(N_{max})(\Delta(\alpha_i))^2} \ln B$.

*Proof.* One can observe:

$$\frac{(32\mathrm{Var}[r_i^{BE}] + 16)}{(\Delta_i^{BE})^2} \geq \frac{(32\mathrm{Var}[r_i^{BE}])}{(\Delta_i^{BE})^2} > \frac{16}{\Delta(\alpha_i)^2(N_{max})},$$

where first inequality comes from Theorem 3. Now, putting above result with Theorem 2 and Theorem 5, we get the result. □

Note that the better regret upper bound does not always guarantee the better performance since sometimes it is a proof artifact. Since we take quite loose inequalities during the proof of Theorem 5, we can improve the constant factors for BE reward. Still, even with assuming we can use Lemma 1 inequality in BE reward (which has better guarantee then Bernstein's inequality), the result of Collorary 1 still holds which shows the distinction between two reward designs in terms of regret as in Theorem 3.

### G.6 ASSUMPTION ON ACCEPTANCE RATE

**IID assumption** Here, we formally define the assumption on the acceptance rate which is used throughout our analysis.

**Assumption 1.** *Denote* $\alpha_{i,t}$ *as the acceptance rate for* $t$-*th token generated by* $i$-*th model. Then, for any instance of* $x^{1:B}$ *generated by the target model,* $\alpha_{i,t}$'s *are i.i.d. from a distribution* $\nu_i$ *with expectation* $\alpha_i$. *In other words, following holds for all drafter* $i \in [K]$.

$$\alpha_{i,t} = 1 - d_{TV}\left(p^t(\cdot|x^{1:t-1}), q_i^t(\cdot|x^{1:t-1})\right) \overset{i.i.d.}{\sim} \nu_i, \mathbb{E}[\alpha_{i,t}] = \alpha_i. \tag{30}$$

Above assumption shows that the acceptance rate for each token only depends on the drafter index $i$. We empirically verify the validity of the assumption by observing the TV distance between a target model and a drafter is well concentrated (F.4). Also, note that we make Assumption 1 for any temperature $T$ which include greedy decoding. Assumption 1 assumes i.i.d. of acceptance rate $\alpha_{i,t}$ in every instance and this might include the case where $\alpha_i$ can vary for every generation. However, this does not affect the analysis of Theorem 2 since our algorithm reset the bandit instance in every new generation.

**Comparison with (Leviathan et al., 2023; Yin et al., 2024)** In (Leviathan et al., 2023), authors assume fixed value of $\alpha_i$ where they show expected number of generated token in each round is a fixed value. Our assumption is more general than this and variance of the acceptance rate is critical factor to obtain a concentration bound as stated in Lemma 3 and Lemma 4 which is impossible when assuming fixed acceptance rate. (Yin et al., 2024) analyze the most general case where they provide the expected number of total rejected tokens as follows:

$$\mathbb{E}[N_{rej}] = \sum_{t=1}^{T} \mathbb{E}_{x_{1:t-1} \sim p^t}[d_{TV}(p^t(\cdot|x^{1:t-1}), q_i^t(\cdot|x^{1:t-1}))] \tag{31}$$

This is general than Assumption 1 where we assume previous context $x^{1:t-1}$ does not affect the TV distance between target model and a drafter. Relaxing the assumption and considering context-dependent reward distribution will be related to a contextual bandit problem (Li et al., 2010) while we leave investigating on this as an interesting future direction.

### G.7 Randomness of the target sequence length B

We can consider general scenarios where we take all possible instances generated by a target model when using temperature sampling with $T > 0$. In this scenario, we define the expected regret over the probability space induced by the target model. To do so, we first provide a formal definition of a target sequence length $B$.

**Definition 4** (Target sequence length B). *Target sequence length $B$ is a stopping time which is defined as follows:*

$$B = \min\{t \in \mathbb{N} : x^t = EOS\}, \tag{32}$$

*where $x^t \sim p^t(\cdot|x^{1:t-1})$ with $p^t$ being a probability distribution from the target model given context $x^{1:t-1}$ and EOS refers to the end of sentence token.*

According to Definition 4, target sequence length is a random variable (a stopping time). With this, one can observe following lemma holds:

**Lemma 12.** *For $b \in \mathbb{N}$,*

$$\mathbb{P}(B = b) = \mathbb{E}_{x^{1:b-1} \sim p}\left[\left(\prod_{t=1}^{b-1}(1 - p^t(EOS|x^{1:t-1}))\right) \cdot p^b(EOS|x^{1:b-1})\right] \tag{33}$$

*Where, $p^t(\cdot|x^{1:t-1})$ refers to the conditional probability distribution from a target model for $t$-th token generation when given context $x^{1:t-1}$. Moreover, expectation of a target sequence length becomes:*

$$\mathbb{E}[B] = \mathbb{E}_p(B) = \sum_{b=1}^{\infty} b \cdot \mathbb{P}(B = b). \tag{34}$$

*Here, $\mathbb{E}_p$ denotes the expectation taken over the probability distribution induced by the target model $p$.*

Then the general version of stopping time which includes every instance of given context can be analyzed with the following objective.

**Definition 5** (General version of stopping time regret).

$$\text{REG}(\pi, B) = \mathbb{E}_{p,\pi}\left[\tau(\pi, B)\right] - \mathbb{E}_{p,\pi^\star}\left[\tau(\pi^\star, B)\right], \tag{35}$$

*where, $\mathbb{E}_p$ denotes the expectation taken over from a probability space induced by the randomness of target model generation and $\mathbb{E}_\pi, \mathbb{E}_{\pi^\star}$ refers to the expectation taken over from the probability space generated by a bandit policy $\pi$ and the optimal policy $\pi^\star$ respectively.*

In order to analyze the general version of the stopping time regret which includes the randomness of $B$, we first take additional assumption on acceptance rates which is stated as follows.

**Assumption 2.** *Denote $\alpha_{i,t}$ as the acceptance rate for $t$-th token generated by $i$-th model. Then, for any instance $x^{1:B}$ generated by the target model, $\alpha_{i,t}$'s are i.i.d. from a distribution $\nu_i$ with expectation $\alpha_i$. In other words, following holds for all drafter $i \in [K]$.*

$$\alpha_{i,t} = 1 - d_{TV}\left(p^t(\cdot|x^{1:t-1}), q_i^t(\cdot|x^{1:t-1})\right) \overset{i.i.d.}{\sim} \nu_i, \mathbb{E}[\alpha_{i,t}] = \alpha_i. \tag{36}$$

*Moreover, $\alpha_i$ is independent of $B$ and its conditional expectation over the events with given $B$ is same for every $B$.*

The above assumption implies acceptance rate for each drafter is i.i.d. from a stationary distribution of a given instance and its mean value is independent of $B$. Now, with the generalized regret objective and Assumption 2, one can obtain regret upper bound in terms of expectation of total generated tokens.

**Theorem 6** (General version of the Theorem 2). *Under Assumption 2, following regret bound holds for Meta-UCB with general stopping time regret:*

$$\text{REG}(\pi, B) < \sum_{i \neq i^\star} \frac{8}{(N_{max})\Delta(\alpha_i)^2}\left(\ln\left(\mathbb{E}[B]\right) + \ln\left(\ln\left(\sum_{i \neq i^\star} \frac{1}{\Delta(\alpha_i)^2}\right)\right) + C'\right) + C. \tag{37}$$

*Here, $C, C' > 0$ are again constants that are independent from $B$ and $\Delta(\alpha_i)$.*

*Proof.* Since drafter selection from the policy $\pi$ is independent from B under Assumption 2, we can decouple eq. 38 as follows:

$$\text{REG}(\pi, B) = \mathbb{E}_B[\mathbb{E}_\pi[\tau(\pi, B)] - \mathbb{E}_{\pi^\star}[\tau(\pi^\star, B)]], \tag{38}$$

where first expectation is taken over with respect to a probability distribution of $B$ generated from $p$. Using Jensen's inequality and combining with Theorem 2, we get the result. $\square$

### G.8 FURTHER ANALYSIS ON HYPER-PARAMETER $\beta$

Although original UCB-1 algorithm in (Auer, 2002) is based on using fixed value of $\beta = 1$, following works (Audibert et al., 2009; Bubeck, 2010) show the regret can indeed be dependent on the exploration parameter $\beta$. We provide a general results which includes a hyperparameter $\beta$ in MetaSD-UCB algorithm. In the following, we borrow the analysis of (Bubeck, 2010) for the general version of Theorem 2 that includes $\beta$.

**Theorem 7** (Regret upper bound containing $\beta$). *For $\beta > 0.5$ and with Assumption 1, the regret upper bound in Theorem 2 can be generalized as follows:*

$$\text{REG}(\pi, B) < \sum_{i \neq i^\star} \frac{8\beta^2}{(N_{max})\Delta(\alpha_i)^2}(\ln B + \ln(\ln(\sum_{i \neq i^\star} \frac{1}{\Delta_i^2})) + C') + C. \tag{39}$$

*Proof.* The proof is based on modifying Lemma 8 to the equation (2.15) in (Bubeck, 2010) which is stated here for the completeness.

$$\mathbb{E}[n_i(\tau(B))|\tau(B)] \leq \frac{8\beta^2 \ln \tau(B)}{\Delta_i^2} + 1 + \frac{4}{\ln(2\beta^2 + \frac{1}{2})}\left(\frac{2\beta^2 + \frac{1}{2}}{2\beta^2 - \frac{1}{2}}\right)^2 \tag{40}$$

Rest of the procedure is same with Theorem 2. $\square$

Note that extra $\beta^2$ appears in the regret bound and supports and constant term can arbitrarily blow up when $\beta$ becomes close to the $\frac{1}{2}$ by the right term in eq. 40. We refer (Bubeck, 2010) for further details of the calculations.

## H EXTENDED SCENARIOS FOR THE METASD FRAMEWORK

Our MetaSD framework is universal as it can incorporate various bandit algorithms tailored for different scenarios. However, establishing optimality guarantees for existing algorithms in this framework requires careful analysis or one should look for the different algorithm designs. This is due to two key distinctions in our problem formulation: (i) stochastic stopping time, and (ii) a new regret objective defined in terms of this stopping time (Definition 2).

This section explores two distinct scenarios and introduces possible algorithms for each. First, we address a scenario when switching costs is not negligible anymore. In MetaSD framework, this happens when substantial computational or memory overhead is incurred when changing drafters. Second, we consider non-stationary environment where the characteristics of the context change within a one generation. Finally, we briefly discuss on other possible extensions of our framework.

### H.1 SWITCHING COSTS

**Switching costs for multiple drafters** In order to use multiple drafters in SD, one need to replace all missing key-value(KV) cache values for the model whenever switching one drafter to another. Reading and writing KV cache is one of the factor which can decrease the inference speed, and we define any decrease of inference speed by changing drafter as the switching cost. Formally, switching cost is defined as $\lambda(a_t, t) = \lambda(l(t) - l(\tau_i(t))) \cdot \mathbb{I}[a_{t-1} \neq a_t]$ where $l(t)$ is number of processed tokens by the target model in round $t$, $\tau_i(t)$ is the latest round where $i$-th drafter is selected before round $t$, $\mathbb{I}$ is an indicator function, and $\lambda$ is a constant. we first define the pseudo regret objective in the presence of switching costs.

---

**Algorithm 4:** Pure exploration-then-commit (PETC)

---

INPUT  Drafter pool $[K]$, initial prompt sequence $x^{1:l}$, target sequence length $B$, exploration
    rounds $B_0$.
1: **for** $l = 1, 2, ..., B_0$ **do**
2:     Run SH algorithm with budget $B_0$ (in Algorithm 5)
3: **end for**
4: $\hat{i}_\star$ be the survived index.
5: **while** $l < B$ **do**
6:     SD with a single drafter $\hat{i}_\star$.
7: **end while**

---

**Definition 6.** *With bandit policy $\pi$ and the given budget $B$, we define the regret as follows:*

$$\text{REG}_{switch}(\pi, B, \lambda) = \mathbb{E}[\tau(\pi, B)] - \mathbb{E}[\tau(\pi^\star, b)] + \sum_{t=2}^{\tau(B)} \lambda_t \mathbb{P}(a_{t-1} \neq a_t). \tag{41}$$

To minimize the above regret, observe $\lambda(\pi, B) = \lambda \sum_{t=1}^{\tau(B)} \lambda(a_t, t) = \lambda \sum_{i=1}^{K} B_i$, where $B_i$'s are total number of tokens generated by the $i$-th drafter after the final round. Intuitively, this implies that total cost decreases when employing elimination-type of algorithms (Audibert & Bubeck, 2010; Karnin et al., 2013), which successively eliminate sub-optimal drafters and exclude those drafters from future selection. Consequently, the total regret $\text{REG}_{switch}(B, \lambda)$ can be reduced from early elimination of poor-performed drafters. However, regret can still increase if the best drafter is mistakenly eliminated early on. Therefore, it is essential to strike a balance between elimination-based algorithms and standard MAB algorithms. For this, we design a new algorithm **Pure Exploration-Then-Commit** (PETC) in Algorithm 4 which effectively balances these two approaches.

PETC (Algorithm 4) divides the MetaSD into two phases. In the first phase $l < B_0$, the algorithm tries to eliminate sub-optimal drafters as quickly as possible. In the bandit literature, this is related to the pure exploration (or best arm identification) problem (Lattimore & Szepesvári, 2020) and we select using SH Algorithm 5 for our analysis. After the exploration period for estimating the best drafter, the algorithm exclusively selects this drafter for the remaining rounds.

Now, we provide how to find the optimal $B_0$ which by the following theorem:

**Theorem 8** (Regret upper bound on PETC). *By choosing $B_0 = c \cdot \ln B$ for some constant $c > 0$ and using Algorithm 5 for the pure exploration in the for the first phase in Algorithm 4, $\text{REG}_{switch}(\pi, B, \lambda) \leq O(\ln B)$ holds.*

*Proof.* First, we can decompose the regret as:

$$\text{REG}_{switch}(\pi, B, \lambda) = \sum_{t=1}^{\tau(B_0)} \text{REG}(\pi, t) + \sum_{t=\tau(B_0)+1}^{\tau(B)} \text{REG}(\pi, t) + S_T,$$

where $\text{REG}(\pi, t)$ denotes original regret objective eq. 3 for one round $t$ and $S_T$ denotes the total switching cost. First term can be bounded by the stopping time of selecting the worst drafter every round until $B_0$ which can be bounded by $\tau(B_0) = O(\ln B)$ according to Lemma 10. To bound the second term, we borrow Theorem 4.1 in Karnin et al. (2013), where they prove the probability of Sequential Halving algorithm to select the suboptimal arm after $B_0$ round can be bounded by $3 \log_2 K \cdot \exp(-\frac{B_0}{8H_2 \log_2 K})$, where $H_2 := \max_i \frac{i}{\Delta_i^2}$. Then we have

$$\sum_{t=\tau(B_0)+1}^{\tau(B)} \text{REG}(\pi, t) \leq \tau(\pi^{i_w}, B) \cdot 3 \log_2 K \cdot \exp(-\frac{B_0}{8H_2 \log_2 K}) = O(\ln B),$$

where $i_w$ denotes the worst drafter, $\tau(\pi^{i_w}, B)$ denotes the stopping time for generating $B$ tokens using only the worst drafter. The last term is bounded by $\lambda B_0 = O(\ln B)$ and this concludes the proof. $\qquad\square$

---

**Algorithm 5:** Sequential Halving (SH) (Karnin et al., 2013)

---

INPUT  Total budget $T$, drafter pool $[K]$
 1: **Initialize** $S_0 \leftarrow [K]$
 2: **for** $t = 0, 1, \ldots, \lfloor \log_2(K) \rfloor - 1$ **do**
 3:   Pull each drafter in $S_t$ for $n_t = \left\lfloor \frac{T}{|S_t| \lfloor \log_2(K) \rfloor} \right\rfloor$ additional times
 4:   $R_t(i) \leftarrow \sum_{j=1}^{n_t} r_{i,j}$ for $i \in S_t$
 5:   Let $\sigma_t$ be a bijection on $S_k$ such that $R_t(\sigma_t(1)) \leq R_t(\sigma_t(2)) \leq \ldots \leq R_t(\sigma_t(|S_t|))$
 6:   $S_{k+1} \leftarrow [i \in S_k | R_t(\sigma_t(i)) \leq R_t(\sigma_t(\lceil |S_k|/2 \rceil))]$
 7: **end for**
OUTPUT  Singleton element of $S_{\lfloor \log_2(K) \rfloor}$

---

Here, we can improve constant term in regret upper bound in Theorem 8 by controlling $c$ according to the switching cost $\lambda$ and given budget $B$ or we may use more advanced proof techniques in the best arm identification literature such as in Zhao et al. (2023). We leave these as a future work.

### H.2  NON-STATIONARY ENVIRONMENT

In real-world scenarios, the reward distribution for each drafter may evolve over time and past information becomes less relevant for decision-making. This phenomenon, referred to as non-stationarity, challenges traditional MAB algorithms that operate under the assumption of stationary reward distributions. In SD, non-stationarity can stem from various factors. For example, during a long-form text generation task, the optimal drafter may change as the topic or style of the text evolves. Consider the prompt: 'Please summarize and reason about the following article on climate change...'. Initially, a drafter specialized in summarization might be most effective. However, as the generation progresses towards the reasoning part, a drafter trained on logical reasoning tasks could become more suitable.

**Non-stationary MetaSD**   Standard analyses of non-stationary bandits (Auer et al., 2002; Kocsis & Szepesvári, 2006; Garivier & Kaufmann, 2016) often define $L$ to quantify the number of times the reward distributions change over $T$ rounds. Another line of work (Slivkins & Upfal, 2008; Besbes et al., 2014) quantifies the non-stationarity using $V$, the total variation of the means. In both cases, the regret (which is often called as dynamic regret) is defined as the cumulative expected difference between the rewards of the optimal arm and the selected arm at each round.

$$\text{REG}(\pi, B, L) = \sum_{t=1}^{\tau(B)} (\max_{i \in [K]} \mu_{i,t} - \mathbb{E}[\mu_{a_t,t}]) \tag{42}$$

where, as before, $B$ is the number of total tokens we have to generate, $\mu_{i,t}$ is the mean reward of choosing drafter $i$ in $t$-th round, and $\tau(B)$ is the total round. However, the regret upper bound on eq. 42 does not always guarantee the performance of the SD as we discussed in Section 3.1. Instead, we can use our original regret objective using stopping time Definition 2 without any modification.

Here, we introduce two types of algorithms within our MetaSD framework: Discounted-UCB (D-UCB) algorithm (Kocsis & Szepesvári, 2006) (Algorithm 6) and Sliding-window UCB (Garivier & Moulines, 2011) (Algorithm 7). Discounted UCB-SD estimates mean reward by computing the mean of discounted cumulative rewards as shown in the line 9 of Algorithm 6. By assigning less weight to the past observations, the algorithm finds a balance between accumulating knowledge and adapting to the changing environment. Similarly, sliding-window UCB utilizes a fixed-length window to calculate mean reward as demonstrated in the line 9-10 of Algorithm 7. By focusing only on recent information, it is also expected to achieve a balance with careful choose of the window size $\tau$. Garivier & Moulines (2011).

One interesting point is that in the non-stationary MetaSD problem, the definition of non-stationarity $L$ does not fit naturally into our problem. The reason behind this is that under non-stationary context generations, number of distribution changes happen at the token level, not the round level. This can disrupt existing regret analysis because a single round might involve multiple reward distribution changes (e.g., one round of speculative decoding could have two changing points). Whether above

---

**Algorithm 6:** Discounted UCB in MetaSD

---

INPUT Drafter pool $[K]$, initial prompt sequence $x^{1:l}$, target sequence length $B$, exploration
    strength $\beta$, decaying parameter $\gamma$.

1: $t \leftarrow 0$
    /* Phase 1: Meta-draft each drafter in $[K]$ once and do one round of speculative decoding. */
2: **for** $i \in [K]$ **do**
3:     Do one round of SD with drafter $i$ and obtain $N_{acc}(i, t), r_{i,t}$ (by eq. 1)
4:     $\hat{\mu}_{i,t}, n_i, l, t \leftarrow r_{i,t}, 1, l + N_{acc}(i, t) + 1, t + 1$
5: **end for**
    /* Phase 2: Meta-draft the draft following the UCB bandit until target sequence length $B$ */
6: **while** $l < B$ **do**
7:     $a_t \leftarrow \arg\max_{i \in [K]} \hat{\mu}_{i,t} + \beta\sqrt{\frac{2\ln t}{n_i}}$
8:     Do one round of SD with drafter $a_t$ and obtain $N_{acc}(a_t, t), r_{a_t, t}$ (by eq. 1)
9:     $\hat{\mu}_{a_t, t} = \frac{1}{n_{a_t}} \sum_{s=1}^{t} \gamma^{t-s} r_{a_s, s} \mathbb{I}[a_s = a_t]$
10:     $n_{a_t}, l, t \leftarrow n_{a_t} + 1, l + N_{acc}(a_t, t) + 1, t + 1$
11: **end while**

---

---

**Algorithm 7:** Sliding-window UCB in MetaSD

---

INPUT Drafter pool $[K]$, initial prompt sequence $x^{1:l}$, target sequence length $B$, exploration
    parameter $\beta$, window size $\tau$.

1: $t \leftarrow 0$
    /* Phase 1: Meta-draft each drafter in $[K]$ once and do one round of speculative decoding. */
2: **for** $i \in [K]$ **do**
3:     Do one round of SD with drafter $i$ and obtain $N_{acc}(i, t), r_{i,t}$ (by eq. 1)
4:     $\hat{\mu}_{i,t}, n_i, l, t \leftarrow r_{i,t}, 1, l + N_{acc}(i, t) + 1, t + 1$
5: **end for**
    /* Phase 2: Meta-draft the draft following the UCB bandit until target sequence length $B$ */
6: **while** $l < B$ **do**
7:     $a_t \leftarrow \arg\max_{i \in [K]} \hat{\mu}_{i,t} + \beta\sqrt{\frac{2\ln t}{n_i}}$
8:     Do one round of SD with drafter $a_t$ and obtain $N_{acc}(a_t, t), r_{a_t, t}$ (by eq. 1)
9:     $\hat{\mu}_{i,t} \leftarrow \frac{1}{n_i(t)} \sum_{s=t-\tau+1}^{t} r_{a_s, s} \mathbb{I}[a_s = i] \, \forall i \in [K]$
10:     $n_i(t) \leftarrow \sum_{s=t-\tau+1}^{t} \mathbb{I}[a_s = i] \, \forall i \in [K]$
11:     $l, t \leftarrow l + N_{acc}(a_t, t) + 1, t + 1$
12: **end while**

---

algorithms maintain optimal regret bounds in our regret definition in this non-stationary setting
presents an interesting direction for future theoretical analysis.

### H.3 OTHER POSSIBLE SCENARIOS

**Adversarial environment** EXP3 (Auer et al., 2002) is designed to handle adversarial changes
of reward distributions by continuously updating its estimates of the arm rewards and adjusting its
exploration strategy accordingly. It achieves this by maintaining a probability distribution over the
arms and exponentially weighting the rewards based on their recent performance. By incorporating
EXP3 into our framework (Algorithm 8), we can enable the system to adapt to evolving reward
distributions and dynamically select the optimal drafter even in adversarial environments. We utilize
this algorithm as a baseline in our experiments.

---

**Algorithm 8:** MetaSD-EXP3 (Auer et al., 2002)

---

INPUT Drafter pool $[K]$, initial prompt sequence $x^{1:l}$, target sequence length $B$, $\gamma \in (0, 1]$

1: $t \leftarrow 0$, $w_t(i) \leftarrow 1$ for $i = 1, \ldots, K$
2: **while** $l < B$ **do**
3:

$$p_t(i) = (1 - \gamma)\frac{w_t(i)}{\sum_{i=1}^{K} w_t(i)} + \frac{\gamma}{K} \quad i = 1, \ldots, K.$$

4:    Draw $a_t$ randomly according to the probabilities $p_t(1), \ldots, p_t(K)$.
5:    Do one round of SD with drafter $a_t$ and obtain $N_{acc}(a_t, t)$, $r_{a_t, t}$ (by eq. 1)
6:    **for** $j = 1, \ldots, K$ **do**
7:

$$\hat{r}_{j,t} = \begin{cases} r_{j,t}/p_t(j) & \text{if } j = a_t \\ 0 & \text{otherwise}, \end{cases}$$

$$w_{t+1}(j) = w_t(j) \exp\left(\frac{\gamma \cdot \hat{r}_{j,t}}{K}\right)$$

8:    **end for**
9:    $l, t \leftarrow l + N_{acc}(a_t, t) + 1, t + 1$
10: **end while**

---

# I   FURTHER DISCUSSION

## I.1   IS SCALING UP DRAFTER SIZE ALWAYS BETTER?

While increasing the drafter size might seem like a straightforward path to improved performance, it can be less efficient than our MetaSD approach, especially considering memory bandwidth constraints. Larger models demand more memory for storing weights and activations, increasing data movement between memory and processing units. This can become a bottleneck, particularly in high-performance computing where memory bandwidth is often a limiting factor. It is also discussed in Yi et al. (2024) in SD scenarios. Moreover, this phenomenon is well-illustrated by the roofline model, which highlights the trade-off between computational intensity and memory bandwidth (Cai et al., 2024). As model size increases, computational intensity might improve, but the memory bandwidth demands can quickly limit overall speedup.

In contrast, MetaSD utilizes multiple smaller drafters with lower individual memory requirements. By efficiently switching between these drafters, MetaSD can achieve comparable or superior performance to a single large drafter while mitigating the memory bandwidth bottleneck. This is because, despite having multiple drafters, MetaSD only utilizes one drafter for computation at any given time. Thus, the memory bandwidth requirement does not scale with the combined size of all drafters, but rather with the size of the individual drafter being used. Provided sufficient GPU DRAM, this approach does not have any bottleneck compared to the single drafter SD. Furthermore, MetaSD offers the flexibility to incorporate diverse drafters with specialized capabilities. This specialization can be more effective than simply increasing the size of a single general-purpose drafter, particularly for tasks demanding domain-specific knowledge.

## I.2   COMPUTATIONAL OVERHEAD ANALYSIS

**Training overhead**   While specialization may require additional training efforts compared to an OFA (One-size-Fits-All) drafter, we emphasize that our approach is designed to handle real-world scenarios where heterogeneous drafters already exist in public repositories. MetaSD focuses on optimizing the utilization of such heterogeneous drafters, dynamically selecting the most suitable drafter during inference. This shifts the problem from retraining models to developing an effective strategy for utilizing pre-existing resources. Therefore, while training specialized drafters may involve additional costs in certain cases, the broader applicability and versatility of MetaSD provide substantial practical value. Additionally, the cost of training drafters is a general challenge shared across the speculative decoding research domain, not limited to our work.

**Inference memory-bandwidth efficiency** The inference memory-bandwidth efficiency of MetaSD remains comparable to single-drafter methods. Although MetaSD employs multiple drafters, the additional memory requirements are minimal. Specifically, MetaSD increases DRAM usage by only 2 GB (from 17 GB to 19 GB), as the drafters' weights are preloaded into DRAM. However, this does not affect VRAM bandwidth, as only the active drafter interacts with VRAM during inference. As a result, the VRAM bandwidth demands remain identical to those of single-drafter methods. This efficient memory management ensures that MetaSD maintains competitive performance without introducing significant overhead.

By ensuring that only the active drafter interacts with the VRAM, MetaSD maintains parity with single-drafter approaches in terms of VRAM bandwidth demands.

**Serving complexity** Using multiple drafters in MetaSD does not inherently increase serving complexity. Modern distributed systems already employ model parallelism techniques to allocate workloads across multiple GPUs effectively. In MetaSD, drafters are evenly distributed across GPUs, with each GPU independently handling its assigned drafter without added coordination costs. This design ensures the following:

- Load balancing: Drafters are distributed across GPUs based on their assigned tasks, maintaining equivalent complexity to single-drafter systems.
- Minimal communication overhead: MetaSD requires no additional inter-GPU communication beyond standard model parallelism setups.

**Justification of overhead** The modest increase in DRAM memory usage (+2 GB) and marginal training cost for specialized drafters is justified by the significant performance gains achieved through adaptive optimization. MetaSD dynamically selects the most suitable drafter for each task, consistently outperforming single-drafter methods across diverse scenarios, as highlighted in our experimental results. Furthermore, MetaSD addresses an important real-world challenge: effectively utilizing publicly available, pre-trained heterogeneous drafters. By providing a generalizable strategy for optimizing these resources, MetaSD adds practical value beyond specialized retraining, supporting diverse and evolving task requirements.

