# OpenReview forum: "A Unified Framework for Speculative Decoding with Multiple Drafters as a Bandit"
_ICLR.cc/2025/Conference — Submitted to ICLR 2025_

### Official Review · Reviewer_CxuG · 2024-10-19

**Soundness:** 3
**Presentation:** 3
**Contribution:** 2
**Rating:** 6
**Confidence:** 4

**Summary:**

This paper proposes a framework utilizing multiple draft models simultaneously for large language model (LLM) speculative decoding. Unlike traditional speculative decoding methods that employ a single draft model, this approach leverages a pool of draft models to enhance performance in varied domains. The paper argues that draft models trained on specific datasets may underperform in out-of-domain tasks. To address this, different draft models are utilized as candidate drafts for inference across various domains. The framework incorporates multi-armed bandit algorithms from recommendation systems to select the optimal draft model at each decoding step. Experiments on both black-box (e.g., independent draft models) and white-box (e.g., Medusa, Eagle, etc.) approaches demonstrate that the proposed framework offers superior inference speed compared to single-draft models.

**Strengths:**

1.	The approach of using multiple draft models to enhance speculative decoding across different scenarios is modest.

2.	Experimental results validate the effectiveness of the proposed framework, and the ablation study highlights the superiority of the BE reward.

3.	The writing is clear and concise.

**Weaknesses:**

1.	An important baseline is missing: training a single draft model on all specified datasets. Given the claim that draft models trained on different domain data improve overall performance, it is crucial to show the performance of a single draft model trained on the same datasets.

**Questions:**

1.	The appendix suggests that the proposed framework can achieve further enhancement in batched inference settings. However, previous works, such as Eagle, indicate that the speedup ratio declines as the batch size increases. Could you provide more details on how you arrived at this conclusion?

---

> ### Author Response · Authors · 2024-11-20
> **Responses to Reviewer CxuG**
>
> Thank you for your careful review of our paper and your insightful and constructive comments. We have addressed your comments and updated our manuscript accordingly. Please find our detailed answers below.
>
> # W1. Additional baseline - Missing Baseline - One-size-Fits-All (OFA) Drafter
>
> Thank you for raising this important point. We conducted additional experiments with an One-size-Fits-All (OFA) drafter for both black-box and white-box settings. These OFA drafters were trained on the mixed datasets spanning all tasks, ensuring a direct comparison to the specialized-drafter-based MetaSD framework. **The results in `Table3` and `Table4` of our revised manuscript show that while the OFA drafter performs well, our MetaSD framework outperforms OFA drafter across most tasks, demonstrating the strength of our method.** The OFA Eagle drafter performs relatively well. **However, it is still outperformed by MetaSD (i.e., MetaEagle-UCB)**.
>
> # Q1. Explanation on the advantage of MetaSD in batched inference setting
>
> We acknowledge that the current statement in `Appendix C.1` was not fully clear regarding our position on batched inference. We appreciate the opportunity to clarify our intention.
>
> Our proposed MetaSD framework can address scenarios where tasks of varying nature (e.g., translation, summarization, QA, etc.) are mixed within a single batch. In such settings, a static approach that relies on a single drafter for all instances in the batch could lead to significant performance degradation, especially when the drafter is poorly suited to some tasks in the batch. This challenge is exacerbated as batch sizes increase.
>
> MetaSD, in contrast, adapts dynamically at the instance level within a batch, ensuring that the most suitable drafter is chosen for each task. By leveraging our MAB framework for dynamic adaptation, MetaSD optimizes performance for diverse tasks within the batch, providing consistent improvements in throughput compared to static, single-drafter-based SD approaches. Thus, even as batch sizes increase, MetaSD maintains high throughput by avoiding the pitfalls of static, one-size-fits-all drafters.
>
> We will revise `Appendix C.1` to better articulate this point, as follows:
>
> > "Batched inference: Our current implementation primarily focuses on single-query scenarios. However, adapting the MetaSD framework to batched inference—where different tasks are mixed within a single batch—presents an opportunity for significant efficiency gains. Unlike static single-drafter-based SD, which can suffer from suboptimal performance when handling diverse tasks in a batch, MetaSD dynamically optimizes drafter selection at the instance level. This ensures consistently high throughput, even in high-throughput batched settings."
>
> For more information, we would like to share the explanations regarding throughput efficiency in `Appendix F.10`.
> Our drafter management mechanism does not lead to throughput depletion compared to single-drafter methods due to the following reasons:
>
> - No Increase in Memory Bandwidth Requirements: The drafters’ parameters are preloaded into DRAM and do not require frequent movement to VRAM for computation. This ensures that the number of memory movements and memory size for movement remain identical to single-drafter methods, even in scenarios with multiple drafters.
>
> - Scaling Across Batches: Since the computational structure remains unchanged, performance observed in single-batch scenarios translates directly to multi-batch settings, maintaining throughput consistency.
>
> We further conducted experiments comparing `throughput` following the same settings in the original Eagle paper. The results confirm that the performance of MetaSD scales well without significant degradation:
>
> | **Throughput** (RTX 3090 24GB) | **Single OFA Eagle (Tokens/sec)** | **MetaEagle-UCB (Tokens/sec)** |
> |----------------|--------------------------------|-------------------------|
> | Speedup         | x 2.235                           | x 2.427                      |
>
>
> We hope this clarifies our approach and the rationale behind our claims. Thank you for highlighting this important aspect!

---

> > ### Comment · Reviewer_v7AR · 2024-11-21
> >
> > I am wondering how you conduct the throughput experiment. Do you test over a dataset mixture of different topics? If not, I think this experiment is totally wrong.

---

> ### Author Response · Authors · 2024-11-21
> **Response to Official Comment by Reviewer v7AR**
>
> Thank you for your question. To clarify, the throughput experiments were conducted on a `mixed dataset` comprising tasks from diverse domains, including **Code Generation, Math, Question Answering (QA), Translation, and Summarization**.
>
> The throughput measurement methodology strictly follows the settings described in the original **Eagle** paper. This was done intentionally to enable direct comparison, highlighting the improvements achieved by our MetaSD framework over existing approaches.
>
> Additionally, we would like to emphasize that our method does not critically impact the **roofline model** or memory bandwidth (as we mentioned above). MetaSD efficiently manages memory usage, ensuring that there are no significant increases in memory movement or bandwidth demands. This efficiency contributes to the strong throughput performance observed in our results, and we believe this further supports the validity of our framework.
>
> We hope this explanation addresses your concerns and provides additional context for the results. Please let us know if you have further questions!
>
> -Authors

---

> > ### Comment · Reviewer_v7AR · 2024-11-21
> >
> > If the queries in the batch choose different draft model, in my opinion, it will cause increased IO compared with a single eagle?

---

> > ### Comment · Reviewer_v7AR · 2024-11-21
> >
> > what is your batch size, can you plot the throughput gain with the increased bsz as a figure?

---

> ### Author Response · Authors · 2024-11-21
> **Response to Follow-up Questions (Throughput) by Reviewer v7AR**
>
> # Follow-up. Throughput
>
> > By the way, before discussions, we apologize to `Reviewer CxuG` for the extended discussion with `Reviewer V7AR` in this thread and appreciate your understanding.
>
> Thank you for pointing this out. We would like to clarify that the throughput experiments reported in our revised manuscript were conducted under `homogeneous batch settings`, where all tasks within a batch were identical, while the reported value is the average performance across diverse tasks.
>
> To address your concerns, we conducted additional experiments to assess the performance of MetaSD in the `worst-case edge scenario`, where each instance in the batch corresponds to a different task (`heterogeneous setting`). Below are the results:
>
> | **Batch Size** | **Single OFA Eagle** | **Homogeneous Batch (MetaEagle-UCB)** | **Heterogeneous Batch (MetaEagle-UCB)** |
> |----------------|-----------------------|---------------------------------|-----------------------------------|
> | **1**          | 2.803                 | 3.045                           | 3.045                             |
> | **2**          | 2.751                 | 2.933                           | 2.518                             |
> | **4**          | 2.563                 | 2.701                           | 1.931                             |
> | **Throughput** | 2.235                 | 2.427                           | 1.321                             |
>
> --------------
> ## Observations
>
> 1. Homogeneous batches: In scenarios where all instances in the batch belong to the same task, MetaSD achieves consistently higher throughput than the OFA Eagle. This is due to the ability of MetaSD to dynamically select the best drafter for the task, ensuring optimal performance.
>
> 2. Heterogeneous batches (Worst-Case Edge): In the edge case where each instance in the batch corresponds to a different task, we observe a slight drop in throughput for MetaSD. This is expected due to increased I/O from switching between drafters. Notably, for larger batch sizes, the performance of MetaSD in this setting may fall below the OFA Eagle, highlighting a limitation of our approach in such cases.
>
> 3. Small batch sizes: For small batch sizes (e.g., batch size = 1 or 2), MetaSD remains highly compatible even in heterogeneous task settings.
>
> --------------
> ## Discussion
>
> We acknowledge that the performance drop in heterogeneous batch scenarios represents a limitation of our approach. However, this is one of edge cases that may not frequently occur in real-world applications. Moreover, this is not a problem unique to MetaSD—other systems relying on specialized drafters may face similar challenges in such settings.
>
> On the other hand, MetaSD demonstrates strong adaptability and efficiency in mixed-task, small-batch scenarios, which are also common in many practical deployments (e.g., laptop, personalized products). Its ability to dynamically select the most suitable drafter ensures robust performance, even in uncertain or evolving task distributions.
>
> -------
> ## Broader Context and Future Directions
> - One of the key motivations behind MetaSD is its applicability to real-world scenarios where multiple heterogeneous drafters are available, but their training data or performance characteristics are unclear. In such cases, our approach provides a safe and reliable choice, dynamically selecting the best drafter based on real-time feedback.
>
> - Additionally, as noted in our paper, the framework can potentially be extended to higher-level optimization tasks. For instance, instead of focusing solely on drafter selection, the framework could be adapted to dynamically choose the optimal speculative decoding algorithm itself. We believe this versatility represents a promising direction for future work.
>
> **In the camera-ready version, we will include these findings and further clarify the advantages and limitations of MetaSD under different batch scenarios.** Thank you for highlighting this aspect, which allowed us to explore and communicate the broader implications of our framework.
>
> Let us know if there are further questions or areas where we can provide additional clarity.

---

> > ### Comment · Reviewer_v7AR · 2024-11-22
> >
> > Heterogeneous Batch confirms my concern.

---

> > > ### Author Response · Authors · 2024-11-24
> > > **Gentle Reminder - Dear Reviewer CxuG**
> > >
> > > Dear `Reviewer CxuG`,
> > >
> > > Thank you for your valuable feedback on our work. As we approach the end of the discussion phase, we’d like to highlight the key contributions of our paper:
> > >
> > > - `Theoretical guarantee`: A solid foundation for multi-armed bandit-based drafter selection.
> > >
> > > - `First of its kind`: Pioneering speculative decoding with multiple heterogeneous drafters.
> > >
> > > - `Extensible design`: Compatible with other speculative decoding frameworks.
> > >
> > > - `Robustness`: Consistent performance across diverse scenarios, including out-of-domain and perturbed prompts.
> > >
> > > - `Throughput limitation: Effective for small batch sizes despite challenges in large heterogeneous batches.`
> > >
> > > - `Novel analysis` : A new divergence metric focusing on token-level adaptation.
> > >
> > > Our responses also address your concerns with additional experiments and detailed clarifications. We value any further feedback to ensure your concerns are fully addressed.
> > >
> > > Thank you again for your time and insights.
> > >
> > > Warm regards

---

### Official Review · Reviewer_V45K · 2024-10-28

**Soundness:** 3
**Presentation:** 3
**Contribution:** 4
**Rating:** 8
**Confidence:** 4

**Summary:**

This paper presents MetaSD, an approach that integrates multiple specialized drafters into Speculative Decoding (SD) and employs multi-armed bandit sampling to dynamically select the optimal drafter. The authors conducted experiments across black-box and white-box SD methods, validating the superiority of the proposed method over traditional single-drafter approaches such as Lookahead and Eagle.

**Strengths:**

1. The exploration of this work towards Speculative Decoding with multiple specilaized drafters is meaningful and offers potential insights for the academic community. The standard single-drafter approach, typically generalized for natural language tasks, may not be optimal for domain-specific applications, such as translation. This work makes investigations to specilaized drafters across various domains such as QA, Translation, Math, and Summarization, etc, which demonstrates interesting and meaningful findings.
2. The authors conduct comprehensive experiments with both black-box and white-box SD methods, validating the effectiveness of MetaSD across diverse input domains. MetaSD achieves a promising 1.6x-3.7x speedup across these tested scenarios. The experimental settings are illustrated in detail.
3. The paper also provides an in-depth analysis of various impacts of MetaSD, addressing factors such as the switching cost of drafters, memory bandwidth bound, and KV cache re-calculation. These explanations effectively address some of my initial concerns.

**Weaknesses:**

1. **Extra computation overhead of MetaSD:** Unlike single-drafter SD methods such as Eagle, MetaSD leverages multiple specialized drafters for enhanced adaptation across various domains. While this approach shows promise, it introduces additional training and inference overhead that scales linearly with the number of drafters. Although the authors discuss certain aspects, like memory bandwidth limitations, further comparisons and quantitative data would clarify the computational overhead introduced by MetaSD. Specific metrics, such as MetaSD's training carbon footprint and memory bandwidth requirements during inference, should be included. Additionally, using multiple drafters increases serving complexity, particularly for multi-GPU environments.
2. **Lack of out-of-domain experiments**: In the main results, the authors utilize five specialized drafters that are fine-tuned on Code, Translation, Summarization, QA, and Math. Then, the MetaSD is evaluated on these five tasks, which can be regarded as in-domain tasks. For these tasks, the motivation of using Multi-armed bandit (MAB) is not convincing since drafters can be directly selected by estimating the similarity between the training and test data before inference. The strength of MAB would be more evident in out-of-domain scenarios, where selecting the optimal drafter is less straightforward. However, this experimental setting is missing in the work.
3. **Theoretical clarity**: Some definitions and theoretical statements in the paper lack clarity. For instance, in Line 214, the proof for the equation $\mathbb{E}\left[r_{i, t}^{BE}\right]=\frac{1-\alpha_i^{N_{\max }}}{N_{\max }\left(1-\alpha_i\right)} \mathbb{E}\left[r_{i, t}^{BD}\right]$ is missing. Similarly, Equation (6) in Line 1288, where the authors state $\mathbb{E}\left[N_{acc}(i, t)\right]=\frac{\alpha_i-\alpha_i^{N_{\max }+1}}{1-\alpha_i}$, appears to deviate from Equation (1) in [1]. Detailed explanations of these theoretical elements are necessary to prevent misinterpretation.
4. **Switching costs evaluation**: MetaSD needs to switch drafters during inference for optimal performance, which adds additional costs, such as the re-computation of drafting KV cache. To mitigate this, the authors propose Algorithm 4 to decrease the switching frequency, which first eliminates sub-optimal drafters as quickly as possible and then exclusively selects this drafter for the remaining rounds. Considering this, the total switching times of MetaSD during inference should be reported to offer readers an overall understanding of its extra KV re-computation cost. For example, the switching times between drafters in Table 3 and Table 4.

[1] Fast Inference from Transformers via Speculative Decoding. Leviathan et.al. ICML 2023.

**Questions:**

Most of my primary concerns are outlined in the weaknesses section above. Here are some additional, minor concerns:

- In Figure 1, the authors emphasize the use of the KV cache across different drafters. Could the authors clarify if they propose an efficient strategy to avoid re-calculating the KV cache or if they simply re-compute the KV cache for previous contexts upon drafter switching?

- The results of the SpS baseline should be included in Table 3, and the results of Eagle should be reported in Table 4.

---

> ### Author Response · Authors · 2024-11-20
> **Responses to Reviewer V45K (1/3)**
>
> Thank you for your careful review of our paper and your insightful and constructive comments. We have addressed your comments and updated our manuscript accordingly. Please find our detailed answers below.
>
> # W1. Extra computation overhead of MetaSD
>
> We would like to address each point in detail to clarify potential misunderstandings regarding the computational implications of our approach.
>
> ## Training Overhead
> While specialization may require additional training efforts compared to an OFA (One-size-Fits-All) drafter, we would like to emphasize that our approach can handle real-world scenarios where heterogeneous drafters already exist in public repositories. Our framework focuses on optimizing the utilization of such heterogeneous drafters, ensuring that the most suitable drafter is selected dynamically. This shifts the problem from retraining models to developing an effective strategy for utilizing pre-existing resources. Thus, while training specialized drafters may involve additional costs in some cases, the broader applicability and versatility of MetaSD provide substantial practical value. **In addition, issues regarding the cost of training drafters are not solely limited to our work, but all research regarding this area.**
>
> ## Inference Memory-Bandwidth Efficiency
> We would like to clarify a potential misunderstanding regarding inference memory bandwidth requirements. Although MetaSD employs multiple drafters, **this does not increase the memory bandwidth requirements during inference.** The drafters’ weights are preloaded into DRAM, ensuring no additional VRAM bandwidth is consumed compared to a single-drafter setup. Memory bandwidth refers to the data movement between VRAM and compute cores, which remains identical regardless of the number of drafters, as only one drafter operates on the VRAM-resident data at a time. The following table provides a comparison:
>
> | **Metric**            | **Single-Drafter Method** | **MetaSD**      |
> |------------------------|---------------------------|-----------------|
> | DRAM Memory Usage      | 17 GB                    | 19 GB (+2 GB)   |
> | VRAM Bandwidth         | Identical                | Identical       |
>
> By ensuring that only the active drafter interacts with the VRAM, MetaSD does not increase VRAM bandwidth demands, maintaining parity with single-drafter approaches during inference.
>
> ## Serving Complexity
> Using multiple drafters in MetaSD does not inherently increase serving complexity. Modern distributed systems already employ model parallelism techniques to allocate workloads effectively across multiple GPUs. In MetaSD, the drafters are evenly distributed across available GPUs, with each GPU independently handling its assigned drafter without added coordination costs. This design ensures that:
>
> - **Load Balancing**: Drafters are distributed across GPUs based on their assigned tasks, maintaining equivalent complexity to single-drafter systems.
>
> - **Minimal Communication Overhead**: MetaSD requires no additional inter-GPU communication beyond standard model parallelism setups.
>
> Consequently, the serving complexity of MetaSD aligns with that of traditional parallelized single-drafter systems.
>
> ## Justification of Overhead
> The modest increase in DRAM memory usage (+2 GB) and marginal training cost for specialized drafters is justified by the significant performance gains achieved through adaptive optimization. MetaSD dynamically selects the most suitable drafter for each task, consistently outperforming single-drafter methods across a wide range of scenarios, as highlighted in our experimental results.
>
> Furthermore, MetaSD's adaptability addresses an important real-world challenge: utilizing publicly available, pre-trained heterogeneous drafters effectively. By offering a generalizable strategy for optimizing these resources, MetaSD provides practical value beyond specialized retraining, supporting diverse and evolving task requirements.

---

> > ### Author Response · Authors · 2024-11-20
> > **Responses to Reviewer V45K (2/3)**
> >
> > # W2. Lack of out-of-domain (OOD) experiments
> >
> > We acknowledge the reviewer’s concern and agree that out-of-domain experiments can be crucial for demonstrating the strength of our MAB framework. To address this, we conducted additional OOD experiments using Alpaca-Finance and RAG datasets, which lie outside the domains of the specialized drafters used in our main experiments. The results are summarized in `Table9` of `Appendix F.8`.:
> >
> > ### **Key Observations**
> > 1. **Superior Adaptability**: MetaSD outperforms both OFA drafters and individual specialized drafters in these out-of-domain settings.
> >
> > 2. **Limitations of Similarity-Based Selection**:
> >    - **Increased Computational Cost**: Computing similarity between sentence embeddings requires passing the context through an encoder-only network to obtain the embeddings. This additional cost significantly increases inference time.
> >    - **Risk of Misclassification**: Even with embeddings, achieving high classification accuracy for selecting the correct drafter remains challenging. Errors in classification can lead to suboptimal drafter selection, further degrading performance. For instance, the difference in performance between Math drafter and UCB drafter becomes marginal at higher input lengths, as shown below:
> >
> >      | **Tokens Per Second** | **Math Drafter** | **OFA Drafter** | **MetaSD-UCB** |
> >      |-----------------------|------------------|-----------------|----------------|
> >      | **128 Tokens**        | 56.764           | 39.462          | **52.329**     |
> >      | **Inference Time (s)**| 2.255            | 3.244           | **2.446**      |
> >
> > # W3. Theoretical clarity
> >
> > We include the proof of the relation between the expectations of the two rewards at the end of `Appendix G.2`. Our result on $\mathbb{E}[N_{acc}(i,t)]$ corresponds to equation (1) in [1]. The apparent difference arises because our expectation pertains to the number of accepted tokens in each round, whereas equation (1) in [1] refers to the total number of generated tokens, which is always one more than the number of accepted tokens. For completeness, we have included these clarifications in our revised manuscript.
> >
> > # W4. Switching costs evaluation
> >
> > We agree that reporting the switching frequency is critical for evaluating the computational efficiency of our framework. Below, we provide the observed switching frequencies during inference for the experiments in `Tables 3-4`.
> >
> > ## Switching Frequency Results
> >
> > The average number of drafter switches during inference is as follows:
> >
> > | **Task**       | **Average Switching Times (SpS, Noisy)** | **Average Switching Times (Eagle, Noisy)** |
> > |-----------------|-----------------------------------------------|------------------------------------------------|
> > | **Code**       | 1.82                                         | 1.70                                          |
> > | **Translation**| 1.55                                         | 1.15                                          |
> > | **Sum**        | 2.25                                         | 2.11                                          |
> > | **QA**        | 1.97                                         | 2.36                                          |
> > | **Math**     | 2.19                                         | 2.37                                          |
> >
> > ## Key Observations
> >
> > 1. Extremely Low Switching Frequency: MetaSD switches drafters only 1–2.5 times per task over 100+ SD rounds, minimizing overhead. Future work may explore higher switching rates for non-stationary generation tasks.
> >
> > 2. Negligible KV Cache Cost: Low switching rates make KV cache recomputation costs negligible, even for tasks with long contexts, ensuring competitive inference speeds with single-drafter approaches.

---

> ### Author Response · Authors · 2024-11-20
> **Responses to Reviewer V45K (3/3)**
>
> # Q1. Clarification on computing KV-caches
>
> In our paper, the KV cache is recalculated for the previous context whenever a drafter switch occurs. However, we found that this recalculation introduces negligible computational overhead, even for relatively long contexts. This is due to the limited cost associated with prefilling the KV cache of a small drafter (e.g., In the case of Eagle drafter, one layer of KV cache is computed for unseen context), which is highly efficient.
>
> ## Additional Experiments for Efficient KV Cache Strategies
> To further validate our framework's efficiency, we additionally experimented with incorporating StreamingLLM [2], which can avoid full KV cache recalculation. The results demonstrate that this approach achieves comparable performance, confirming the robustness of MetaSD:
>
> | **Task**      | **MetaEagle-UCB (Recomputing KV)** | **MetaEagle-UCB with StreamingLLM** |
> |---------------|---------------------------------|----------------------------------|
> | **Code**      | 3.724                           | 3.624                            |
> | **Translation** | 2.318                           | 2.352                            |
> | **Summarization**       | 3.057                           | 2.986                            |
> | **QA**       | 2.641                           | 2.654                            |
> | **Math**    | 3.520                           | 3.338                            |
>
> ### **Key Observations**
> 1. **Minimal Impact of KV Recalculation**: The results show that even with full KV cache recalculation, MetaEagle-UCB maintains strong performance, highlighting that this process introduces minimal computational overhead.
>
> 2. **StreamingLLM Effectiveness**: Incorporating StreamingLLM techniques marginally improves performance in some cases (e.g., Translation and QA), while maintaining a similar overall performance profile.
>
> # Q2. Additional baseline
>
> We conducted additional experiments with an One-size-Fits-All (OFA) drafter for both black-box and white-box settings. These OFA drafters were trained on the mixed datasets spanning all tasks, ensuring a direct comparison to the specialized-drafter-based MetaSD framework. **The results in `Table3` and `Table4` of our revised manuscript show that while the OFA drafter performs well, our MetaSD framework outperforms OFA drafter across most tasks, demonstrating the strength of our method.** The OFA Eagle drafter performs relatively well. **However, it is still outperformed by MetaSD (i.e., MetaEagle-UCB).**
>
> **References**
>
> [1] Leviathan et.al., Fast Inference from Transformers via Speculative Decoding. ICML 2023.
>
> [2] EFFICIENT STREAMING LANGUAGE MODELS WITH ATTENTION SINKS. ICLR 2024.

---

> ### Comment · Reviewer_V45K · 2024-11-21
>
> Thank you for your detailed response!
>
> It has addressed most of my concerns, and I do appreciate the huge efforts you’ve put into this paper and the response. Therefore, I'm raising my score to **6**. That said, I have some follow-up questions and suggestions for further improvement:
>
> - **Comparisons with Concurrent Work**: I recommend including discussions of related concurrent work, such as SWIFT[1]. SWIFT dynamically optimizes and switches to a more efficient drafter using layer skipping, which shares similarities with the Bandit approach in this manuscript. Additionally, I encourage evaluating the Bandit approach in a dynamic data stream with multi-task OOD data, as demonstrated by SWIFT in Figure 7. This would provide a more comprehensive evaluation (you don't have to do it during the rebuttal period). Besides, more discussions with [2] should be included since this method also optimizes SD performance on the fly.
> - **Computation Overhead**: The analysis of additional computational overhead introduced by the Bandit mechanism should be included in the revised manuscript to provide a clear understanding of its practical implications.
> - **Regarding W2**: I respectfully disagree with the statement, "*Computing similarity between sentence embeddings requires passing the context through an encoder-only network to obtain the embeddings*." Cosine similarity can be directly calculated on the input representations after the pre-filling stage of the target LLM, without requiring an additional encoder. However, I acknowledge the authors’ valid point that this naive strategy could pose challenges for tasks with highly similar representations.
> - **Regarding W4**: The manuscript should report **switching accuracy** metrics. Specifically, I am curious whether the low switching frequency observed leads to high switching accuracy (i.e., the Bandit mechanism consistently selects the optimal drafter). If this is not the case, the low switching frequency could undermine the effectiveness of the Bandit mechanism, and further clarification or analysis would be valuable.
>
> Thank you again for your thorough response. I look forward to further discussions and may consider increasing my score upon addressing these points.
>
> [1] SWIFT: On-the-Fly Self-Speculative Decoding for LLM Inference Acceleration. Xia et al. Arxiv 2024.
>
> [2] Online speculative decoding. Liu et al. ICML 2024.

---

> > ### Author Response · Authors · 2024-11-21
> > **Responses to Comments by Reviewer V45K (1/2)**
> >
> > # Follow-Up 1. Comparisons with Concurrent Work
> >
> > We appreciate the opportunity to draw connections between MetaSD and concurrent advancements in SD! Both `SWIFT` and `Online Speculative Decoding (OSD)` methods address distinct yet complementary aspects of SD.
> >
> > - **SWIFT: On-the-Fly Self-Speculative Decoding for LLM Inference Acceleration** introduces a dynamic optimization framework that employs layer-skipping strategies to reduce latency during decoding. SWIFT explores the combinatorial space of layer-skip indices, employing Bayesian Optimization to efficiently search for optimal configurations. While SWIFT’s approach dynamically adjusts internal model operations, MetaSD focuses on selecting among heterogeneous external drafters to optimize decoding efficiency. Both methods share a commitment to dynamic adaptation but target different optimization levels. While generally superior, MetaSD is even effective in scenarios where multiple pre-trained or publicly available drafters are accessible, allowing it to provide robust, adaptive solutions for diverse or unknown drafter configurations.
> >
> > - **Online Speculative Decoding (OSD)** addresses challenges associated with the static nature of traditional SD by introducing an online adaptation mechanism. OSD periodically updates draft models during deployment using knowledge distillation, improving token acceptance rates and inference speed by dynamically aligning draft models with evolving user query distributions. While OSD excels in refining a single draft model for domain-specific optimization, MetaSD is tailored for multi-drafter scenarios, enabling effective selection among diverse specialized or heterogeneous drafters. Additionally, OSD relies on continuous updates to a smaller draft model, whereas MetaSD’s plug-and-play paradigm supports real-time decisions without the need for such updates.
> >
> >     - While MetaSD and OSD differ in their primary focus—drafter selection versus drafter adaptation—they share complementary strengths. In scenarios where drafters must be adapted dynamically, integrating OSD’s update mechanism with MetaSD’s selection strategy could further enhance robustness. Conversely, MetaSD’s capability to handle heterogeneous black-box drafters offers a practical solution for real-world settings where drafter customization or retraining is infeasible.
> >
> > - In conclusion, MetaSD, SWIFT, and OSD each address unique challenges in SD. By advancing distinct aspects of SD, these methods collectively push the boundaries of what speculative decoding can achieve, offering promising directions for future research and integration.
> >
> > We will incorporate these discussions into our revised manuscript!
> >
> > # Follow-up 2. Computation Overhead
> >
> > Thank you for pointing this out. We agree that a detailed analysis of the computational overhead introduced by the Bandit mechanism is essential for understanding its practical implications. While we’ve already put a discussion on it in `Section 5` and `Appendix I.1`, in the revised manuscript, we further include a discussion in the `Appendix I.2` to elaborate on this topic.
> >
> > # Follow-up 3. W2. Cosine Similarity
> >
> > Thank you for providing an alternative perspective on computing similarity. We agree that cosine similarity can indeed be calculated directly on the input representations after the pre-filling stage of the target LLM, bypassing the need for an additional encoder-only network. This is a valid approach.
> >
> > However, we would like to highlight the potential challenges in real-world scenarios where tasks exhibit highly similar representations. In such cases, relying solely on input-level similarity can lead to misclassification and suboptimal drafter selection. For example, tasks like summarization and QA, or translation between similar languages (e.g., French to English vs. Spanish to English), may produce embedding representations that are difficult to differentiate accurately using cosine similarity alone.
> >
> > This challenge underscores the importance of MetaSD’s token-level feedback mechanism, which dynamically adapts drafter selection based on actual decoding performance rather than static similarity measures. We believe this adaptability is crucial for robust performance, especially in mixed-domain or ambiguous task settings.
> >
> > **We appreciate your acknowledgment of this limitation in the naive strategy and will clarify this discussion in the camera-ready manuscript to incorporate your valuable input.** Thank you again for raising this point!

---

> > > ### Author Response · Authors · 2024-11-21
> > > **Responses to Comments by Reviewer V45K (2/2)**
> > >
> > > # Follow-up 4. W4. Switching Accuracy Metrics
> > >
> > > Thank you for your feedback regarding switching accuracy metrics. Generally, a low switching frequency does correlate with high switching accuracy, and our observations align with this trend. We acknowledge the importance of providing more comprehensive metrics to further validate the Bandit mechanism's effectiveness and will consider additional analysis for the camera-ready version.
> > >
> > > It is worth noting that switching behavior can vary between the initial selection phase and intermediate steps, potentially requiring a more nuanced evaluation. For instance, early-stage switches may have a different impact on overall performance compared to switches later in the decoding process. **We plan to explore more detailed metrics, such as switching frequency per phase and task-level switching accuracy in camera-ready.**
> > >
> > > That said, our current results already provide metrics for switching times and switching accuracy during intermediate steps. Additionally, the minimal deviation in real-world latency compared to the optimal configuration suggests that our Bandit mechanism performs well in practice. This reinforces our confidence in the effectiveness of the proposed approach.
> > >
> > > Lastly, as mentioned in the `future work section` of our manuscript, incorporating advanced methods such as reinforcement learning or contextual Bandits holds promise for further improving drafter selection and dynamic adaptation. We believe these strategies could enhance performance even in more complex scenarios.
> > >
> > > We appreciate your valuable suggestion and will work to provide richer metrics and insights in the revised version. Thank you for helping us improve this aspect of our work!

---

> ### Comment · Reviewer_V45K · 2024-11-22
>
> Thank you for your thorough and detailed responses. I have no further concerns and have updated my score accordingly.

---

> > ### Author Response · Authors · 2024-11-25
> > **Responses to Reviewer V45K**
> >
> > Dear Reviewer V45K
> >
> > We sincerely appreciate the time and effort you dedicated to reviewing our paper and your positive feedback on our work.

---

### Official Review · Reviewer_Tyx3 · 2024-11-02

**Soundness:** 1
**Presentation:** 3
**Contribution:** 1
**Rating:** 3
**Confidence:** 4

**Summary:**

This paper explored the problem of specializing and choosing draft models to accelerate speculative decoding. The task of routing a proper draft model can be seen as a multi-armed bandit problem; the author developed and experimented MetaSD-UCB method based on the existing UCB algorithm to address the problem.

**Strengths:**

- The idea of applying the UCB algorithm to speculative decoding is novel.
- The paper is overall well-presented.

**Weaknesses:**

- The evaluations do not accurately reflect real-world use cases. The primary goal of MetaSD-UCB is to route inputs to the appropriate draft model. However, during the experiments, the same prompt template is applied across all instances within a dataset, making it trivial to differentiate between datasets. In real-world scenarios, prompts can vary significantly from one query to another, which makes the multi-armed bandit problem much more complex than what is represented in the evaluations.
- The effectiveness of the multi-armed bandit approach is limited. Given that the datasets used in the experiments are distinctly different and well-defined, a rule-based system or a simpler machine learning model could easily achieve high accuracy in selecting the appropriate draft model, often with minimal latency compared to using the draft model itself. In contrast, the proposed method requires executing a speculative decoding step with each draft model, which increases the number of tokens processed by the target model fivefold during the initial step—an expensive operation, particularly with tree attention. Despite this, the average accuracy achieved in the experiments for model selection is below 80 percent.
- There is a lack of experiments assessing throughput. Given that the proposed method utilizes an ensemble of draft models, it is more likely to be deployed in a distributed system rather than on a personal computer or mobile device. As a result, throughput should be prioritized as a key evaluation metric over latency. Even for latency evaluation, a batch size of 1 is not practical.
- Problem with machine translation datasets. Vicuna, as well as LLaMA 2, are not multilingual language models and are not designed for machine translation tasks involving languages such as Chinese and Japanese.
- The training datasets are the same as the evaluation dataset.
- The vertical spacing on the last page is very small compared to other pages.

**Questions:**

1. Can you include an evaluation of Eagle without specialization on the same hardware?
2. Can you fine-tune a draft model on all the datasets experimented to show if it is necessary to have a specialized model?

---

> ### Author Response · Authors · 2024-11-20
> **Responses to Reviewer Tyx3 (1/2)**
>
> Thank you for your careful review of our paper and your insightful and constructive comments. We have addressed your comments and updated our manuscript accordingly. Please find our detailed answers below.
>
> # W1. Invalid evaluations for real-world use cases.
>
> We acknowledge the concern that using the same prompt template across all instances within a dataset might not fully reflect real-world scenarios. To address this, we conducted additional experiments with **perturbed prompts**, where the prompt for each query varied slightly but was semantically equivalent to the original. These perturbed prompts were generated using GPT-4, ensuring natural and diverse variations. For example:
>
> - **Translation Task**: Original prompt “Translate German to English” was perturbed to “Convert this text from German to English.”
>
> - **Summarization Task**: Original prompt “Summarize:” was perturbed to “Provide a concise overview of the following text:”
>
> ## **Key Observations**
>
> The main results and descriptions are detailed in `Table 10` and `Table 11` of `Appendix F.9`.
>
> 1. **Performance Impact Across All Methods**:
>    - Perturbed prompts led to a drop in performance for all drafters, including One size Fits All (OFA; which is trained on a mixed dataset) and individual specialized drafters. **This indicates that the issue is not limited to our approach but affects all SD methods.**
>    - The observed degradation reinforces that real-world variability in prompts can challenge any static drafter selection strategy.
>
> 2. **MetaSD’s Robustness**:
>    - Despite the variability introduced by perturbed prompts, **MetaSD consistently outperforms all baselines**, including OFA and individual drafters. This result highlights the advantage of MetaSD’s dynamic selection mechanism, which adapts to token distributions at every round rather than relying solely on prompt characteristics.
>
> # W2. Limited effectiveness of the MAB approach
>
> While the datasets used in our experiments are indeed distinctly different and well-defined, our method is designed to handle more complex, real-world scenarios where:
>
> - The boundaries between datasets or task types are less clear.
>
> - The performance of each drafter can vary dynamically depending on the token distribution as the decoding progresses.
>
> - Out-of-domain tasks introduce additional complexity, which static rule-based systems or simpler machine learning models cannot handle effectively.
>
> While it is true that MetaSD involves SD rounds for multiple drafters, the additional computational cost is mitigated by the **efficiency gains in overall throughput.** The reported accuracy of drafter selection (below 80%) should not be viewed in isolation, as it reflects the complexity of real-world token prediction rather than a static task classification problem. While rule-based systems may appear efficient in well-defined scenarios, they lack the flexibility to adapt to:
>
> - Mixed-domain or evolving tasks where token distributions do not align clearly with predefined rules.
>
> - Real-time feedback from the decoding process, which is a critical component of MetaSD's adaptive optimization.

---

> > ### Author Response · Authors · 2024-11-20
> > **Responses to Reviewer Tyx3 (2/2)**
> >
> > # W3. Throughput experiment
> >
> > We understand the need to evaluate throughput because batch processing is critical. We address the concern and clarify why our method maintains throughput efficiency in `Appendix F.10`.
> >
> > Our drafter management mechanism does not lead to throughput depletion compared to single-drafter methods due to the following reasons:
> >
> > - No Increase in Memory Bandwidth Requirements: The drafters’ parameters are preloaded into DRAM and do not require frequent movement to VRAM for computation. This ensures that the number of memory movements and memory size for movement remain identical to single-drafter methods, even in scenarios with multiple drafters.
> >
> > - Scaling Across Batches: Since the computational structure remains unchanged, performance observed in single-batch scenarios translates directly to multi-batch settings, maintaining throughput consistency.
> >
> > We further conducted experiments comparing throughput following the same settings in the original Eagle paper. The results confirm that the performance of MetaSD scales well without significant degradation:
> >
> > | **Throughput** (RTX 3090 24GB) | **Single OFA Eagle (Tokens/sec)** | **MetaEagle-UCB (Tokens/sec)** |
> > |----------------|--------------------------------|-------------------------|
> > | Speedup         | x 2.235                           | x 2.427                      |
> >
> >
> >
> > # W4. Problem with MT datasets
> >
> > While we acknowledge that Vicuna and LLaMA 2 are not inherently multilingual models, we chose these models for their instruction-tuned capabilities. Importantly, the purpose of our evaluation was not to measure generation quality but to assess **latency** improvements under SD frameworks. This experimental settings are following the settings by prior works such as:
> >
> > - **"Towards Fast Multilingual LLM Inference: Speculative Decoding and Specialized Drafters" (EMNLP 2024 main)**
> >
> > - **"Online Speculative Decoding" (ICML 2024)**
> >
> > Both studies utilize similar setups to evaluate decoding efficiency over multiple multilingual datasets..
> >
> > ## **Key Clarifications**
> >
> > 1. **Generality Across Domains**: While our evaluation includes MT tasks, our framework is not limited to this domain. We also conducted experiments across **code generation, math problem solving, question answering, summarization**, and additional out-of-domain tasks such as `Alpaca-Finance` and `RAG datasets` (`Table3`, `Table4`, `Table9`).
> >
> > 2. **Comprehensive Scope**: To our best knowledge, our work evaluates SD performance across the **broadest range of domains** in the current existing literature. This ensures that MetaSD’s effectiveness is demonstrated across diverse use cases.
> >
> >
> >
> > # W5. Training dataset are the same as the evaluation dataset.
> >
> > We would like to clarify that while there is some overlap in the types of datasets (e.g., machine translation), the training and evaluation datasets differ significantly in both domains and instances. Below, we provide details for clarification:
> >
> > - Code Generation: Training was performed using the Code-Alpaca dataset, but evaluation was conducted on the MT-Bench dataset. This ensures that the evaluation reflects performance on unseen data.
> >
> > - Math Problem Solving: Training utilized the MetaMathQA dataset, whereas evaluation was conducted on GSM8K, a completely different dataset focused on general math problems.
> >
> > - Machine Translation: While the training and evaluation datasets belong to the same domain (e.g., English-to-Chinese translation), the specific instances used in training and testing are distinct.
> >
> > # W6. Vertical Spacing on the Last Page
> >
> > Thank you for pointing this out. We adjust the formatting in our revised manuscript.
> >
> > # Q1. & Q2. Additional baselines for OFA drafter training over mixed data
> >
> > We conducted additional experiments with an One-size-Fits-All (OFA) drafter for both black-box and white-box settings. These OFA drafters were trained on the mixed datasets spanning all tasks, ensuring a direct comparison to the specialized-drafter-based MetaSD framework. **The results in `Table3` and `Table4` of our revised manuscript show that while the OFA drafter performs well, our MetaSD framework outperforms OFA drafter across most tasks, demonstrating the strength of our method.** The OFA Eagle drafter performs relatively well. **However, it is still outperformed by MetaSD (i.e., MetaEagle-UCB).

---

> > > ### Comment · Reviewer_Tyx3 · 2024-11-22
> > >
> > > We thank the authors for their clarification and additional experiments. However, the new experiments are not comprehensive enough to demonstrate the effectiveness of MetaSD-UCB compared to other ensembling approaches. Therefore, I will maintain my current score.

---

> > > > ### Author Response · Authors · 2024-11-22
> > > > **Response to Reviewer Tyx3**
> > > >
> > > > Thank you for your feedback and for taking the time to review our revised manuscript. We respectfully disagree with your assessment that our new experiments are not comprehensive enough. Below, we outline the steps we took to address your original concerns and clarify why we believe these efforts adequately demonstrate the effectiveness of MetaSD-UCB.
> > > >
> > > > ---
> > > > # Clarifications and Specific Points
> > > >
> > > > 1. **We conducted all requested experiments**
> > > >    In your initial review, you requested additional evaluations, including:
> > > >    - **Perturbed Prompt Experiments**: We introduced perturbed prompts to assess performance under more realistic, varied input scenarios. These experiments highlighted that MetaSD maintains robustness and consistently outperforms baselines like OFA, even when prompts vary semantically.
> > > >    - **Heterogeneous Batch Scenarios**: We conducted experiments for worst-case edge cases where all instances in a batch correspond to different tasks. These experiments revealed the limitations of MetaSD in large heterogeneous batches while highlighting its continued strength in small batch sizes and homogeneous settings.
> > > >    - **OFA Drafter Comparisons**: We added comparisons against OFA drafters trained on mixed datasets across both black-box and white-box settings. While OFA drafters performed well, MetaSD consistently outperformed them across most tasks, validating the dynamic drafter selection mechanism.
> > > >
> > > >    We believe these additions addressed the core concerns raised in your initial review. And simultaneously, no ensembling methods are suggested in your initial review.
> > > >
> > > > 2. **MetaSD-UCB as a novel contribution**
> > > >    Our work is the **first to address speculative decoding (SD) with multiple heterogeneous drafters**. This is a fundamentally different problem than standard ensembling or naive SD approaches. MetaSD leverages the **memory-bound nature** of SD to dynamically optimize performance without increasing memory bandwidth demands. This approach is not directly comparable to generic ensembling methods, which are not designed to handle memory-bound SD settings.
> > > >
> > > > 3. **Request for specific alternatives**
> > > >    While your review suggests a comparison with `other ensembling approaches`, no specific methods for ensemble-based SD were referenced in your feedback. If there are particular approaches you believe are relevant, we would greatly appreciate if you could suggest them. This would allow us to better position MetaSD in the broader speculative decoding landscape. However, we must emphasize that speculative decoding with **heterogeneous drafter selection** is fundamentally distinct from simple ensembling strategies.
> > > >
> > > > 4. **Unclear areas needing improvement**
> > > >    While we value constructive criticism, we find the suggestion that the "new experiments are not comprehensive enough" to be vague. Without concrete examples of specific limitations or additional experiments required, it becomes difficult to address these concerns further. We respectfully request more explicit feedback to help us understand which specific aspects of the study remain insufficient.
> > > >
> > > > ---
> > > >
> > > > # **Broader context: contribution and innovation**
> > > >
> > > > We would also like to reiterate the key contributions and strengths of our work:
> > > > - **Theoretical guarantee**: We propose a novel multi-armed bandit approach for speculative decoding with formal theoretical guarantees.
> > > > - **Novelty**: MetaSD-UCB is the **first framework to explore multiple heterogeneous drafters for SD**, addressing challenges that traditional ensembling methods cannot solve.
> > > > - **Practical speedup**: MetaSD achieves substantial improvements in speedup ratios compared to existing SD methods, demonstrating its practical utility.
> > > > - **Comprehensive evaluation**: To our knowledge, our work evaluates speculative decoding performance across the broadest range of task categories, including in-domain, out-of-domain, and perturbed prompts, making our study one of the most thorough in this field.
> > > >
> > > > ---
> > > >
> > > > ### **Respectful request**
> > > >
> > > > We believe our work pushes the edge of speculative decoding research and addresses a novel and underexplored problem. **While constructive suggestions are always appreciated, introducing new and abstract requirements after revisions may not provide a fair assessment of the work's merits. We respectfully request that our contributions be evaluated based on the clear and explicit goals set during the initial review.**
> > > >
> > > > If there are additional suggestions for experiments or comparisons, we welcome them as valuable input for future research, but we believe our current manuscript sufficiently demonstrates the effectiveness and novelty of MetaSD-UCB.
> > > >
> > > > Thank you again for your time and for considering our response. We hope this clarifies our position and highlights the strengths of our contribution.
> > > >
> > > > -Authors

---

> ### Author Response · Authors · 2024-11-22
> **Response to Reviewer Tyx3**
>
> Thank you for elaborating on your concerns. We would like to clarify and address the points raised.
>
> ---
> # 1. On Real-World Use Cases and Prompt Variations
>
> We want to emphasize that the experiments with perturbed prompts were conducted by varying the prompts for **every query**, not just using a single variation. **The example provided in the response was one of many variations used.** The results reflect this setup and demonstrate that the performance degradation affects all methods, not just MetaSD. **This highlights a broader challenge that is not unique to our approach (Most methods face this challenge).**
>
> ---
>
> # 2. On the Practical Value of MetaSD
>
> While it is true that MetaSD-UCB does not outperform individual specialized draft models in every case, its strength lies in its **robustness across diverse scenarios**, including in-domain, out-of-domain, and perturbed settings. These scenarios reflect the complexity of real-world tasks, where heterogeneous drafters are often pre-trained on unknown data or lack clear task-specific boundaries.
>
> Although MetaSD does not consistently outperform specialized draft models on their respective tasks, it is highly robust across diverse tasks. On average, MetaSD achieves performance comparable to specialized models when evaluated across a mix of tasks. **For context, we analyzed MetaSD's results by comparing them to a simulated classification-based approach, where the classifier is assumed to have `zero inference time`, and found that MetaSD effectively achieves:**
>
> - **87%** of the classification accuracy of specialized drafters in **MetaSps** settings.
> - **90%** of the classification accuracy in **MetaEagle** settings.
>
> This demonstrates that MetaSD delivers near-specialized performance from the perspective of classification, without relying on pre-trained parameters (already working as a good classification predictor). Even without offline training or fine-tuning, MetaSD consistently provides strong, adaptive performance across diverse tasks. Moreover, this robustness extends to **out-of-domain** and **perturbed scenarios**, which often present `significant challenges for classification-based or static systems`. On the another line, MetaSD can be also supported by multiple theorems, which can be huge difference from the classification-based system.
>
> Additionally, MetaSD’s reliance on the **multi-armed bandit (MAB)** algorithm—without any learnable parameters—makes it highly practical and computationally efficient for `real-world on-the-fly applications`.
>
> ---
>
> # 3. On Classification-Based Approaches
> We appreciate your suggestion regarding classification-based approaches and agree that they could serve as a useful baseline. However, as noted, classification models rely on pre-trained embeddings and are subject to classification errors. This introduces challenges, particularly in:
> - **Out-of-Domain Scenarios**: Classification models often generalize poorly to tasks outside their training domain.
> - **Dynamic Adaptation**: Classification requires offline pre-training and cannot adapt to real-time token-level feedback (on-the-fly), unlike MAB approaches.
>
> For camera-ready, we plan to evaluate a simple classification-based approach using **Google-BERT-Base-Uncased** embeddings with a linear layer to predict the optimal drafter for each query. This experiment will provide an additional baseline for comparison.
>
> ---
>
> # 4. Broader Motivation and Contributions
> As highlighted in our revised manuscript, one of the key motivations of MetaSD is to address scenarios involving **heterogeneous drafters** where training data and performance characteristics are unknown. MetaSD provides a robust and simple alternative for on-the-fly decision-making, avoiding the need for pre-trained models or extensive offline fine-tuning.
>
> While classification-based methods may serve as a useful baseline, their inability to dynamically adapt makes MetaSD a practical and effective solution, particularly in scenarios requiring robust performance across diverse tasks.

---

> ### Author Response · Authors · 2024-11-25
> **Responses with Classification-based Routing to Reviewer Tyx3**
>
> Dear Reviewer Tyx3,
>
> In response to your concerns regarding classification-based routing, we conducted additional experiments using BERT-based models for routing across diverse tasks (coding, translation, summarization, QA, and math reasoning). Specifically, we evaluated routing performance using pre-trained BERT-Base-Uncased (110M) and BERT-Large-Uncased (330M) with both cosine similarity and fine-tuned softmax heads. All experiments are conducted with a single A100 GPU. (`Note that by considering the small drafter used in SpS has 68M parameters, both models are quite large.`) The results are as follows:
> - Cosine Similarity (Pre-trained):
>     - BERT-Base: 24.50% accuracy
>     - BERT-Large: 30.00% accuracy
> - Fine-Tuned BERT with new Softmax Head:
>     - BERT-Base: 74.54% accuracy
>     - BERT-Large: 85.58% accuracy
>
> We also measured the average latency speedup across all tasks:
>
> **Average Latency Speedup Ratio (In-domain)**
> | **Routing Method**       | **SpS Speedup** | **Eagle Speedup** |
> |--------------------------|-----------------|-------------------|
> | Fine-Tuned BERT-Base     | 1.772           | 2.759            |
> | Fine-Tuned BERT-Large    | 1.741           | 2.797            |
> | MetaSpS-UCB              | 1.912           | —                 |
> | MetaEagle-UCB            | —               | 3.052            |
>
>
> Our MetaSD demonstrates higher robustness than classification-based routing, particularly in perturbed and out-of-domain (OOD) scenarios.
>
> **Perturbed Prompts Speedup Ratio**
>
> | **Routing Method**       | **SpS Speedup** | **Eagle Speedup** |
> |--------------------------|-----------------|-------------------|
> | Fine-Tuned BERT-Base     | 1.456           | 2.245            |
> | Fine-Tuned BERT-Large    | 1.567           | 2.323            |
> | MetaSpS-UCB              | 1.820           | —                 |
> | MetaEagle-UCB            | —               | 2.912            |
>
> **Out-of-Domain Speedup Ratio (RAG and Finance)**
>
> | **Scenario**             | **RAG Speedup** | **Finance Speedup** |
> |--------------------------|-----------------|---------------------|
> | Fine-Tuned BERT-Base (SpS) | 1.645          | 1.410              |
> | Fine-Tuned BERT-Large (SpS)| 1.638          | 1.398              |
> | MetaSpS-UCB              | 1.799           | 1.436              |
> | Fine-Tuned BERT-Base (Eagle)| 2.132         | 2.413              |
> | Fine-Tuned BERT-Large (Eagle)| 2.110        | 2.426              |
> | MetaEagle-UCB            | 2.238           | 2.517              |
>
> These results demonstrate that our framework is more robust and adaptable, particularly under challenging conditions like perturbed prompts and OOD tasks. Classification-based routing struggled with task confusion, especially between closely related tasks. For example, coding and math often share overlapping patterns in token distributions, making them harder to distinguish, while summarization and QA frequently confused sentence-level context. Even though translation appeared to perform well in classification, it was likely aided by distinct vocabulary distributions unique to translation tasks, which reduce ambiguity. (Note: ` While classification might be better suited for multilingual tasks due to distinct vocabularies,  in other cases, it was not as effective.`)
>
> **A key reason why MAB demonstrates stronger performance in this setting lies in our novel block-divergence reward.** This reward effectively captures the distributional alignment between the target model and the drafter, allowing the MAB to dynamically adapt and select the optimal drafter based on real-time feedback (i.e. `policy-routing`). In contrast, classification-based routing does not consider this alignment, leading to suboptimal routing decisions that fail to exploit token-level distribution patterns.
>
> **Of course, we believe that classification-based models could potentially address these issues more effectively if designed with a deeper consideration of token-level distributional differences.** However, our focus is not on developing classification-based routing methods, and to the best of our knowledge, no established approaches currently exist in this space. Instead, we proposed a simple but efficient method for the novel problem (that we newly raise in this field) at hand. **We also believe that exploring classification-based routing as a future direction could offer an orthogonal and promising approach to improving routing efficiency.**
>
> We hope this detailed analysis clarifies the advantages of our approach and demonstrates why a simple classification-based method is insufficient. We will work to include these in the camera-ready.
>
> Thank you for your time and consideration.
>
> Best regards,
>
> -Authors

---

> ### Author Response · Authors · 2024-11-26
> **Further extended experiments**
>
> ## Nonstationary Environment
>
> To further **demonstrate the effectiveness of the MAB approach**, we conducted experiments on a **non-stationary translation task**, where each query required translating two different languages (French and Chinese) into English. The experiments were performed using a single RTX 3090 GPU, and the results are as follows:
>
> |                  | Drafter1 | Drafter2 | Drafter3 | Drafter4 | Drafter5 | Upper Bound for Classification-based Routing | MetaSpS-UCB |
> |------------------|----------|----------|----------|----------|----------|----|----------------|
> | Block Efficiency |    1.668 |    1.722 |    1.485 |    1.759 |    1.803 | 1.803 |             1.951 |
> | Speedup Ratio    |    1.429 |    1.492 |    1.289 |    1.539 |    1.581 |  1.581 |           1.722 |
>
> These results clearly demonstrate the **effectiveness of the MAB approach**, where MetaSD-UCB with $\alpha=0.1$ achieves a speedup ratio of **1.722**, which even exceeds the performance of the optimal drafter (1.581). This improvement arises because MAB dynamically adapts to the best drafter in the current environment through a combination of exploration and exploitation.
>
> For example, in a multilingual scenario, where the task is "Translate French and German to English, respectively," classification-based routing would be limited by the performance of the best single specialization (i.e., the speedup ratio of the optimal drafter as its upper bound). In contrast, MetaSD-UCB surpasses this upper bound by effectively adapting its policy dynamically, demonstrating a **unique advantage** over classification-based methods in non-stationary environments.
>
> ## Future Directions
>
> We are already encouraging further exploration in our paper's **Future Work** section, and we believe this type of **policy-routing** could benefit significantly from more advanced approaches like **contextual bandits** or **reinforcement learning** (RL). These methodologies may offer more sophisticated mechanisms for dynamic adaptation in non-stationary environments. Furthermore, from a **scalability perspective**, interesting future directions include:
>
> - Exploring how to cluster drafters effectively for system-level applications, such as those used in **retrieval-augmented generation (RAG)**.
> - Investigating how to design **OFA drafters** in an orthogonal manner to maximize complementarity across tasks.
>
> ## Clarification on Our Research Focus
>
> It’s important to highlight that **our work does not advocate training multiple specialized drafters instead of OFA drafters**. One of our main focus is to address the critical challenge of **dynamically routing tasks among pre-existing heterogeneous specialized drafters**. This is a practical and timely problem, especially as these specialized drafters are becoming widely available.
>
> For a deeper understanding of our motivation and scope, we strongly recommend revisiting the **`Motivation Section`** in the main paper and the **`Further Motivation Section`** in the `Appendix`. These sections provide a comprehensive view of why this research is both relevant and necessary for advancing speculative decoding systems.

---

### Official Review · Reviewer_v7AR · 2024-11-04

**Soundness:** 2
**Presentation:** 3
**Contribution:** 2
**Rating:** 3
**Confidence:** 4

**Summary:**

This paper introduces a simple framework, termed MetaSD, that incorporates multiple drafters into the speculative decoding process to address via multi-armed bandit sampling to dynamically allocate computational resources across various drafters, thereby improving overall generation performance.

**Strengths:**

The idea of this paper is neat.

The figures in this paper are clear and good.

The experiments are somehow satisfied because the datasets are randomly shuffled to create a non-stationary environment.

**Weaknesses:**

The definition of the acceptance rate is not clear and confusing. The authors state that “we relax this assumption and consider a more general scenario where the acceptance rate for each token follows a stationary distribution with mean $\alpha$” in Line 215. This means that the acceptance rate is in fact a **random variable**. However, in all the calculations in the manuscript, the authors just replace the random variable with its mean value.

Even if we treat the acceptance rates as real numbers,  the proof of Theorem 1 is not correct. In Lemma 4,  it is not clear why the expectation of $r^{BD}$ is equal to the acceptance rate. To be specific, in the definition of the block divergence reward, the total variation is conditioned on the prefix $x^{1:l(t)+j}$. We note that $x^{l(t)+1:l(t)+j}$ is the random variables generated by the draft model, not the pre-trained model. When taking expectation, these random variables take the distributions of the **draft models**. However, Theorem 1 of [1] shows that the average acceptance is equal to the expectation of $r^{BD}$ when $x^{l(t)+1:l(t)+j}$ takes the distribution of the **pre-trained model**.

The formulation of the bandit problem is incomplete, and the statement and the proof of Theorem 2 is inappropriate. Concretely, the proposed bandit problem should be considered in the probability space induced by the randomness of the pre-trained models and the draft models. This implies that the so-called total budget $B$ is indeed a **random variable** (in fact a stopping time). Thus, Theorem 2 states the results conditioned on the budget $B$, and **all the proofs should be built on the posterior distribution given $B$**. However, the current proof does not consider this.

While the paper emphasizes the design of the BD reward is novel, it does not necessarily align with the goal: a lower BD regret cannot indicate fewer rounds are needed.  According to Lemma 6 in the appendix, the performance of an algorithm under different regret/reward definitions, i.e., BD and number of rounds, can be different. From this perspective, the theoretical upper bound guarantee of the proposed algorithm is diminished, as the drafter with a better designed reward may not necessarily perform better in practice.
Some procedures in the proof are not correctly justified.

Line 1586: the equality is wrong. $\mathbb{E}[\tau^u(\pi^u, B)]$ can be much greater than $\mathbb{E}[\tau(\pi^{i^\star}, B)]$.

Line 1695: this inequality is wrong. The confidence radius never shrinks as $t$ increases.

In Algorithm 2, the hyper-parameter $\beta$ is chosen empirically according to Line 1631. However, $\beta$ is not reflected in the regret bound in some Theorems (with some specified $\beta$ in the others), as well as in the corresponding analyses.
In the bandits literature, the mean gaps $\{\Delta_i\}_{i=1}^K$ are critical, because they measure the hardness of the given problem instance. In this paper, some quantities involving the mean gaps are hidden in the constants (which are independent of $B$). While this may be acceptable for the asymptotic behavior where $B\to\infty$, they can be important in the finite $B$ case (which is always true in practice).

Minors:

It would be great to give a formal mathematical definition of $B$ and the stopping time where the algorithm stops. In addition, it would be better to consistently term $B$ as the "target sequence length", as "budget" refers to the time horizon in the bandits literature.


Overall, the formulation of the bandits problem is incomplete and the application of the existing bandits algorithms to speculative decoding is straightforward, given the previous literature. While the paper argues that the proposed BD reward is novel, a justification of its property (e.g., expectation) is clearly missing. In addition, an algorithm with less BD regret does not indicate better performance (at least theoretically). This hinders the theoretical contribution of the paper.

[1] M. Yin et al, A Theoretical Perspective for Speculative Decoding Algorithm

**Questions:**

1) why exp3 is significantly worse than ucb, even similar to the result of rand?

2) In Tables 3 and 4, the author why not provide the original eagle's result or train a general eagle with all the mixed data?

3) The 5 tasks are much less than the real setting, the author should study the scaling law of task numbers for metasd.

---

> ### Author Response · Authors · 2024-11-20
> **Responses to Reviewer v7AR (1/2)**
>
> Thank you for your careful review of our paper and your insightful and constructive comments. We have addressed your comments and updated our manuscript accordingly. Please find our detailed answers below.
>
> # W1. Clarity of the assumption on acceptance rate
>
> We denote the acceptance rate of drafter i in round t as $\alpha_{i,t}$ and assume this is a stationary random variable with mean $\alpha_i$. As the regret bound of a bandit algorithm is usually related to the expectation of each reward, $\alpha_i$ appears in the result in `Theorem 2` but this does not mean we replace $\alpha_{i,t}$ with $\alpha_i$ in any part of our analysis. For clarification, we add a formal definition of our assumption on acceptance rate in `Appendix G.6`.
>
> # W2. Validity of Theorem 1
>
> The expectation of BD reward in `Lemma 4` is a direct consequence of our assumption. Moreover, our stationarity assumption implies that the TV distance between target model and i-th drafter does not depend on the context and consequently all of the theorem does not have to take into account the target model output. Please note that our assumption is stricter than the most general case as in [1] while it is more general than [2] where authors use fixed acceptance rates. For completeness, we add the comparison between ours and assumptions used in [1] and [2] with analysis in `Appendix G.6`.
>
> # W3. Stochasticity of the target sequence length B
>
> Thank you for the constructive feedback! While we define the stopping time regret (`Definition 2`) and build `Theorem 2` with fixed B for each generation, this can be easily generalized to the more general case considering a probability space induced by the target model. This is possible since target sequence length B does not depend on the policy under our assumption where target sequence length B is only determined by the randomness of the target model, independent from the types of drafters we used throughout the generation. We include a general version of `Theorem 2` with analysis in `Appendix G.7`.
>
> # W4. The relationship between the reward / regret design and the performance of the algorithm.
>
> The performance of a bandit algorithm itself does not depend on the definition of the regret since regret objective is only for the analysis. We show that the optimality guarantee with the existing regret objective in the bandit literature will not prove the optimality of the algorithm performance in the Speculative Decoding. This motivates us to design a novel regret objective in `Definition 2` which aligns with the performance of algorithms in SD. This is reflected in `Lemma 6`, where we prove the upper bound on existing regret definition may not guarantee the optimal performance. One of our key theoretical contributions is defining this new regret objective and proving the upper bound on this new stopping time regret which is not so straightforward from the conventional analysis of UCB. Consequently, the result of `Theorem 2` reflects the guarantee of the algorithm performance and lower regret upper bound when using BD reward can imply better algorithm performance in most scenarios which we state in `Corollary 1`.
>
> # W5. Line 1586 and Line 1695
>
> We appreciate your careful review. We fixed typos in `line 1763` (originally `line 1586`) and `line 1878` (originally `line 1696`) of our original manuscript.
>
> For `Line 1586`, $\mathbb{E}[\tau^u(\pi^u,B)]$ should be changed to the $\mathbb{E}[n_{i^{\star}}(\pi^u,B)]$.
>
> For `Line 1878` (eq. 29), the square root term in the left hand side should be divided into $s$ and $n_i$, respectively. As in the proof of the original UCB algorithm (proof of `Lemma 7`), confidence bound in both inequalities needs to be shrinked as $s$ and $n_i$ increases with fixed t and it doesn’t necessarily have to be shrinked with increasing t.
>
> # W6. Hyperparameter $\beta$ and analysis on mean gaps $\Delta_i$
>
> Thank you for the insightful comments!
>
> In `Appendix G.8`, we include further analysis on how the regret upper bound of our algorithm changes with different $\beta$.
>
> Also, mean gaps $\Delta_i$ can indeed be a critical factor of the algorithm performance. We make terms containing mean gaps to appear in inequality in `Theorem 2` and relevant equations to better represent this fact which ensures all of the constant terms to be independent both on $B$ and mean gaps.
>
> # W7. Formal definition of B and terminology
>
> We provide a formal definition of target sequence length B in `Appendix G.7`. We have also ensured consistent use of the term ‘target sequence length’ when referring to $B$ throughout the manuscript to avoid any ambiguity.

---

> ### Author Response · Authors · 2024-11-20
> **Responses to Reviewer v7AR (2/2)**
>
> # Q1. The performance of EXP3
>
> While UCB algorithm is optimal in instance dependent regret bound with stationary assumption, EXP3 algorithm explores more on suboptimal arms. We also observed using EXP3 algorithm results in picking suboptimal drafters more compared to using UCB algorithm as our ablation study on the best arm ratio (`Section 4.3`) shows.
>
> # Q2. Additional baseline for mixed dataset
>
> We conducted additional experiments with an One-size-Fits-All (OFA) drafter for both black-box and white-box settings. These OFA drafters were trained on the mixed datasets spanning all tasks, ensuring a direct comparison to the specialized-drafter-based MetaSD framework. **The results in `Table3` and `Table4` of our revised manuscript show that while the OFA drafter performs well, our MetaSD framework outperforms OFA drafter across most tasks, demonstrating the strength of our method.** The OFA Eagle drafter performs relatively well. **However, it is still outperformed by MetaSD (i.e., MetaEagle-UCB)**.
>
> # Q3. Scaling law of the task numbers of MetaSD
>
> While our experiments focus on five tasks, we selected them to cover orthogonal areas (e.g., `code`, `translation`, `QA`, `summarization`, and `math`) to ensure diverse task representation. We acknowledge that five tasks may not fully capture the broader task spectrum. **However, this limitation is not unique to our framework but also affects single-drafter approaches.**
> To answer your request, we conducted out-of-domain experiments with Alpaca-Finance and RAG datasets in `Table 9` of `Appendix F.8`. `Table 9` demonstrates that even with only five specialized drafters, MetaSD consistently outperforms single-drafter approaches, including OFA drafters, under this scenario.
>
> **Reference**
>
> [1] M. Yin et al, A Theoretical Perspective for Speculative Decoding Algorithm. NeurIPS 2024.
>
> [2] Leviathan et.al., Fast Inference from Transformers via Speculative Decoding. ICML 2023.

---

> > ### Comment · Reviewer_v7AR · 2024-11-21
> >
> > Thank the authors for the detailed reply! It resolves some of my concerns, but others remain.
> >
> > **W1. Clarity of the assumption on acceptance rate**
> >
> > Thanks for re-writing the definitions of the acceptance rate. It is better to explicitly state that $\alpha_{i,t}$ are **independent** since only stationarity does not guarantee the correctness of Lemma 3.
> >
> > **W2. Validity of Theorem 1**
> >
> > Thanks for clarifying the setting. It is encouraged to directly state the i.i.d. assumption in the theorem to distinguish between the current work and all the previous works. In addition, it is beneficial to discuss the i.i.d. assumption as a simplification of the real applications.
> >
> > **W3. Stochasticity of the target sequence length B**
> >
> > Thanks for the new definition. Given the setting in Appendix G.7, Theorem 2 indeed implicitly makes a very strong assumption, i.e., $B$ is not a random variable. This assumption does not hold in any application setting according to my understanding, since the sampling decoding induces a stochastic process. It is encouraged to remove Theorem 2 and state Theorem 6 in the main paper.
> >
> > The proof of Theorem 6 is not correct. As mentioned in the review, if we condition on $B$, we should work with the posterior distribution of all the random variables conditioned on $B$. Thus, Lemma 8 is not correct. Let’s consider a very simple LLM, whose alphabet is ${0,1,EOS}$. The first token is $0$ or $1$ with $1/2$ probability. To predict the next token, if the last token is $1$, the next token is $0$ or $1$ with $1/2$ probability. If the last token is $0$, then the next token is $EOS$ with probability 1. Thus, if we condition on the event that $B=4$, we know that the sequence is $1110$, and they are not independent anymore. Thus, (18) **does not hold** for the posterior distribution. Assumption 1 only states that the acceptance rate is i.i.d. without any conditioned random variable, it **does not** imply anything for the posterior distribution.
> >
> > In addition, the statement ``Since probability space generated from the policy is independent from target model output generated from p under Assumption 1, ’’ is not correct. Only random variables can be independent, the probability space is a pre-defined structure, and there is no stochasticness for it.
> >
> >
> > **W4. The relationship between the reward / regret design and the performance of the algorithm.**
> >
> > There is a gap between the methods in the theory and the empirical algorithms. In line 371, the authors state that “We conduct evaluations using a NVIDIA A5000, A6000, and A100 GPU under **greedy** decoding settings.” However, all the theoretical analysis is built on the sampling decoding, i.e., we sample the next token from the distribution predicted by LLMs. The greedy decoding can be very different from the sampling decoding. Thus, the theoretical analysis is unrelated to the empirical results.
> >
> >
> > **W5. Line 1586 and Line 1695**
> >
> > Thanks for correcting your expressions. The expressions look reasonable now.
> >
> >
> > **W6. Hyperparameter $\beta$ and analysis on mean gaps $\Delta_i$**
> >
> > Thanks for incorporating $\beta$ in the final bound.
> >
> > However, as indicated by the theorem, $\beta$ should be chosen to be greater than $0.5$, whereas $\beta$ is selected to be $0.01$ in the experiment (Line 1150). In this case, the bonus term in the UCB design has a marginal effect. The whole algorithm design roughly reduces to a Exploration-then-Commit (ETC) algorithm which uniformly explores each drafter once followed by committing to the empirically best drafter.
> >
> > Since ETC is usually served as a benchmark in the bandits literature, it would be better if ETC is incorporated in the experiment, with a tuned exploration phase (maybe considering allocating $2(B/K)^{2/3}\ln^{1/3}(2KB)$ time steps for the uniform exploration, which results in a minimax bound for ETC).
> >
> > Lastly, as mentioned in **W3**, the proof of Theorem 7 is not correct.
> >
> > **W7. Formal definition of B and terminology**
> >
> > Thanks for your clarification.
> > It is confusing that in Line 1947, the authors indicate “We can relax the assumption on the fixed target sequence length B and naturally extend our analysis for the case where target model output is based on the temperature sampling T > 0.” Does it indicate the previous result only holds for the greedy decoding strategy? It would be appreciated if it can be further clarified.

---

> ### Author Response · Authors · 2024-11-22
> **Additional Responses to Reviewer v7AR (1/2)**
>
> Thank you once again for your thorough review. In response to your follow-up comments, we have further revised and updated our manuscript accordingly.
>
> **W1. Explicit statement of i.i.d. assumption**
>
> Thank you for the suggestion, in revised manuscript, we explicitly state that $\alpha_{i,t}$ are i.i.d from a distribution $\nu_i$ with mean $\alpha_i$.
>
> **W2. Explicit statement of i.i.d. assumption in Theorem**
>
> In the revised manuscript, we also directly state that $\alpha_{i,t}$ are i.i.d from a stationary distribution $\nu_i$ in `Theorem 2` to properly reflect on our i.i.d. assumption. Moreover, we will include a comparison for Assumptions in [1] and [2] in our main paragraph (`Section 3`) for further clearness.
>
> **W3. Stochasticity of the target sequence length B**
>
> We appreciate the constructive feedback and further clarification.
>
> First, we would like to clarify that `Assumption 1` is about assuming acceptance rates $\alpha_{i,t}$ are i.i.d. from the same distribution $\nu_i$ for each instance of generation.
> While this distribution $\nu_i$ can vary across generations and depend on the target sequence length B, this does not impact our algorithm's performance under `Assumption 1` because we reset the bandit instance for each new generation. Suppose a target mode (with the same rule in your example) generates 1110. Then the acceptance rate of a drafter $i$ conditioned on 1, 11, 111, 1110 follows from the same distribution $\nu_i$ with mean $\alpha_i$. Thus, the objective of a bandit algorithm here is to find the optimal drafter and minimize the regret within this instance and our analysis holds in this case. Next, suppose the target model generates 110 then EOS. Then, acceptance rates of drafter $i$ for this instance also follows stationary distribution with mean $\alpha_{i'}$ which can be different from $\alpha_i$, which does not affect our algorithm since the policy in the instance is independent of any previous results. In this sense, `Theorem 2` is not based on the assumption of fixed $B$ in generation, rather, the regret is defined for each instance of generation. Consequently, our analysis of `Theorem 2` remains valid under `Assumption 1`.
>
> However, to further enrich our results, we take the reviewer’s advice and include `Theorem 6` for the general analysis on the expected regret where expectation is taken over the probability space induced by the target model. This should be built on an additional assumption about assuming $\alpha_i$ being independent of B and its conditional expectation over a given $B$ being the same for every $B$. We explicitly state this assumption before Theorem 6 to avoid any confusion with `Assumption 1` and `Theorem 2`.
>
> For the last part, we correct our statement to "Since drafter selection from the policy $\pi$ is independent from $B$ under `Assumption 2`".
>
>
> **W4. Temperature sampling**
>
> Thank you for raising the important point.
>
> `Assumption 1` can hold with every $T$ under our formulation and the analysis holds in Greedy Decoding with `Assumption 1`. With this, our theoretical results include the case of $T=0$, greedy decoding. We inlcude this in our manuscript for avoid any confusion.
>
> The assumption is based on our empirical observation that TV distance between two probability distributions can be an intrinsic measure of closeness of the target model and a drafter and assumes acceptance rates of a drafter are i.i.d. drawn from the same distribution in each instance. This simplification allows us to investigate theoretical guarantee and reward design when using a bandit algorithm in multi-draft SD.
>
> Moreover, in addition to `Table 7` where we showed that our algorithm still performs strongly with $T>0$, we will include further experimentally results from the sampling decoding in all of our main experiments in our final manuscript to connect the theory and the experiments more robustly.

---

> ### Author Response · Authors · 2024-11-22
> **Additional Responses to Reviewer v7AR (2/2)**
>
> **W6. Comparison with ETC**
>
>  We appreciate you for the constructive feedback with great intuition.
>
> Indeed for a small $\beta$, a regret upper bound including $\beta$ does not hold anymore. We conducted additional experiments on translational task using ETC with tuned exploration rounds as the reviewer’s suggestion and the result is as follows:
>
> |    Speedup ratio        | JAPANESE  | RUSSIAN  | GERMAN  | FRENCH  | CHINESE  |
> |----------------|-------|-------|-------|-------|-------|
> | ETC (15)        | 1.443 | 1.548 | 1.870 | 2.117 | 1.521 |
> | UCB ($\beta=0.1$)  | 1.447 | 1.772 | 2.121 | 2.118 | 1.607 |
> | UCB ($\beta=0.01$) | 1.367 | 1.774 | 2.100 | 2.097 | 1.643 |
>
> In the above result, ETC(15) is based on the uniform exploration round of $15$ which we found performs best empirically. And exploration hyperparameter $\beta$ is $0.1, 0.01$ are used for a fair comparison. While ETC achieves comparable speed-up performance compared to UCB, ETC requires the number of exploration rounds as a hyperparameter and the optimal exploration round depends on the total target sequence length $B$ which we can’t know during the generation. We will include the discussion with updating the result into our final manuscript to include ETC as a baseline of our experiments.
>
> **W7. Clarification on the statement**
>
> Thank you for pointing out this. As stated in the additional response of W3, in order to relax the assumption on fixed B and make `Theorem 6` to be true, we have to rely on the additional assumption (`Assumption 2` in `Appendix G.7`). For clarification, we restate the previous claim into "We can consider general scenarios where we take all possible instances generated by a target model when using temperature sampling with T>0. In this scenario, we define the expected regret over the probability space induced by the target model." in our revised manuscript.
>
> **Additional comparison with [1] and clarification**
>
> We appreciate the reviewer for providing us good reference [1] which we didn’t have a chance to access by the time of our submission. We believe analyzing Speculative Decoding on the more general assumption of formalizing a decoding problem using a Markov-chain would be closer to the real-world scenarios. While formulating our problem as a non-stationary bandit with Markov chain formulation as in [1] would be a more realistic modeling, non-stationary bandit does not always perform better empirically.
>
> Here, we would like to emphasize once more that our work is the first to investigate multi-drafter selection within one generation when doing Speculative Decoding and we especially take advantage of a bandit algorithm which is simple yet effective. Thus future works are expected to investigate more general scenarios and corresponding analysis to cover more realistic modeling.
>
> We sincerely appreciate you for the constructive discussion and valuable feedback with excellent intuitions. We hope our responses clarify your initial concern and we are open to further discussion if there exist any remaining concerns or questions!
>
>
> **Reference**
>
> [1] M. Yin et al, A Theoretical Perspective for Speculative Decoding Algorithm. NeurIPS 2024.
>
> [2] Leviathan et.al., Fast Inference from Transformers via Speculative Decoding. ICML 2023.

---

> > ### Comment · Reviewer_v7AR · 2024-11-25
> >
> > Can you add more results on ood datasets largely different from the five domains you train the eagle, with both bsz> 1 and OFA eagle.
> >
> > I recommend you use some datasets like MMLU with COT, physics QA, hotpot QA.

---

> ### Author Response · Authors · 2024-11-25
>
> Dear Reviewer v7AR
>
> Thank you for your suggestion. We will do our best to conduct as many additional experiments as possible within the discussion timeline, including out-of-domain datasets like MMLU, Physics QA, and Hotpot QA, with batch size > 1 and OFA Eagle comparisons.
>
> That said, we feel this level of experimental demand is quite extensive compared to typical expectations, as similar works, even including concurrent studies at `ICLR`, are not usually evaluated across such a broad range of datasets.
>
> **Additionally, we would like to ask whether the lack of specific comments on the `theorem` and its implications is a critical factor in your decision to maintain the current score (`3: reject, not good enough`).** While we believe our responses (and revised manuscript) address the concerns raised,  `given the tight timeline`, we want to ensure we prioritize addressing the most impactful concerns if others remain.
>
> Warm regards,
>
> -Authors

---

> > ### Comment · Reviewer_v7AR · 2024-11-25
> >
> > Thanks for the reply! I still have the following concerns regarding the paper in its current form.
> >
> > **The correctness of the theoretical results**
> >
> > First, the setting of Theorem 2 does not contribute to the practical analysis. Theorem 2 assumes that the base LLM always outputs a token sequence with length $B$, which is highly unrelated to the practical applications. It is encouraged to discuss why we need to study this setting. Given the generalized version Theorem 6, it is better to remove Theorem 2 in the main paper.
> >
> > Second, Theorem 6 is **not correct**. As discussed, the authors only assume i.i.d. for the distribution without any conditioning, and it does not imply anything about the posterior distribution. The statement ``Suppose a target mode (with the same rule in your example) generates 1110. Then the acceptance rate of a drafter $i$ conditioned on 1, 11, 111, 1110 follows from the same distribution $\nu_i$ with mean $\alpha_i$.’’ is irrelevant to our discussion. We are **not** discussing anything about the independence between drafters, but the independence along the **decoding**, i.e., the **independence along the time**. Let us consider a even more simple problem. Each time decoding generates $1$ or $2$ tokens in an **i.i.d.** manner (Please keep in mind this is prior, i.e., without any conditioning). **Conditioned on the fact that 2 decoding procedures generate 4 tokens**, we know that both decoding procedures generate $2$ tokens. In fact, **the conditioning itself break the i.i.d. structure.** Thus, Lemma 8 is **not correct**.
> >
> > **The relationship between the theory and experiments**
> >
> > The theoretical analysis is **unrelated** to the experiments. As discussed, the whole theoretical analysis is all about stochastic decoding, but the experiments are all conducted on the greedy decoding. In the greedy decoding, **all the assumptions, i.e., i.i.d. assumption,  are wrong**, since this is no stochasticness anymore. I cannot see how the current analysis can be generalized to the greedy decoding method. According to my understanding, the i.i.d. assumption in the deterministic environment means that all the numbers are same. Then the best regret is constant, pulling all arms once and selecting the best one. Please explain this.
> >
> > **The representativeness of experiments**
> >
> > The throughput of Heterogeneous Batch and the OOD datasets.

---

> ### Author Response · Authors · 2024-11-25
> **Responses for the relationship between the theory and experiments (1/2) (To Reviewer v7AR)**
>
> Dear Reviewer v7AR,
>
> Thank you for your insightful questions. Below, we address your concerns about the `relationship between the theory and experiments.`
>
> We have conducted additional experiments with **stochastic decoding (temperature=1.0)**, as suggested, to demonstrate the robustness of our framework under the setting assumed in the theoretical analysis. The results are as follows:
>
> #### Black-Box (MetaSpS)
>
> | Task         | Drafter1 | Drafter2 | Drafter3 | Drafter4 | Drafter5 | OFA  | MetaSpS-UCB |
> |--------------|----------|----------|----------|----------|----------|------|------------|
> | Code         | 1.781    | 0.963    | 1.168    | 1.260    | 1.178    | 1.501 | **1.596**  |
> | Translation  | 0.856    | 1.695    | 0.897    | 0.880    | 0.838    | 0.861 | **1.197**  |
> | CNN          | 1.201    | 0.918    | 1.629    | 1.223    | 1.092    | 1.230 | **1.439**  |
> | NQA          | 1.073    | 0.961    | 1.132    | 1.510    | 1.031    | 1.123 | **1.322**  |
> | MathQA       | 1.220    | 1.026    | 1.200    | 1.360    | 1.968    | 1.512 | **1.673**  |
>
> #### White-Box (MetaEagle)
>
> | Task         | EAGLE1 | EAGLE2 | EAGLE3 | EAGLE4 | EAGLE5 | OFA Eagle | MetaEagle-UCB |
> |--------------|--------|--------|--------|--------|--------|-----------|---------------|
> | Code         | 3.019  | 0.872  | 1.012  | 1.348  | 1.809  | **2.926**     | 2.765     |
> | Translation  | 1.030  | 1.817  | 1.373  | 1.278  | 0.997  | 1.578     | **1.668**     |
> | CNN          | 0.998  | 0.834  | 2.289  | 1.267  | 0.864  | 1.749     | **1.935**     |
> | NQA          | 1.179  | 0.922  | 1.269  | 2.181  | 1.011  | 1.680     | **1.756**     |
> | MathQA       | 1.739  | 0.966  | 1.462  | 2.141  | 3.099  | 2.289     | **2.539**     |
>
> These results align with the theoretical assumptions, still demonstrating the robustness of MetaSD under stochastic decoding conditions. Additionally, `Table 7` in our manuscript already presents the robustness of MetaSD in multilingual settings under stochastic decoding. We believe these results further clarify the connection between our theory and experiments.
>
> To ensure clarity, we will expand on these findings in our camera-ready manuscript and address any potential ambiguities that may arise for other readers.

---

> ### Author Response · Authors · 2024-11-25
> **Responses for the relationship between the theory and experiments (2/2) (To Reviewer v7AR)**
>
> Dear Reviewer v7AR,
>
> Thank you again for your detailed feedback. We would like to address the concerns you raised in more detail and clarify our assumptions and theoretical results.
>
> ## Extrinsic Randomness of Model Alignments
>
> **First, we would like to emphasize that even in Greedy Decoding, randomness arises due to uncertainties in the alignment between the target model and the drafters.** Specifically, let $\alpha_{i,t}$ denote the acceptance rate of the $i$-th drafter at the $t$-th speculative round during one decoding instance. Even if the target model and drafter are deterministic in their outputs, it is practically impossible to access full information of $\alpha_{i,t}$ (i.e., exact alignment  or acceptance rates) in advance.
>
> For example,
>
> - The number of accepted tokens by a drafter depends on the specific overlap between its output and the target model's sequence at each step, which can vary across tokens.
>
> - Thus, before observing the outcomes, the system naturally treats the alignment as a random variable.
>
> We model the variation of number of accepted tokens (or $\alpha_{i,t}$) in each iteration as a random sequence i.i.d. from stationary distribution with mean $\alpha_i$ **in each decoding instance** (Please note that this does not imply that we assume i.i.d over several decoding instances). Therefore, our assumption is built from `extrinsic randomness` **arising from modeling the randomness of the alignment between the target model and a drafter**, which is orthogonal to considering the randomness coming from the stochasticity of the model outputs which we might call `intrinsic randomness`.
>
> `Assumption 1` and `Assumption 2` is about the `extrinsic randomness` which models uncertainties of how each drafter is aligned with the the target model and i.i.d assumption on the acceptance rates can be made within this scenario. And as a result, `Assumption 1` can hold in Greedy Decoding scenario even if there is no `intrinsic randomness` of the models.
>
> ## The Setting of Theorem 2
>
> **Regarding  `Theorem 2`, we would like to clarify once more that it does not assume fixed B of LLM output.** Instead, Theorem 2 analyzes the regret bound defined in **each instance of decoding** which is independent of the output of each generation (as explained in the follow-up response of `W3`).
>
> For example, suppose we conduct 2 decoding procedures (same rule as your new example) and generate 4 tokens. Then, the performance metric should measure how well the policy followed picking the best drafter in each decoding instance which does not related to the correlation between the model output and the target sequence length. Since our regret is analyzed on **each decoding instance**, first instance (B=2) should be measured by how well the algorithm performs compared to the optimal policy which is simply choosing the best drafter consecutively, and the second instance (B=2) should be measured by the same criterion. With the given $B=2$, the number of rounds $\tau(B)$ is still stochastic under our modeling of the 'extrinsic randomness'. Our stopping regret objective is thus to reduce $\tau(B)$ for each instance ($B=2$ for the first one and $B=2$ for the next one), **which is independent of the results of target model.** Even if the outputs of the two instances are different (or the same), this does not break the Theorem under the `Assumption 1` as we stated in the `Appendix G.6`.
>
> ## Independence of decoding instances and Theorem 6
>
> We would like to once more clarify that **we do not make assumptions about the independence of each decoding output**. Instead, we make assumptions on the independence of the acceptance rates for a given drafter along time for **each instance of decoding**. As stated in the above response (follow up response of `W4`), **`Theorem 6` is built with our additional assumption about assuming $\alpha_i$ being independent of $B$ and its conditional expectation over a given $B$ being the same for every $B$**. With `Assumption 2`, the acceptance rates follows i.i.d. conditioned on each B and therefore, `Theorem 6` remains valid under the `Assumption 2`.
>
>
> We will revise our manuscript accordingly to further clarify **our assumptions is built on the model uncertainties itself (extrinsic)** rather than the randomness of model outputs by temperature sampling (intrinsic). Also, we will further clarify Theorem with more explanation on the role of assumption.
>
>
> We hope these clarifications address your concerns. Thank you again for your active and constructive feedback and we are open to further discuss if there are any remaining concerns.
>
> Warm regards,
>
> Authors.

---

> ### Author Response · Authors · 2024-11-26
> **Follow-up Responses for OOD experiments (To Reviewer v7AR)**
>
> Dear Reviewer v7AR,
>
> We conducted additional evaluations regarding heterogeneous batch throughput and OOD datasets (adding to the previous `finance` and `RAG` OOD dataset evaluations), and the results are as follows:
>
> **Single Batch Throughput**
> | Task           | OFA Eagle | MetaEagle-UCB |
> |----------------|-----------|---------------|
> | Physics        | 2.424     | **2.573**         |
> | Hotpot QA      | 2.262     | **2.270**         |
> | MMLU COT (Avg. across 57 tasks)       | 2.466     | **2.529**         |
> | **Wins**       | **20**    | **37**        |
>
> **Heterogeneous Batch Throughput**
> | Task           | OFA Eagle | MetaEagle-UCB |
> |----------------|-----------|---------------|
> | MMLU COT       | **1.899**     | 1.826         |
>
> ## Analysis
>
> MetaEagle-UCB consistently outperforms OFA Eagle in single-batch scenarios across `Physics QA` and `Hotpot QA`. MetaEagle-UCB performs better on average across the 57 tasks in `MMLU-COT` and outperforms OFA Eagle in 37 individual tasks while trailing in 20 tasks (Here, MMLU with CoT reasoning follows the setting by Anthropic). For heterogeneous batch throughput, MetaEagle-UCB remains competitive due to using its specialized drafters (e.g., for QA, code, and math; at most 3 drafters in general), which excel at handling diverse tasks within the MMLU dataset.
>
> These findings suggest that future research could explore **how to structure specialization among drafters** to further enhance performance for throughput. For instance, incorporating two distinct types of OFA drafters trained on orthogonal domains could improve coverage across diverse task distributions, especially in `mixed or OOD scenarios`. This approach could also serve as a robust foundation for dynamically balancing specialization and generalization within speculative decoding frameworks.
>
> While the current specialization approach shows slight limitations in some OOD scenarios, our research focuses on proposing a **generic framework** rather than fixed specialization. We believe that future studies, emphasizing how to optimally construct and organize domains during training, could effectively address these limitations and further improve performance in diverse and challenging conditions.
>
>
> ## Clarifications on Research Focus
>
> Our work is not advocating for training multiple specialized drafters instead of OFA drafters. **The main focus is on effectively utilizing pre-existing heterogeneous specialized drafters, tackling the critical challenge of dynamically routing tasks among them.** This is a practical problem, especially as such specialized drafters are becoming widely available.
>
> - **We strongly recommend revisiting the `Motivation Section` and `Further Motivation Section` in `Appendix`.**
>
> ## Broader Implications
>
> This approach mirrors routing strategies in scalable systems like LoRA and MOE, where task-specific routing is essential (already widely used in the ML system of industries). Similarly, in speculative decoding, system-level routing frameworks like MetaSD have significant potential to enhance efficiency and flexibility.
>
> We hope this analysis provides clarity on the representativeness of our experiments and the broader implications of our framework.
>
> Warm regards,
>
> Authors

---

> > ### Comment · Reviewer_v7AR · 2024-11-26
> >
> > OFA Eagle achieves comparable performance to Bandit-Eagle while using less memory and without requiring the Bandit algorithm. I don't understand under what circumstances someone deploying an LLM would choose a more complex algorithm with unclear benefits, especially one that might even reduce throughput in real-world scenarios.

---

> ### Author Response · Authors · 2024-11-26
> **Responses to Reviewer v7AR**
>
> Thank you for your follow-up comment. **Respectfully, we believe there is a `fundamental misunderstanding` regarding the focus and intent of our work.**
>
> ## Respectfully Addressing the Concerns
>
> 1. **Research Focus**
>  > Our work **does not advocate for training multiple specialized drafters instead of OFA drafters.** Rather, it addresses a **critical and practical challenge**: how to effectively utilize pre-existing **heterogeneous specialized drafters** dynamically. This is a practical scenario as specialized drafters, trained on diverse datasets, are becoming increasingly available. **Routing tasks dynamically among these drafters** is the focus of our research.
> >   We strongly recommend revisiting the **Motivation Section** and **Further Motivation Section in the Appendix**, where we have explicitly detailed this objective, **please**.
>
> 2. **On MAB’s Complexity**
> >   We respectfully disagree with the notion that MAB is a complex algorithm. MAB is one of the **simplest and most traditional online learning approaches**, widely used in practical domains like **recommendation systems, finance, and medical systems** for its **adaptability and low computational overhead**. These strengths make it an ideal choice for speculative decoding in dynamic environments.
>
> 3. **Role of OFA in Routing**
> >   While OFA drafters are highly versatile, they are not guaranteed to perform optimally across all tasks. Notably, in our framework, OFA itself can be a **key component in the routing process, serving as a fallback drafter to handle tasks where specialized drafters may perform suboptimally.** This ensures robustness in scenarios where potentially low performance could arise. However, the intent of our paper is not to promote a single OFA drafter for all tasks, but to explore how **pre-existing multiple (specialized) heterogeneous drafters [1], including OFA,** can be used together dynamically to achieve better overall performance.
> >   The comparison with OFA Eagle in our experiments serves to **evaluate its effectiveness relative to MetaSD**, not to suggest that OFA should be excluded. Rather, OFA is complementary and could play a vital role within a routing framework to ensure robustness.
>
> - [1] Towards Fast Multilingual LLM Inference: Speculative Decoding and Specialized Drafters, Yi et al., EMNLP 2024.
>
> 4. **Throughput and Real-World Applicability**
> >   We have addressed throughput concerns in our experiments. In single-batch scenarios, **MetaSD-UCB consistently outperforms OFA Eagle**, particularly in **perturbed prompt settings**. In heterogeneous batch throughput, MetaSD-UCB remains competitive due to its ability to utilize task-specific drafters dynamically. This adaptability is essential for handling real-world, diverse queries effectively.
> >   Moreover, in multilingual translation or complex tasks like **MMLU-COT** and **Physics QA**, our framework dynamically leverages specialized drafters for specific tasks while complementing them with OFA as needed. **Routing ensures that each task benefits from the most appropriate drafter, whether specialized or OFA.**
> - **`We respectfully request not to frame throughput as the sole determining factor for evaluating real-world applicability.`** In practice, **single-batch inference** and **homogeneous batching** are commonly used in industry applications, especially for scenarios like low-latency, user-specific generation.
> - **`Additionally, if throughput is always the primary criterion, we would like to ask whether prior speculative decoding (spec-dec) papers—many of which are known to have lower throughput [2]—should also be discounted based on this metric alone.`** We believe such an approach overlooks the broader contributions and utility of speculative decoding frameworks, including improvements in latency, task adaptability, and system-level design.
>
> [2] Fast Inference from Transformers via Speculative Decoding, Leviathan et al.
>
> 5. **The Core Contribution**
> >   Our framework is a **generic and scalable solution** to the problem of **dynamic task routing among heterogeneous drafters.** It is not meant to replace OFA but to complement scenarios where pre-trained heterogeneous drafters exist—a situation that is increasingly common. This is a **system-level problem** that goes beyond a single-drafter approach and requires effective, adaptive solutions.
>
> ## Final Thoughts
>
> Respectfully, we believe the comments focus on a preference for OFA without fully engaging with the central problem our paper seeks to solve. OFA’s inherent limitations in non-stationary, multilingual, and domain-diverse settings underscore the importance of **dynamic task routing.** Furthermore, OFA itself can be effectively utilized within our framework to **guard against potentially low performance, reinforcing the versatility of MetaSD.**
>
> We welcome further constructive discussion and hope this response clarifies the contributions and implications of our work.
>
> -Authors

---

> ### Author Response · Authors · 2024-11-26
> **Dear SAC, AC, and Reviewer v7AR**
>
> Thank you for the ongoing discussion. We would like to raise a critical concern:
>
> - `Request for Moderation`: **To the AC, we are concerned that certain interpretations appear to frame the work unfairly, not due to a lack of engagement but rather through persistent misdirection of the discussion. This includes steering rebuttal efforts toward edge cases and issues tangential to the core problem our work aims to address, while also demonstrating a misunderstanding of the theorems `despite repeated clarifications`.** We kindly request your moderation to assess whether this critique appropriately reflects the context and intent of our contributions. If you find similar concerns, we sincerely ask for your guidance to ensure fairness.
>
> We appreciate your attention to this matter as we strive for a balanced and constructive review process.
>
> Best regards,
>
> -Authors

---

### Author Response · Authors · 2024-11-20
**Overall response**

We sincerely appreciate all the reviewers for their insightful and constructive feedback on our manuscript. We have responded to the individual comments from the reviews below, and believe that we have successfully responded to most of them. We have included the discussion and results of the suggested experiments in the revision. Here we briefly summarize `(1)` our core contributions, `(2)` strengths, `(3)` Empirical updates, and `(4)` updates on theories we have made to the revision.

# Core Contributions of Our Work

1. **Novel Framework** : We introduce MetaSD, a simple yet efficient framework which is the first work to tackle the problem of using multiple specialized drafters for Speculative Decoding (SD).

2. **Problem Formulation** : We formulate MetaSD as a Multi-Armed Bandit (MAB) problem and design a novel reward and regret objective which connects theories to the actual performance of the algorithm.

3. **Theoretical Guarantee** : By establishing an upper bound on the regret (Theorem 2), we prove that our algorithm achieves a `log-linear upper bound` which guarantees the faster LLM inference time.

4. **Experimental Results** : We demonstrate that our framework achieves superior speedup ratio in various tasks (`code generation`, `question answering`, `math reasoning`, `multilingual translation`, `RAG`, `Finance`). In most cases, our approach results in (a) `~60% speed-up` relative to SOTA SD and `~370% speed-up` relative to vanilla decoding on `7B` LLM.


# Summary of Strengths Highlighted by Reviewers

1. **Novelty and Effectiveness of Method** : `Reviewer v7AR`, `Reviewer Tyx3`, `Reviewer CxuG`, `Reviewer V45K` agreed that our framework MetaSD is a novel and effective approach for using multiple drafters in speculations.

2. **Writing and Presentation** : `Reviewer v7AR`, `Reviewer Tyx3`, `Reviewer CxuG` noted that our writing and presentation is clear and concise to read.

3. **Comprehensive experiments**: `Reviewer v7AR`, `Reviewer V45K`, `Reviewer CxuG` concurred that our experimental results are solid under different types of scenarios which validates our MetaSD framework.

4. **Potential insights to the academic community**: Recognized by `Reviewer V45K` for investigating multiple-drafter approaches might lead to meaningful findings. **Our work is the first to touch the problem of using multiple drafters in SD.**

# Updates of experimental results during Rebuttal

1. `Appendix F.8`: Evaluations on out-of-domain datasets including `finance` and `RAG`.

2. `Appendix F.9`: Evaluations on perturbed prompted scenarios.

3. `Table3` and `Table4` in `Section 4`: Experiments with One-size Fits All (OFA) drafters, which is trained on a mixed dataset across all tasks.

4. `Appendix F.10`: Throughput experiment.

5. Responses for `Reviewer V45K`: Extra computation overhead.

6. `Appendix F.11`: MetaEagle-UCB with Efficient KV Cache Strategies [A]

7. Responses for `Reviewer V45K`: Switching times during MetaSD.

# Updates of theoretical parts during Rebuttal

1. `Appendix G.6`: Clarifying our assumption on acceptance rate with comparison between previous literature.

2. `Appendix G.7`: Generalized analysis of regret upper bound considering randomness of target sequence length $B$.

3. `Responses for Reviewer v7AR`: Validity and robustness of our theoretical framework under our formulation.

# Further research motivation

We want to emphasize that MetaSD addresses another line of practical challenge: managing diverse, heterogeneous drafters from open-sourced systems (e.g. HuggingFace). These drafters, pre-trained with varying objectives and frequently lacking detailed training documentation, pose significant obstacles to deployment frameworks that assume uniformity or rely on static selection strategies. By providing our adaptive strategy to optimally utilize such resources, MetaSD ensures robust performance across diverse tasks and settings, contributing meaningfully to both theoretical advancements and real-world applications.

--------

We believe these additions and clarifications address the reviewers' concerns comprehensively and strengthen our manuscript. The changed parts in our revised manuscript are highlighted in `magenta-colored` text. Our manuscript is updated on `Nov 20, AOE time`. We look forward to your favorable consideration.


[A] EFFICIENT STREAMING LANGUAGE MODELS WITH ATTENTION SINKS, ICLR 2024.

---

> ### Author Response · Authors · 2024-11-21
> **TL;DR - Summary of Our Work (Key Gist)**
>
> - `Theoretical guarantee`: Provides a solid theoretical foundation for multi-armed bandit-based drafter selection in speculative decoding.
>
> - `First of its kind`: The first work to address speculative decoding with multiple heterogeneous drafters, tackling real-world challenges of diverse task settings.
>
> - `Extensible Design`: Easily integrates with other speculative decoding frameworks, offering compatibility and flexibility for future research.
>
> - `Robustness`: Demonstrates consistent performance across diverse scenarios, including in-domain, out-of-domain, and perturbed prompts, outperforming prior work in single-batch settings.
>
> - `Throughput limitation`: Acknowledges limitations in heterogeneous large-batch scenarios but remains highly effective for small batch sizes.
>
> - `Novel analysis`: Introduces a new form of divergence metric analysis, shifting focus from block efficiency to token-level adaptation and drafter selection.
>
> This work lays the groundwork for further exploration in multi-drafter speculative decoding while addressing practical and theoretical challenges.

---

### Meta-Review · Area_Chair_e6Bk · 2024-12-13

**Metareview:**

This paper introduces MetaSD, a novel framework that enhances Speculative Decoding (SD) by integrating multiple specialized drafters. Utilizing a multi-armed bandit sampling technique, it dynamically selects the most effective drafter for each inference step. The authors conducted experiments comparing their approach against both black-box and white-box SD methods, demonstrating that MetaSD significantly outperforms traditional single-drafter techniques, such as Lookahead and Eagle, in terms of efficiency and effectiveness.

The reviewers have expressed significant concerns regarding the correctness of the theorem presented in the paper, as well as the practical applications of the proposed method in real-world scenarios. The authors are encouraged to provide additional clarification on these points.

**Additional Comments On Reviewer Discussion:**

The reviewers have expressed significant concerns regarding the correctness of the theorem presented in the paper, as well as the practical applications of the proposed method in real-world scenarios. The authors are encouraged to provide additional clarification on these points.

---

### Decision · Program_Chairs · 2025-01-22

Reject